## OPEN

# Wildfire-dependent changes in soil microbiome diversity and function

Amelia R. Nelson [1], Adrienne B. Narrowe[1,2], Charles C. Rhoades[3], Timothy S. Fegel[3], Rebecca A. Daly [1], Holly K. Roth[4], Rosalie K. Chu [5], Kaela K. Amundson[1], Robert B. Young [6], Andrei S. Steindorff [7], Stephen J. Mondo [7,8], Igor V. Grigoriev [7,9], Asaf Salamov[7], Thomas Borch [1,4,10] and Michael J. Wilkins [1]✉

Forest soil microbiomes have crucial roles in carbon storage, biogeochemical cycling and rhizosphere processes. Wildfire season length, and the frequency and size of severe fires have increased owing to climate change. Fires affect ecosystem recovery and modify soil microbiomes and microbially mediated biogeochemical processes. To study wildfire-dependent changes in soil microbiomes, we characterized functional shifts in the soil microbiota (bacteria, fungi and viruses) across burn severity gradients (low, moderate and high severity) 1 yr post fire in coniferous forests in Colorado and Wyoming, USA. We found severity-dependent increases of Actinobacteria encoding genes for heat resistance, fast growth, and pyrogenic carbon utilization that might enhance post-fire survival. We report that increased burn severity led to the loss of ectomycorrhizal fungi and less tolerant microbial taxa. Viruses remained active in post-fire soils and probably influenced carbon cycling and biogeochemistry via turnover of biomass and ecosystem-relevant auxiliary metabolic genes. Our genome-resolved analyses link post-fire soil microbial taxonomy to functions and reveal the complexity of post-fire soil microbiome activity.

Changes in climate coupled with the effects of long-term fire suppression and shifting land use patterns have increased the frequency, severity and season length of wildfires in the western United States[1–3]. In 2020 and 2021, the western United States experienced severe, record-breaking wildfires[2]. High-severity wildfires cause greater erosion[4], soil carbon (C) and nitrogen (N) losses[5], and nutrient and sediment export in stream water[6], so the increasing occurrence of severe wildfires may have important consequences for both terrestrial and aquatic ecosystems. Shifting wildfire patterns have also been linked to slow post-fire revegetation and tree seedling recruitment[7] and thus delayed watershed recovery[8] in western US forests. Although ecosystem recovery from severe wildfires is closely linked to belowground biological processes, little is known about the impact of high-severity fire on soil microbiome function in high elevation, coniferous ecosystems.

The soil microbiome regulates soil organic matter (SOM) decomposition and stabilization[9], soil nutrient dynamics[10] and rhizosphere function[11]. During wildfires, the soil microbiome can be impacted immediately by the loss of heat-sensitive taxa and thereafter by lasting changes in soil chemistry and vegetation shifts[12]. Wildfires reduce soil microbial biomass and community diversity in numerous ecosystems[13–16] and such changes probably influence and inhibit post-fire plant recovery[17].

Post-fire shifts in soil microbiome composition[14,18,19] and assembly processes[20–22] are relatively well-characterized across different ecosystems, with some studies explicitly linked with corresponding shifts in microbially mediated C and N cycling[23–26]. This work

has been complemented by laboratory studies with pure cultures of pyrophilous taxa that demonstrate their ability to persist during stressful conditions[27,28] and utilize aromatic C[29–32]. Metagenomic approaches can bridge insights between field-based compositional analyses and more controlled laboratory studies. So far, two studies have applied gene-resolved metagenomic analyses to post-fire soils[23,33]. Genome-resolved metagenomic tools can link potential pyrophilous traits (for example, fast growth rate, heat resistance) to specific organisms that thrive in burned soils and support laboratory observations[34]. Furthermore, this approach enables a broader understanding of microbiome function through identification of co-occurring functional traits, potential interspecies interactions, and viral-host dynamics.

Here we bridge laboratory studies and field-based compositional investigations through a genome-resolved multi-omic approach to characterize wildfire impacts on soil microbiome function. Furthermore, the work represents a holistic understanding of the post-fire soil microbiome, including comprehensive characterization of interacting bacterial, fungal and viral communities. We studied burn severity gradients in two recent forest wildfires to characterize how fire severity influences C composition and the intimately connected soil microbiome. We hypothesized that higher-severity wildfire results in an increasingly altered soil microbiome and that taxa colonizing burned soils would encode functional traits that favour their persistence. These analyses advance the understanding of linkages between the soil microbiome and post-fire forest biogeochemistry.

[1]Department of Soil and Crop Sciences, Colorado State University, Fort Collins, CO, USA. [2]Eastern Regional Research Center, Agricultural Research Service, Wyndmoor, PA, USA. [3]Rocky Mountain Research Station, U.S. Forest Service, Fort Collins, CO, USA. [4]Department of Chemistry, Colorado State University, Fort Collins, CO, USA. [5]Environmental Molecular Sciences Laboratory, Pacific Northwest National Laboratory, Richland, WA, USA. [6]Chemical Analysis and Instrumentation Laboratory, New Mexico State University, Las Cruces, NM, USA. [7]Department of Energy Joint Genome Institute, Lawrence Berkeley National Laboratory, Berkeley, CA, USA. [8]Department of Agricultural Biology, Colorado State University, Fort Collins, CO, USA. [9]Department of Plant and Microbial Biology, University of California Berkeley, Berkeley, CA, USA. [10]Department of Civil and Environmental Engineering, Colorado State University, Fort Collins, CO, USA. ✉e-mail: Mike.Wilkins@colostate.edu

## Results

**Fire decreases soil microbiome diversity and shifts composition.**
Near surface soils (0–5 cm depth) were collected approximately 1 yr post fire from four burn severity gradient transects (control, low, moderate and high burn severity) at two wildfires that occurred in 2018 along the Colorado–Wyoming border (Extended Data Fig. 1). Bacterial and fungal communities were profiled using marker gene analyses, while a subset of 12 samples (low or high severity-impacted Ryan fire soils) were additionally interrogated with metagenomic and metatranscriptomic sequencing. Bacterial and fungal communities were significantly different between burned ($n = 144$) and unburned ($n = 32$) soils (bacterial analyses of similarity (ANOSIM) $R = 0.57$, $P < 0.05$; fungal ANOSIM $R = 0.72$, $P < 0.05$) (Supplementary Fig. 2).

While shifts in community composition with burn were observed in both surface (0–5 cm) and deep (5–10 cm) soils (Supplementary Note 3 and Extended Data Fig. 2), surface soils were impacted to a greater extent. Microbial diversity generally decreased with increasing severity in surface soils, although differences between moderate and high severity were statistically indistinct (Fig. 1). Similarly, as fungal and bacterial diversity decreased with burn severity, beta dispersion ('distance to centroid') calculations revealed increasingly similar bacterial communities (Supplementary Fig. 3) with less complex community structures (via WGCNA; Supplementary Note 2, Supplementary Table 3). These shifts resulted in significant dissimilarity between microbial communities in surface soils impacted by either low ($n = 24$) or high ($n = 24$) severity wildfire (bacterial ANOSIM $R = 0.15$, $P < 0.05$; fungal ANOSIM $R = 0.25$, $P < 0.05$). In contrast, deep soils displayed an opposite effect, with increasing beta dispersion after wildfire signifying greater bacterial community dissimilarity (Supplementary Fig. 3). Stochastic community shifts in deep soils may follow a wildfire, potentially due to spatially heterogeneous changes in soil chemistry and nutrient availability. Combined amplicon sequencing data analyses highlight the susceptibility of surface soils to wildfire, resulting in less diverse and inter-connected microbial communities. In contrast, the microbiome in deep soil displays a more muted response to wildfire, potentially due to insulation from soil heating (dependent on soil moisture).

**A comprehensive dataset from fire-impacted soils.** While myriad studies have reported changes in microbial community composition following a wildfire[14,18,35], the functional implications of these shifts are difficult to infer from compositional data. We used genome-resolved metagenomics to generate a comprehensive, publicly accessible catalogue of post-fire bacterial, fungal and viral genomes from coniferous forest soils. From metagenomic sequencing of burned (low and high severity) soils, we reconstructed 637 medium- and high-quality bacterial metagenome-assembled genomes (MAGs) (Extended Data Fig. 3) that represent taxa shown to increase following a wildfire in complementary 16S ribosomal RNA gene sequencing data (for example, *Blastococcus*, *Arthrobacter*; Supplementary Note 1). The dataset spans 21 phyla and encompasses 237 MAGs from taxa within the Actinobacteria, 167 from the Proteobacteria, 62 from the Bacteroidota and 52 from the Patescibacteria. Furthermore, we recovered 2 fungal genomes from the Ascomycota, affiliated with *Leotiomycetes* and *Coniochaeta lignaria*. We additionally recovered 2,399 DNA and 91 RNA viral populations (vMAGs) (Supplementary Data 5).

**Actinobacteria respond strongly to high-severity wildfire.** On the basis of consistent high relative abundances across surface soils impacted by high-severity wildfire ('High S') that mirrored 16S rRNA gene data (Supplementary Note 1), 40 MAGs were selected for further genomic analyses. Combined, these MAGs accounted for an average relative abundance of ~60% in High S samples and

~34% in low severity-impacted surface soils ('Low S') and collectively represent the most abundant MAGs responding to altered soil conditions 1 yr post wildfire. Metatranscriptomic read mapping revealed activity of these MAGs in High S samples, accounting for an average of ~50% of total gene expression and 90% of differentially expressed genes in High S vs Low S soils (Supplementary Data 4). These MAGs were also active in Low S samples, albeit to a lesser extent (accounting for ~30% of gene expression). Most of these MAGs (28 of 40) were affiliated with the Actinobacteria phyla, specifically the genera *Arthrobacter* (8 MAGs), *Blastococcus* (5) and *SCTD01* (5) (Supplementary Data 2). Ten of these MAGs (Supplementary Data 2, Sheet D), including 9 Actinobacteria, were significantly enriched in High S relative to Low S samples (pairwise *t*-test, $P < 0.05$; Extended Data Fig. 4), indicating a positive response 1 yr following high-severity wildfire. In general, Actinobacteria dominated the microbiome in burned surficial soils; all 237 Actinobacteria MAGs were responsible for ~56% of gene expression in High S samples and ~47% in Low S samples. Dominant MAGs in high severity-impacted deep samples ('High D') were more diverse (representing Actinobacteria, Eremiobacterota, Acidobacteriota and Proteobacteria), reflecting the probably more heterogeneous impact of wildfire on deeper soils (Supplementary Note 2).

The heat produced during wildfire exerts a pulse disturbance on soils and as such, the relative abundance of two groups of thermal resistance genes—sporulation and heat shock—increased significantly (Welch's *t*-test, $P < 0.05$) from Low S to High S soils (42.8% and 20.4% increase, respectively). Nearly all the aforementioned MAGs (38/40) encoded sporulation genes, indicating that spore formation is probably a trait supporting survival and post-fire colonization. Many genomes (31/40) encoded heat shock proteins and molecular chaperones to further facilitate thermal resistance. In 16 MAGs, thermal resistance was complemented by genes for mycothiol biosynthesis, mycothiol being a compound produced by Actinobacteria that aids in oxidative stress tolerance[36]. Genes for osmoprotectant (trehalose, *otsAB*, *treZY*[37]; glycine betaine, *betAB*[38]) synthesis were widespread among these 40 MAGs (17 and 38 MAGs encoded trehalose and glycine betaine synthesis genes, respectively), which could facilitate cell viability under low soil moisture conditions post fire. We recognize that many well-studied soil taxa encode genes for similar traits, but note that combinations of these traits are probably an emergent property of fire disturbance supporting post-fire dominance of these taxa. Further, MAGs recovered from High S samples also had significantly higher guanine-cytosine (GC) content, which has been linked to thermal stability[39,40], than MAGs from fire-impacted deeper soils (Extended Data Fig. 5; pairwise *t*-test, $P < 0.05$). The lysing of microorganisms during soil heating represents sources of labile organic C and N associated with necromass[41]. All 40 featured MAGs expressed peptidase genes (2,721 total) in High S soils, of which approximately 41 were differentially expressed ($P < 0.05$) between High S and Low S samples. These included genes responsible for peptidoglycan (component of bacterial cell walls) degradation, suggesting that taxa enriched post fire actively utilize microbial necromass.

The ability to grow quickly and occupy newly available niches is probably a key trait for microorganisms colonizing or growing in burned soils[18,26]. We inferred maximum growth rates using codon usage bias across our bacterial MAGs to determine whether colonizing taxa encoded the potential for rapid growth[42,43] (Supplementary Data 2). After removal of MAGs with doubling times >5 h due to model inaccuracies at slower growth rates[42], the average doubling time within our MAG dataset was found to be ~3.2 h. Twenty-two of the 40 MAGs of interest in High S samples had doubling times faster than the dataset average (ranging from ~0.3 to 4.7 h). Further, there was a significant negative correlation (Spearman's $\rho = -0.18$, $P < 0.05$) between MAG relative abundance in High S samples and growth rate (measured as maximum doubling time), indicating that

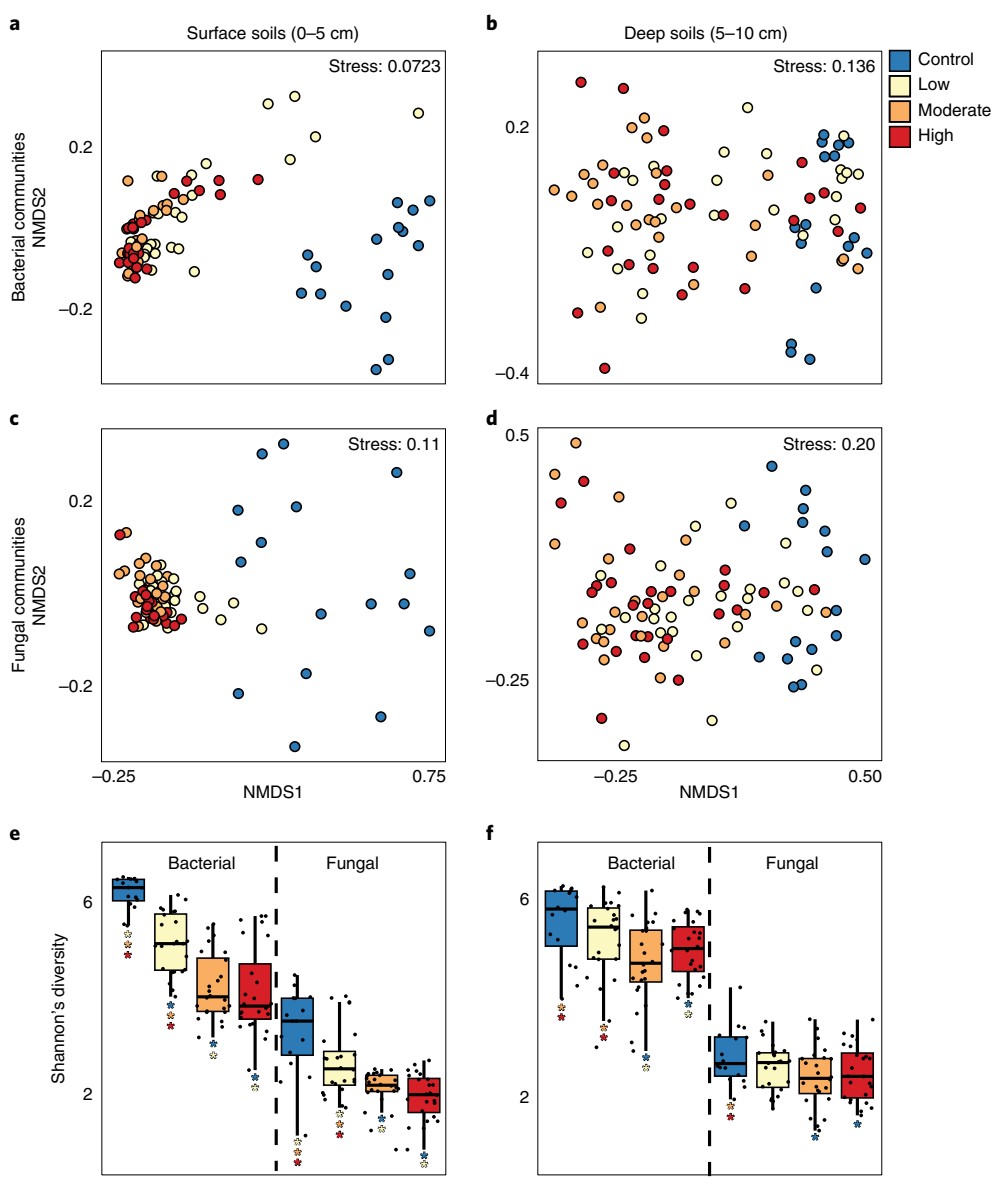

**Fig. 1 | Surface soil microbiome undergoes homogenizing effect with burn. a–d**, NMDS of surface (0–5 cm) (**a**,**c**) and deeper (5–10 cm) soil (**b**,**d**) bacterial (**a**,**b**) and fungal (**c**,**d**) communities shows increased separation of burned and unburned microbial communities in surface soils relative to deep soil communities. **e**,**f**, Shannon's diversity (H) calculated from 16S rRNA and ITS gene sequencing in surface (**e**) and deep soils (**f**) further shows the increased susceptibility of microbiomes in surface soils to wildfire. Asterisks in **e** and **f** denote significant differences (pairwise t-test; $P < 0.05$) between conditions ($n = 16$ for control S and D, $n = 24$ for low, moderate and high severity-impacted S and D samples). Corresponding P values are listed in Supplementary Table 1. The lower and upper hinges of the boxplots represent the 25th and 75th percentiles, respectively, and the middle line is the median. The whiskers extend from the median by 1.5× the interquartile range. Data points represent outliers.

High S conditions may select for microorganisms that can grow quickly (Fig. 2a). These insights suggest that abundant bacteria sampled 1 yr post wildfire occupied niches in the immediate aftermath of wildfire through strategies that probably include rapid growth. In contrast, these patterns were absent from MAGs recovered from other conditions (Fig. 2b–d). Emphasizing the importance of fast growth for colonizing severely burned soils, only 19 MAGs from High S samples had growth rates too slow to accurately estimate (249 MAGs with growth rates >5h). To determine whether these same microorganisms were growing rapidly at the time of sampling (1 yr post wildfire), we investigated gene expression associated with rapid growth[44,45] (ribosomes, central metabolism) through MAG abundance-normalized transcripts (Supplementary Data 4). Results suggested diminished growth rates for the dominant High

S bacteria at the time of sampling relative to other Actinobacteria MAGs that were highly expressing ribosomal and tricarboxylic acid cycle genes in High S samples. Together, these analyses indicate that potential rapid growth could enable these microorganisms to occupy free niche space in soil immediately following a wildfire, but this strategy may not be maintained once those niches are filled.

**Actinobacteria process pyrogenic organic matter.** During wildfire, SOM may be transformed to increasingly aromatic molecular structures that are commonly considered less available for microbial utilization[46]. Similar to other studies[47], mass spectrometry analyses of dissolved organic matter (DOM) revealed severity-dependent aromaticity increases in surface soils 1 yr post fire (Fig. 3). These aromaticity index trends were absent in DOM from more insulated

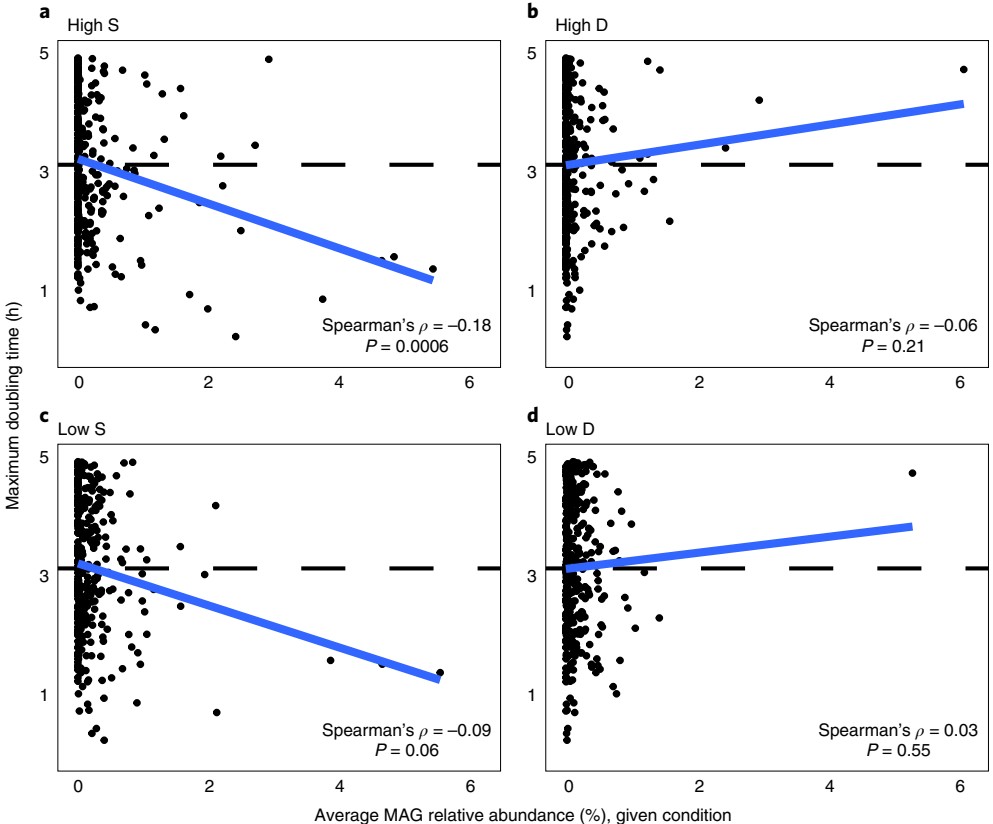

**Fig. 2 | Potential fast growth rate favoured in soils impacted by high-severity wildfire. a–d**, High S conditions (**a**) favour MAGs from organisms with faster potential growth rates (lower maximum doubling time, estimated using gRodon[42]), indicated here by a significant negative correlation (two-sided Spearman's rho test; Spearman's $\rho = -0.18$, $P < 0.05$). This trend is not present in the other three conditions (**b**–**d**). MAG average maximum doubling time is shown by the dashed line.

deep soils (Supplementary Fig. 4). Low-severity wildfire drives an increase in DOM aromaticity but also an accumulation of other unique compounds probably from incomplete combustion of SOM[48,49], whereas moderate and high-severity wildfire in surface soils resulted in the formation of unique aromatic organic compounds (Fig. 3a). The microbial transformation of these compounds is constrained by solubility and thermodynamic thresholds established by available electron acceptors[50] (for example, oxygen). To estimate the potential thermodynamic favourability of this DOM, we calculated the nominal oxidation state of carbon (NOSC); higher NOSC values theoretically yield a lower $\Delta G_{Cox}$ (that is, more favourable) when coupled to reduction of an electron acceptor[51]. Unique formulas in High S samples had significantly higher NOSC values, indicating increasing thermodynamic favourability for oxidation of DOM following severe wildfire (Fig. 3c; pairwise $t$-test, $P < 0.05$). Thus, thermodynamic limitations probably do not influence the lability of pyrogenic DOM in this system and other factors such as solubility or microbial community function probably govern compound processing.

We focused on microbial processing of catechol and protocatechuate—two intermediate products formed during aerobic degradation of diverse aromatic compounds[52]. The genomic potential for these reactions was present across severities and soil depths, and was dominated by Actinobacteria and Proteobacteria (Fig. 4); 80 and 226 MAGs encoded >50% of the catechol and protocatechuate ortho-cleavage pathways, respectively, including most of the featured High S and High D MAGs (Fig. 4c). Meta-cleavage pathways were also broadly represented within the MAGs (Extended Data Fig. 6). In High S samples, the *Arthrobacter* MAG RYN_101 alone

was responsible for ~44% of *catA* (catechol 1,2-dioxygenase) gene expression, and therefore probably plays a key role in catechol degradation. Contrastingly, in High D samples, the Streptosporangiaceae MAG RYN_225 was responsible for ~46% and 23% of expression of *pcaGH* (protocatechuate 3,4-dioxygenase) and *pcaC* (4-carboxymuconolactone decarboxylase), respectively, that catalyses protocatechuate degradation (Fig. 4c). However, no MAGs of interest from High S or High D samples encoded the entire catechol or protocatechuate ortho-cleavage pathway (Fig. 4c), indicating that metabolic hand-offs between community members are probably important for complete compound degradation. Outside of catechol and protocatechuate, there was genomic evidence for the benzoyl-CoA and phenylacetyl-CoA oxidation pathways (Extended Data Fig. 7). These data indicate that post-fire soils support microbiomes that actively degrade some fire-derived aromatic compounds and have implications for C storage in wildfire-impacted ecosystems, since pyrogenic C compounds are considered largely resistant to decay and contribute to C storage[53]. Further work should integrate multi-omics data from field and laboratory studies into ecosystem models to refine the quantification of post-fire C fluxes.

**Viruses impact burned soil microbiome structure and function.**
We recovered 2,399 distinct DNA and 91 distinct RNA viral populations (vMAGs) from the metagenomic and metatranscriptomic assemblies. Of these, 945 were previously undescribed (only clustering with other vMAGs from this study) and 92 were taxonomically assigned, with the majority ($n = 86$) within the *Caudovirales* order (Supplementary Data 5). DNA and RNA viral communities mirrored beta diversity trends observed in bacterial and fungal

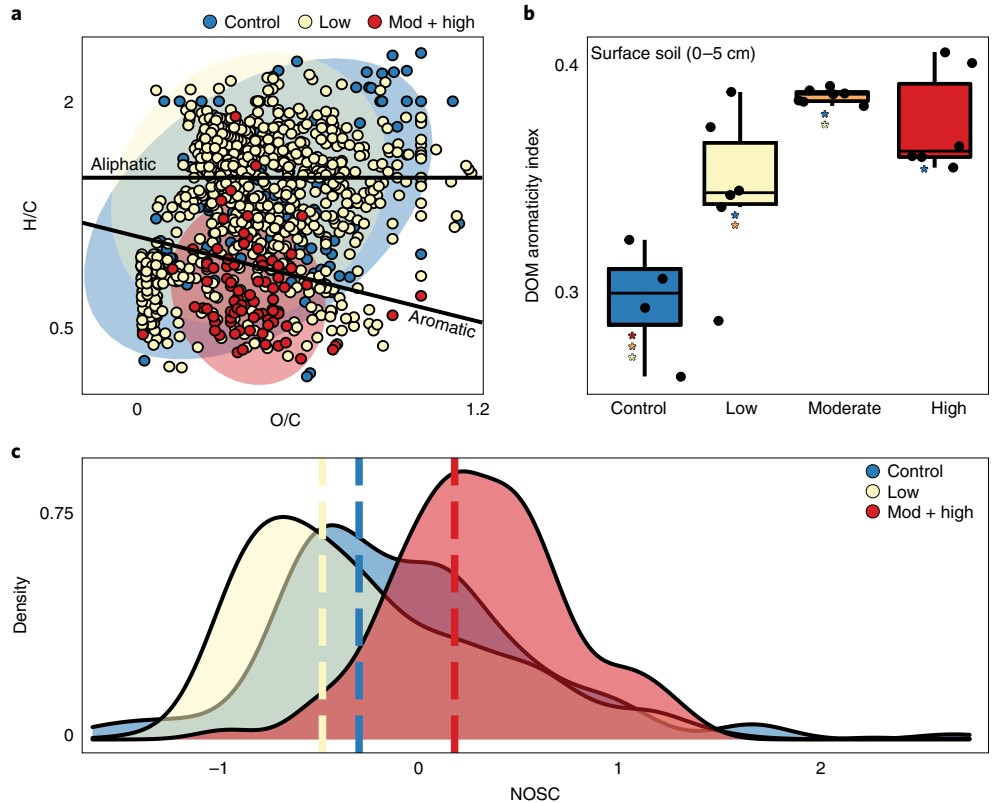

**Fig. 3 | Pyrogenic dissolved organic matter becomes increasingly aromatic with wildfire burn severity. a**, Van Krevelen diagram showing unique formulas in unburned, low, and moderate and high (combined) surface soils. **b**, Aromaticity index of DOM pools extracted from surface soils across the burn severity gradient ($n = 4$ for control, 6 for low, 5 for moderate and 6 for high severity). Corresponding $P$ values are shown in Supplementary Fig. 4b from one-sided pairwise $t$-test. The lower and upper hinges of the boxplot represent the 25th and 75th percentiles, respectively, and the middle line is the median. The whiskers extend from the median by 1.5× the interquartile range. Data points represent outliers. Coloured asterisks indicate significant difference between the two conditions (pairwise $t$-test, $P < 0.05$). **c**, Density plot of unique formula NOSC value distributions between different conditions. Dashed line shows NOSC median for each condition.

communities; those in deep soils were less homogeneous compared with communities in surface soils, further highlighting the homogenizing influence of wildfire (Supplementary Fig. 5). Additionally, although DNA and RNA viral community composition was indistinct between low and high severity-impacted soils (ANOSIM $R = 0.007$ and $-0.12$, respectively; $P > 0.1$), we did measure significant differences between the two soil depths (ANOSIM $R = 0.59$ and 0.57, respectively; $P < 0.05$).

Given the importance of viral activity on soil microbiomes[54], we identified potential virus-host linkages that could offer insights into how viruses target bacteria. Many abundant and active MAGs ($n = 94$)—including 32 from the Actinobacteria—encoded CRISPR-Cas arrays with an average of ~18 spacers (max 210 spacers; Supplementary Data 2). By matching CRISPR spacers to protospacers in vMAGs, we linked 9 vMAGs with 4 bacterial hosts (RYN_115, RYN_242, RYN_436 and RYN_542) from the Actinobacteria, Planctomycetota and Proteobacteria. While each of these MAGs were active (expressing transcripts), the RYN_242 MAG (Solirubrobacteraceae) was among the top 3% most active MAGs across all conditions, suggesting that viruses are targeting active bacteria. We expanded upon potential virus-host linkages using VirHostMatcher[55] ($d_2^*$ value $< 0.25$), revealing higher numbers of viral linkages with more abundant host MAGs (Fig. 5). For example, the High S and High D MAGs of interest had above average numbers of putative viral linkages (average of 278 compared with the dataset-wide average of 196). Moreover, 129 vMAGs were linked to all 28 featured Actinobacteria MAGs from High S samples,

potentially due to conserved nucleotide frequencies. These shared 129 vMAGs comprised ~7.6% of the viral community in High S samples, again suggesting that abundant and active bacteria in burned soils are actively targeted by abundant phage, potentially impacting soil C cycling via release of labile cellular components following cell lysis (that is, viral shunt)[56]. There is also evidence for the 'piggyback-the-winner' viral strategy, where lysogenic lifestyles are favoured at high microbial abundances and growth rates[57]. Of our 2,399 DNA vMAGs, 185 had putative lysogenic lifestyles based on gene annotations for integrase, recombinase or excisionase genes, and 25 of these had nucleotide frequency-based linkages to all the featured High S Actinobacteria MAGs.

To investigate potential viral roles in post-fire soil C cycling, we characterized the putative auxiliary metabolic genes (AMGs) repertoire of the vMAGs. Viruses use AMGs to 'hijack' and manipulate host metabolism; one permafrost soil study found AMGs associated with SOM degradation and central C metabolism, suggesting that viruses play a direct role in augmenting soil C cycling[54]. There were 773 total putative AMGs detected in 445 vMAGs, including 138 CAZymes targeting diverse substrates (for example, cellulose, chitin, pectin; Supplementary Data 5). Additionally, the AMGs included 105 genes related to growth (for example, ribosomal proteins, ribonucleoside-diphosphate reductase), 21 central C metabolism genes and 21 peptidases. Over 50 of these genes—including some related to SOM and necromass processing (for example, glycoside hydrolases, polysaccharide lyases) and cell growth (pyrimidine ribonucleotide biosynthesis)—were encoded within viral

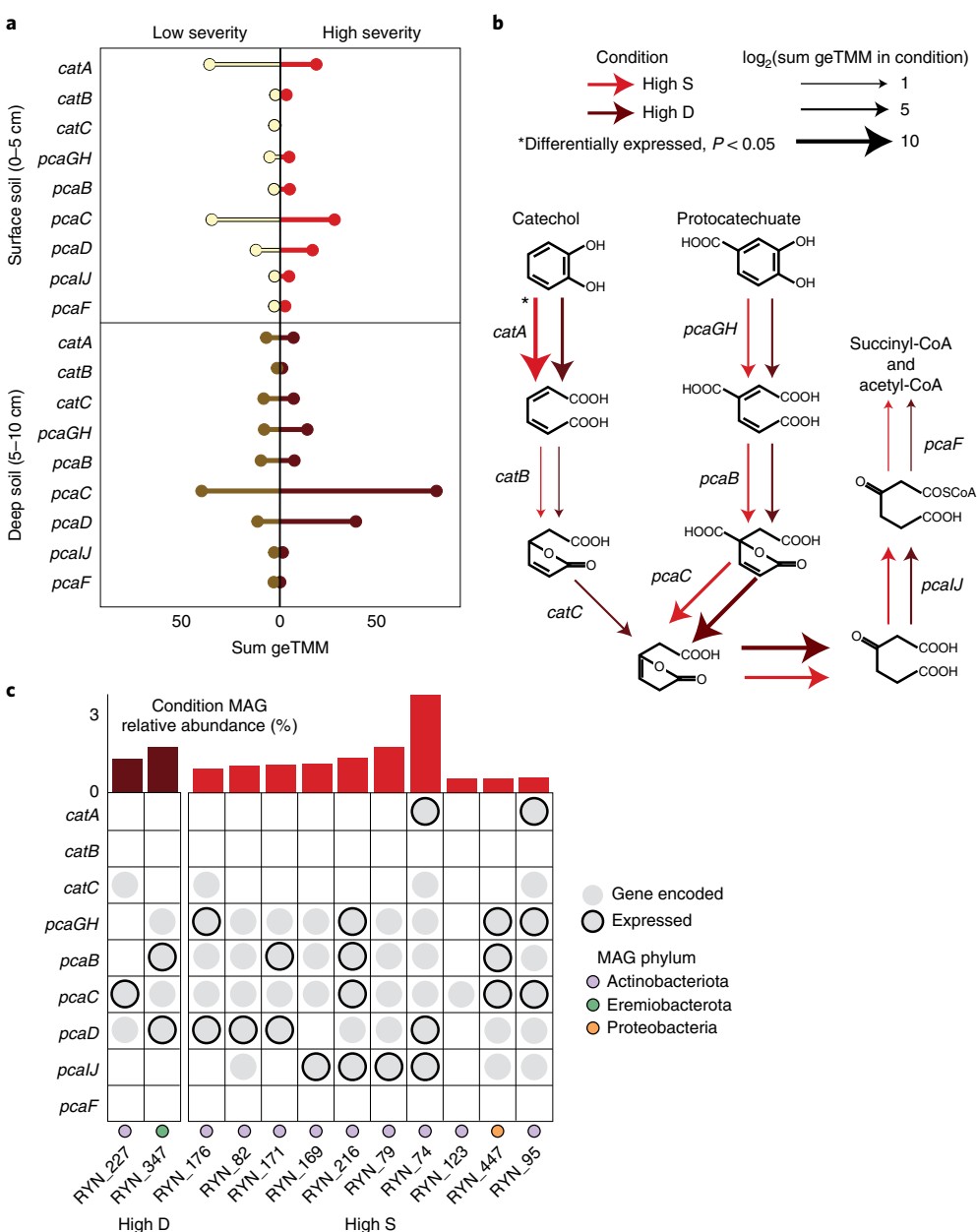

**Fig. 4 | Dominant MAGs express genes for utilizing aromatic carbon. a**, The summed geTMM of each gene for catechol and protocatechuate ortho-cleavage in each condition. **b**, The pathway for catechol and protocatechuate ortho-cleavage, with arrows indicating the log normalized sum geTMM of the gene for high severity surface and deep soils. Asterisk indicates genes that are differentially expressed in the condition (Wald's test in DESeq2; $P = 0.0055$ for *catA* in High S). **c**, The genomic potential and expression of each gene in the pathway for the MAGs of interest in High S and High D samples. The bar chart at the top shows the featured MAG relative abundance in that condition, coloured by featured condition.

genomes linked to all 28 of the featured High S Actinobacteria MAGs. Furthermore, metatranscriptomic analyses indicate that 13 of these AMGs were being actively transcribed, suggesting that prophage manipulate SOM degradation and potential cell growth in active bacteria in High S samples (Supplementary Data 5).

**Fungi are active across burn conditions.** Two fungal Ascomycota MAGs from known pyrophilous taxa, *Leotiomycetes* (R113–184) and *Coniochaeata ligniaria* (R110–5)[58–60], were reconstructed from metagenomes. These taxa were prominently represented in our internal transcribed spacer region (ITS) amplicons; the *Leotiomycetes* class increased in relative abundance by ~215% between control and High S samples (14% to 45%) and the *Coniochaeta* genus

relative abundance increased from 0.003% to 1% from control to High D samples.

Complementing observations from bacterial MAGs, the fungal MAGs encoded and expressed genes for degrading aromatic compounds. Both expressed genes for degrading salicylate (salicylate hydroxylase), phenol (phenol 2-monooxygenase) and catechol (catechol 1,2-dioxygenase), and expression of all three genes increased with fire severity. The MAGs also encoded laccases, which are enriched in pyrophilous fungal genomes[61] and act on aromatic substrates[62]. The *Coniochaeta* MAG additionally encoded hydrophobic surface binding proteins (*hsbA*; PF12296), which may facilitate the degradation of fire-derived hydrophobic compounds and be critical to soil recovery[61]. To compare the fungal and bacterial contribution

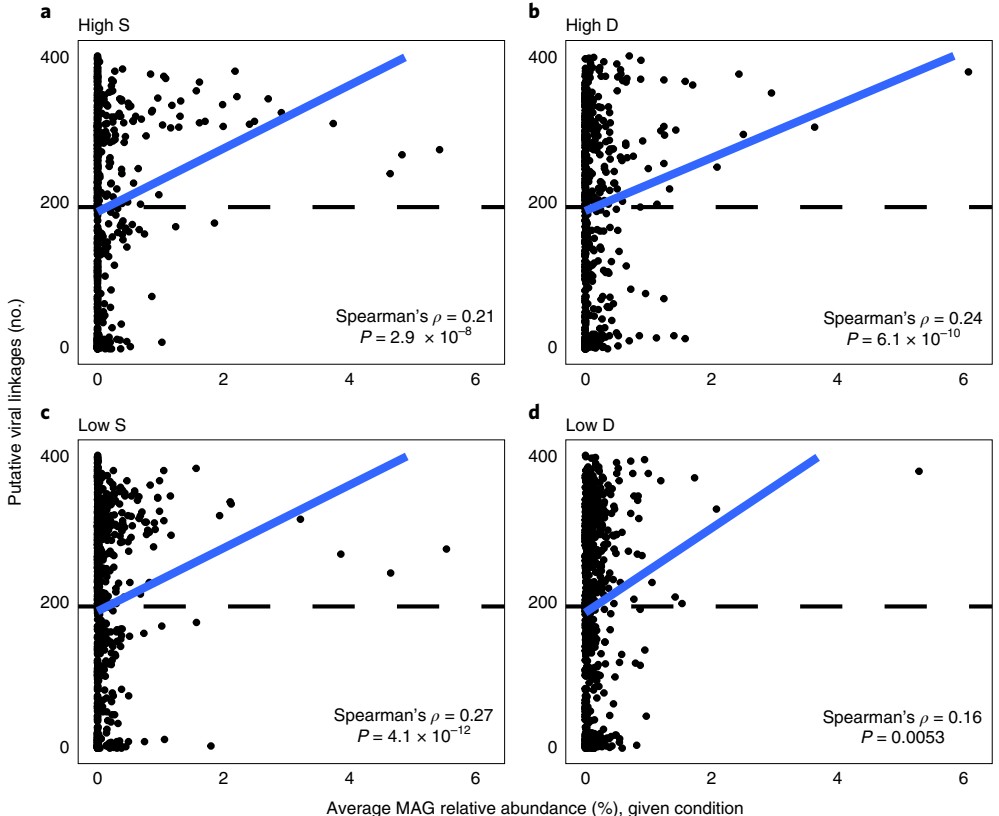

**Fig. 5 | Dominant MAGs are increasingly targeted by viruses in post-fire soils.** Each MAG's relative abundance within (**a**) High S and (**b**) D and (**c**) Low S and (**d**) D conditions plotted against the number of putative viral linkages identified by VirHostMatcher. Dashed line indicates the dataset average of 196 virus-host linkages. Correlation and significance between MAG relative abundance and number of putative viral linkages were assessed using the two-sided Spearman rho test.

to catechol degradation, we compared normalized transcriptomic reads recruited to the gene encoding catechol 1,2-dioxygenase, *catA*. In High S samples, the fungal MAGs generated more than twice the number of transcripts per gene compared with bacterial MAGs, indicating the important role that fungi probably play in aromatic DOM degradation in burned soils. Both fungal MAGs also expressed diverse peptidases (Supplementary Fig. 6), with increased expression from low to high fire severity in both surface and deep samples (~40.4% and 235%, respectively), which could degrade necromass from lysed microorganisms.

**Ecosystem implications of soil microbiome changes.** We observed short-term (1 yr post fire) differences in microbiome composition and function that probably alter biogeochemical cycling and initial post-fire vegetation recovery. We found no expression of the gene catalysing N fixation (*nifH*), despite the key role that N-fixing bacteria play in augmenting plant-available soil N pools[10] following disturbance, the pre- and post-fire abundance of actinorhizal shrubs (*Ceanothus velutinus, Shepherdia canadensis*) and the numerous leguminous forb species that form symbioses with N-fixing bacteria in these ecosystems. Nitrification is another key microbially mediated process that generally increases in post-fire soils due to an influx of ash-derived ammonium (also found here; Supplementary Data 1)[25,63]. Ammonia monooxygenase (*amoA*) transcripts were detected in deep soils but were absent in burned surface soils. Moreover, transcript abundances for both *amoA* and nitrite oxido-reductase (*nxrAB*) were significantly higher in Low D vs High D samples (Welch's *t*-test; $P < 0.05$) (Supplementary Data 4), potentially due to the inability of ammonia-oxidizing bacteria (for example, *Nitrospira*) to withstand post-fire soil conditions. Indeed, *Nitrospira*

was present in both control and burned deep soils but absent in moderate and high severity-impacted surface soils. These observations are supported by other studies; short-term, post-fire decreases in the abundances of genes catalysing N fixation and ammonia-oxidization have been noted in a conifer forest following a wildfire[64].

While pyrophilous taxa were enriched post wildfire, the loss of other soil microorganisms may impact biogeochemical processes and associated soil health. Increasing wildfire severity (from low to high) resulted in large decreases in the relative abundances of both Acidobacteria and Verrucomicrobia in surface soils (Extended Data Fig. 2; relative abundance decreases of 37.6% and 63.6%, respectively). Members of the Acidobacteria frequently play an active role in soil C cycling via decomposition[65–67] and are considered a keystone taxa for SOM degradation[68]. Here, representative MAGs affiliated with Acidobacteria (RYN_25, RYN_26) from the Pyrinomonadaceae family (16S rRNA gene data; −94.9% from Low S to High S) and Verrucomicrobia from the Verrucomicrobiaceae family (16S rRNA gene data; −82.35% Low S to High S) all encoded CAZYmes for targeting complex plant-derived carbohydrate polymers (for example, cellulose, beta-mannans, beta-galactans, xylan). Furthermore, Acidobacteria MAGs RYN_25 and RYN_26 both encoded genes (for example, *epsH*) for the synthesis of exopolysaccharides that play critical roles in soil aggregate formation and SOM stabilization as mineral-associated OM[69]. The loss of these taxa following severe wildfire may reduce the potential for SOM degradation and stabilization in surface soils.

Ectomycorrhizal fungi (EMF) facilitate plant access to limiting nutrients and water in return for photosynthetically derived carbohydrates[70]. We observed a 99% decrease in EMF relative abundances across the burn severity gradient (Supplementary Table 4),

which could be due to heat-induced fungal mortality or plant host death[14]. This has implications for the re-establishment of obligate ectomycorrhizal host plants such as *Pinus contorta*, the dominant tree species in these forests. For example, *Cenoccum geophilum*, a known EMF symbiont of *P. contorta*[71] that is indicative of fast conifer growth[72], was present in unburned sites but absent after fire. Inoculation of *P. contorta* and most conifers with EMF is a standard forest nursery production and reforestation practice[73], but inoculating seedlings destined for post-fire landscapes[74] with a mixture of local EMF species[71] may increase lost soil microbial diversity.

## Discussion

Here we present a genome-resolved multi-omics analysis of the impact of wildfire on the soil microbiome of conifer forest ecosystems, providing functional context to previously observed post-fire shifts in soil microbiome structure. Our results suggest that a combination of life strategies, including heat tolerance, fast growth and the utilization of pyrogenic substrates allow microorganisms to occupy available post-fire niche space. We found the widespread microbial processing of aromatic compounds that were probably generated during wildfire, which has implications for the residence time of pyrogenic C. Carbon processing in burned soils is also influenced by active viruses that target key bacterial community members through viral-mediated cell lysis and activity of AMGs. This rich genome-resolved multi-omic dataset provides invaluable insight into the impact of severe wildfire on the soil microbiome of western US forest ecosystems, which continue to experience unprecedented wildfire disturbances.

## Methods

**Field campaign.** Sampling was conducted in old-growth, lodgepole pine-dominated (*P. contorta*) forests burned by the Badger Creek (8,215 ha) and Ryan (11,567 ha) fires during 2018 in the Medicine Bow National Forest. The average return interval for wildfire within these forests is about 200 yr[75] and the even-aged lodgepole pine stands sampled regenerated from stand-replacing wildfires. Total annual precipitation averages 467 mm and mean annual temperature is 1.9 °C, with average annual minima and maxima of −12.1 °C and 17.1 °C, respectively (Cinnabar Park, SNOTEL site 1046). Soils are formed in metamorphic and igneous parent material and are well-drained, with moderate to rapid permeability. The most abundant soil types are loamy-skeletal Ustic Haplocryepts and fine-loamy Ustic Haplocryalfs (Supplementary Data 1). The plots were at similar elevation (2,480–2,760 m) and on mainly gentle slopes (10/15 plots; little aspect influence). Microbial communities were not statistically different between gentle and moderate sloping plots (ANOSIM; *P* < 0.05) and north or south facing plots when slope was moderate (>10°; ANOSIM; *P* < 0.05). Four burn severity gradients comprising low, moderate and high severity sites and an unburned control were selected on the basis of remotely sensed comparisons of pre and post-fire greenness[76], and then field validated before sampling (early August 2019) using US Forest Service guidelines[77,78]. Low, moderate and high severity sites had >85%, 20–85% and <20% surficial organic matter cover, respectively[77], which we quantified visually within each plot using a point-intercept approach (Extended Data Fig. 1). Low-severity plots had sparse grass and low shrub (*Vaccinium myrtillu*) cover, which we avoided to ensure we sampled root-free soil. Low-severity sites also had a very small litter layer to a depth of <1 mm and Site #4 had live trees remaining but all were >2 m away from the sampling plot. Samples were collected on 16 and 19 August 2019, 2 d without any precipitation events, approximately 1 yr following containment of both fires. At each sampling site, a 3 m × 5 m sampling grid with 6 m² subplots was laid out perpendicular to the dominant slope (Extended Data Fig. 1). Surface (0–5 cm depth) and deeper soil (5–10 cm depth) was collected with a sterilized trowel in each subplot for DNA and RNA extractions and subsequent microbial analyses. Surface soil samples included thin O-horizon at control and low-severity plots and charred mineral soil at moderate and high-severity plots. Deeper (5–10 cm) samples were mineral soils. In three subplots of each plot, additional material was collected for chemical analyses. Samples for RNA analyses were immediately flash-frozen using an ethanol-dry ice bath and placed on dry ice to remain frozen in the field. Samples for DNA extractions and chemical analyses were immediately placed on ice and all samples were transported to the laboratory at Colorado State University (CSU). Soils for DNA and RNA extractions were stored at −80 °C in the laboratory until processing. A total of 176 soil samples were collected (Supplementary Data 1).

**Soil chemistry.** We evaluated soil nutrients and chemistry to gauge changes across a gradient of wildfire severity and to consider the implications of those conditions

on microbial activity or substrate quality. Analyses of inorganic forms of soil N ($NO_3$-N and $NH_4$-N) were conducted on a subset of deep soil samples ($n = 12$ each for low, moderate and high severity, $n = 8$ for control). Samples were passed through a 4 mm mesh sieve and extracted with 2 M KCl within 24 h of sampling. Extracts were analysed for $NO_3$-N and $NH_4$-N by colorimetric spectrophotometry[79] (Lachat). A subset of surface ($n = 15$) and deep soil samples ($n = 45$) were dried (48 h at 60 °C), ground to a fine powder and analysed for total C and N by dry combustion (LECO). We analysed the $NO_3$-N and $NH_4$-N and dissolved organic C (DOC) and total dissolved N (TDN) released during warm water extraction[80] using ion chromatography ($NH_4$-N and $NO_3$-N; Thermo Fisher) and a Shimadzu TOC-VCPN analyser (DOC and TDN; Shimadzu). Soil pH was analysed in a 1:1 soil to deionized water slurry after 1 h of agitation[81] using a temperature-corrected glass electrode (Hach). Soil chemistry data are included in Supplementary Data 1 and discussed in Supplementary Note 1.

**High-resolution carbon analyses by FTICR-MS.** Water extractions were completed on a subset of 47 samples from the Ryan Fire for high-resolution C analyses using Fourier transform ion cyclotron resonance mass spectrometry (FTICR-MS) to analyse DOM. Briefly, 100 ml of milliQ water (>18 mΩ) was added to 50 g of sample in an acid-washed and combusted (400 °C for 6 h) 250 ml Erlenmeyer flask. These were placed on a shaker table for 10 h at 170 r.p.m. Following shaking, liquid was poured off into a 50 ml centrifuge tube and centrifuged for 10 min at 7,500 g and supernatant was filtered through a polypropylene 0.2 μm filter (polypropylene material). The extracts were acidified to pH 2 and additionally pre-treated with solid-phase extractions using Agilent Bond Elut-PPL cartridges (3 ml, 200 mg) (Agilent Technologies) following standard lab protocol[82] and subsequently diluted to 50 ppm. A 12 Tesla (12 T) Bruker SolariX FTICR-MS located at the Environmental Molecular Sciences Laboratory in Richland, Washington, USA was used to collect DOM high-resolution mass spectra from each DOM sample. Samples were injected into the instrument using a custom automated direction infusion cart that performed two offline blanks between each sample and using an Apollo II electrospray ionization source in negative ion mode with an applied voltage of −4.2 kV. Ion accumulation time was optimized between 50 and 80 ms. Transients (144) were co-added into a 4MWord time domain (transient length of 1.1 s) with a spectral mass window of 100–900 *m/z*, yielding a resolution at 400 K at 381 *m/z*. Spectra were internally recalibrated in the mass domain using homologous series separated by 14 Da ($CH_2$ groups). The mass measurement accuracy was typically within 1 ppm for singly charged ions across a broad *m/z* range (100 *m/z*−900 *m/z*). Bruker Daltonics DataAnalysis (version 4.2) was used to convert mass spectra to a list of *m/z* values by applying the FTMS peak picking module with a signal-to-noise ratio threshold set to 7 and absolute intensity threshold to the default value of 100. Chemical formulae were assigned with Formularity[83] on the basis of mass measurement error <0.5 ppm, taking into consideration the presence of C, H, O, N, S and P and excluding other elements. This open-access software was also used to align peaks with a 0.5 ppm threshold. Raw FTICR-MS data are provided in archive (doi:10.5281/zenodo.5182305). The R package ftmsRanalysis[84] was then used to remove peaks that either were outside the desired *m/z* range (200 *m/z*−900 *m/z*) or had a more abundant isotopologue, assign Van Krevelen compound classes and calculate nominal oxidation state of carbon (NOSC) and aromaticity index (AI) on the basis of the number of different atoms using equations (1) and (2) below:

$$NOSC = 4 - \frac{5C + H - 3N - 20 - 2S}{C} \tag{1}$$

$$AI = 4 - \frac{1 + C - O - S - 0.5H}{C - O - S - N - P} \tag{2}$$

Kendrick mass defect (KMD) analysis and plots were employed to identify potentially increasing polyaromaticity across the burn severity gradient. The KMD analysis was done using the $C_4H_2$ base unit (50 atomic mass units, amu) to represent the addition of benzene to a separate molecular benzene. The mass of each identified ion (*M*) was converted to its Kendrick mass (KM):

$$KM = M \left( \frac{50 \text{ amu}}{50.0587 \text{ amu}} \right) \tag{3}$$

with 50 amu being the nominal mass of $C_4H_2$ and 50.0587 being the exact mass of $C_4H_2$. The final KMD was obtained by subtracting the KM from the nominal KM, which is the initial ion mass rounded to the nearest integer. Series were identified as 2 or more formulae with the same KMD and a nominal Kendrick mass (NKM) differing by the $C_4H_2$ base unit (50 g mol$^{-1}$). Series were retained if they were present across all four burn severity conditions (control, low, moderate and high), resulting in 64 total series in the final analysis (Supplementary Note 4).

**DNA extraction, 16S rRNA gene and ITS amplicon sequencing.** DNA was extracted from soil samples using the Zymobiomics Quick-DNA faecal/soil microbe kits (Zymo Research). 16S rRNA genes in extracted DNA were amplified and sequenced at Argonne National Laboratory on the Illumina MiSeq using

251 bp paired-end reads and the primers 515F/806R[85], targeting the V4 region of the 16S rRNA gene. For fungal community composition, the DNA was PCR amplified targeting the first nuclear ribosomal ITS using the primers (ITS1f/ITS2) and sequenced on the Illumina MiSeq platform at the University of Colorado using 251 bp paired-end reads.

For taxonomic assignment, we used the SILVA[86] (release 132) and UNITE[87] (v8.3) databases for bacteria and fungi, respectively. We employed the QIIME2 environment[88] (release 2018.11) for processing of reads, which are both deposited and are available at NCBI under BioProject PRJNA682830. DADA2[89] was used to filter, learn error rates, denoise and remove chimeras from reads. Following this step, 16S rRNA gene and ITS amplicon sequencing reads retained on average 48,379 and 34,004 reads per sample, respectively. Taxonomy was assigned using the QIIME2 scikit-learn classifier trained on the SILVA and UNITE databases for bacteria and fungi, respectively. Ecological guilds were assigned to fungal amplicon sequence variants (ASVs) using FUNGuild[90] (v1.2). Similar to FUNGuild creator recommendations, we accepted guild assignments classified as 'highly probable' or 'probable' to avoid possible overinterpretation and discarded any ASVs classified as multiple guilds.

To characterize how microbial populations differed across burn severities and depths, we used the R[91] vegan[92] (v2.5-7) and phyloseq[93] (v1.28.0) packages. Non-metric multidimensional scaling (NMDS) was conducted on Bray-Curtis dissimilarities to examine broad differences between microbial communities. ANOSIM (vegan) was utilized to test the magnitude of dissimilarity between microbial communities. Mean species diversity of each sample (alpha diversity) was calculated on the basis of species abundance, evenness or phylogenetic relationships using Shannon's diversity index, Faith's phylogenetic diversity and Pielou's evenness. Linear discriminant analysis with a score threshold of 2.0 was used to determine ASVs discriminant for unburned or burned soil[94].

**Metagenomic assembly and binning.** A subset of 12 Ryan Fire samples from a single transect representing low- and high-severity burn from surface and deep soils was selected for metagenomic sequencing to analyse changes in microbial community functional potential ($n = 3$ per condition). The four different conditions are hereafter referred to as 'Low S' (low-severity surface soil), 'High S' (high-severity surface soil), 'Low D' (low-severity deep soil) and 'High D' (high-severity deep soil). Libraries were prepared using the Tecan Ovation Ultralow System V2 and were sequenced on the NovaSEQ6000 platform on an S4 flow cell using 151 bp paired-end reads at Genomics Shared Resource, Colorado Cancer Center, Denver, Colorado, USA. Sequencing adapter sequences were removed from raw reads using BBduk (https://jgi.doe.gov/data-and-tools/bbtools/bb-tools-user-guide/bbduk-guide/) and reads were trimmed with Sickle[95] (v1.33). For each sample, trimmed reads were assembled into contiguous sequences (contigs) using the de novo de Bruijn assembler MEGAHIT v1.2.9 using kmers[96] (minimum kmer of 27, maximum kmer of 127 with step of 10). Assembled contigs shorter than 2,500 bp were discarded for all downstream usages, including gene-resolved analyses for inorganic N cycling and binning into genomes. These assembled contigs (>2,500 bp) were binned using MetaBAT2 with default parameters[97] (v2.12). Metagenome-assembled genome (MAG) quality was estimated using checkM[98] (v1.1.2) and taxonomy was assigned using GTDB-Tk[99] (R05-RS95, v1.3.0). MAGs from all metagenomes were dereplicated using dRep[100] (default parameters, v2.2.3) to create a non-redundant MAG dataset. Low quality MAGs (<50% completion and >10% contamination) were excluded from further analysis[101]. Reads from all samples were mapped to the dereplicated MAGs using BBMap with default parameters (version 38.70, https://sourceforge.net/projects/bbmap/). Per-contig coverage across each sample was calculated using CoverM contig (v0.3.2) (https://github.com/wwood/CoverM) with the 'Trimmed Mean' method, retaining only those mappings with minimum percent identity of 95% and minimum alignment length of 75%. Coverages were scaled on the basis of library size and scaled per-contig coverages were used to calculate the mean per-bin coverage and relative abundance in each sample (Supplementary Data 2). The quality metrics and taxonomy of the subsequent 637 medium- and high-quality MAGs discussed here are included in the Supplementary Information (Supplementary Data 2) and are deposited at NCBI (BioProject ID PRJNA682830). Maximum cell doubling times were calculated from codon usage bias patterns in each MAG with >10 ribosomal proteins using gRodon[42] (Supplementary Data 2). Bacterial MAGs with an average relative abundance >0.5% across triplicates (with a standard deviation less than the average relative abundance) in both High S and High D were selected as MAGs of interest for further genome-resolved discussion and insight into the function of the post-fire microbiome in surface and deep soils (Supplementary Data 2).

Fungal MAGs (R113–184 and R110–5) were identified because they were abnormally large for bacterial MAGs and were confirmed as of eukaryotic origin on the basis of mmseqs2 searches for all available open reading frames in their contigs against the NCBI NR and MycoCosm databases, with best hits to *Coniochaeta ligniaria* and the *Leotiomycetes*/*Helotiales* clade. To identify taxonomy and precisely place the MAGs in the fungal tree (Supplementary Fig. 1), we used 867 single-copy orthogroups from OrthoFinder v2.5.4[102] using default parameters. The protein sequences in each orthogroup were aligned with

MAFFT[103] (–maxiterate 1000–globalpair) and trimmed with TrimA1 v1.4.rev22[104] (-automated1). All the filtered MSAs were concatenated. The phylogenetic tree was built using iqtree v1.6.9[105] detecting the best model for each gene partition, 10,000 ultrafast bootstrap and 10,000 SH-like approximate likelihood ratio test (-m MFP -bb 10000 -alrt 10000 -safe). The tree was visualized using FigTree 1.4.4 (http://tree.bio.ed.ac.uk/software/figtree/) and the support values represent the ultrafast bootstraps/SH-aLRT. Completeness for both MAGs was assessed using BUSCO v4.0.6[106] and CEGMA[107].

**MAG annotation.** Eukaryotic MAGs were annotated using the JGI annotation pipeline, analysed with complementary metatranscriptomics assemblies[108] (RnaSPAdes, v3.13.0) and are deposited on MycoCosm[109] (https://mycocosm.jgi.doe.gov/ColoR110_1 and https://mycocosm.jgi.doe.gov/ColoR113_1). Bacterial MAGs were annotated using DRAM[110] (v1.0). In addition to the DRAM annotations, we used HMMER[111] against Kofamscan HMMs[112] to identify genes for catechol and protocatechuate meta- and ortho-cleavage, naphthalene transformations and inorganic N cycling (Supplementary Data 3).

**Metatranscriptomics.** RNA was extracted from the subset of 12 samples utilized for metagenomics using the Zymobiomics DNA/RNA mini kit (Zymo Research) and RNA was cleaned, DNase treated and concentrated using the Zymobiomics RNA Clean & Concentrator kit (Zymo Research). The Takara SMARTer Stranded Total RNA-Seq kit v2 (Takara Bio) was used to remove ribosomal RNA from total RNA and construct sequencing libraries. Samples were sequenced on the NovaSEQ6000 platform on an S4 flow cell using 151 bp paired-end reads at Genomics Shared Resource, Colorado Cancer Center, Denver, Colorado, USA. Adapter sequences were removed from raw reads using Bbduk (https://jgi.doe.gov/data-and-tools/bbtools/bb-tools-user-guide/bbduk-guide/) and sequences were trimmed with Sickle v1.33[95]. Trimmed reads were mapped to metagenome assemblies using BBMap (parameters: ambiguous, random; idfilter, 0.95; v38.70). Mappings were filtered to 95% identity and counts were generated using HTSeq[113]. For differential expression analysis, the dataset was filtered to transcripts which were successfully annotated by DRAM ($n = 132,665$) and DESeq2[114] was used to identify transcripts that were differentially expressed in any condition (Supplementary Data 4). The same analysis was also run on the combined HMM output described above (1,189 total transcripts). We normalized our dataset by calculating the gene length-corrected trimmed mean of $M$ values[115] (geTMM) using edgeR[116] to normalize for library depth and gene length (Supplementary Data 4). To identify transcripts that were highly expressed in any given condition, we filtered the data to transcripts that were in the upper 20% of TMM for 2 of the 3 samples in any one condition (Supplementary Table 2). To compare bacterial and fungal expression data for individual genes, we normalized the number of either fungal or bacterial transcript reads to the gene coverage in each sample to compare the number of transcripts recruited per gene.

**Viruses.** Viral contigs were recovered from the metagenomic assemblies using VirSorter2[117] (v2.2.2) and only contigs ≥10 kb with a VirSorter2 score >0.5 were retained. Viral contigs were trimmed using checkV[118] (v0.4.0) and the final contigs were clustered using the CyVerse app ClusterGenomes (v1.1.3) requiring an average nucleotide identity of 95% or greater over at least 80% of the shortest contig. The final DNA viral metagenome-assembled genome (vMAG) dataset was manually curated using the checkV, VIRSorter2 and DRAM-v annotation outputs according to protocol[119]. RNA vMAGs were also recovered from metatranscriptome assemblies using VIRSorter2[117] (v2.2.2). The resulting sequences were clustered using ClusterGenomes (v1.1.3) on CyVerse using the aforementioned parameters. To quantify relative abundance of DNA and RNA vMAGs across the 12 samples, we mapped the metagenomic and metatranscriptomic reads to the vMAGs using BBMap with default parameters (v38.70). To determine vMAGs that had reads mapped to at least 75% of the vMAG, we used CoverM (v0.6.0) in contig mode to find vMAGs that passed this 75% threshold (–min-covered-fraction 75). We then used CoverM (v0.6.0) in contig mode to output reads per base and used this to calculate final DNA and RNA vMAG relative abundance in each metagenome and metatranscriptome. vConTACT2 (v0.9.8; CyVerse) was used to determine vMAG taxonomy. Final viral sequences are deposited on NCBI (BioProject ID PRJNA682830 - BioSamples SAMN20555178, SAMN20555179; Supplementary Data 5). We used DRAM-v[110] (v1.2.0) to identify AMGs within the final viral dataset (Supplementary Data 5).

CRISPR-Cas protospacers were found and extracted from MAG sequences using the CRISPR Recognition Tool[120] (minimum of 3 spacers and 4 repeats) in Geneious (v2020.0.3) and CRisprASSembler[121] with default parameters (v1.0.1). BLASTn was used to compare MAG protospacer sequences with protospacer sequences in vMAGs, with matches only retained if they were 100% or contained ≤1 bp mismatch with an $e$-value $\leq 1 \times 10^{-5}$. To identify putative vMAG-MAG linkages, we used an oligonucleotide frequency dissimilarity measure (VirHostMatcher v1.0.0) and retained only linkages with a $d_2^*$ value <0.25[55] (Supplementary Data 5).

**Reporting summary.** Further information on research design is available in the Nature Research Reporting Summary linked to this article.

## Data availability

The metagenomic reads, metatranscriptomic reads, bacterial and viral MAGs, 16S rRNA gene sequencing reads and ITS amplicon reads reported in this paper have been deposited in National Center for Biotechnology Information BioProject PRJNA682830. The two fungal MAGs and corresponding annotations are deposited in the Joint Genome Institute (JGI) MycoCosm portal and can be assessed at https://mycocosm.jgi.doe.gov/ColoR113_1 and https://mycocosm.jgi.doe.gov/ColoR110_1. FTICR-MS data have been deposited in the Zenodo archive with identifier https://doi.org/10.5281/zenodo.5182305. The following databases were also used: Silva (release 132), UNITE (v8.3) and GTDB-Tk (v1.3.0). Processed data are included in the Supplementary Data files which are detailed in the Supplementary Information.

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

## Acknowledgements

This work was supported through a USDA NIFA award (2021-67019-34608) to M.J.W. FTICR-MS analyses were performed under the Facilities Integrating Collaborations for User Science (FICUS) initiative and used resources at the Environmental Molecular Sciences Laboratory (proposal ID 49615), which is a DOE Office of Science User Facility. This facility is sponsored by the Office of Biological and Environmental Research and operated under contract number DE-AC05-76RL01830. Metagenomic and metatranscriptomic sequencing was performed at the University of Colorado Cancer Center's Genomics Shared Resource, which is supported by the Cancer Center Support Grant P30CA046934. This work utilized resources from the University of Colorado Boulder Research Computing Group, which is supported by the National Science Foundation (awards ACI-1532235 and ACI-1532236), the University of Colorado Boulder and Colorado State University. The work conducted by the US Department of Energy Joint Genome Institute, a DOE Office of Science User Facility, is supported by the Office of Science of the US Department of Energy under Contract No. DE-AC02-05CH11231. We thank F. Pulido-Chavez for help with processing the ITS amplicon sequencing data, and J. Emerson and S. Geonczy for discussions on viral analyses.

## Author contributions

A.R.N., C.C.R. and M.J.W. designed the field sampling and downstream analyses. A.R.N., K.K.A. and M.J.W. performed field sampling, while A.R.N. and R.A.D. performed laboratory sample processing. A.R.N. and A.B.N. led the microbial analyses, with assistance from S.J.M., A.S.S., I.V.G. and A.S. for fungal genomics. H.K.R., T.B., R.K.C. and R.B.Y. contributed to high-resolution carbon measurements and analyses. T.S.F. performed bulk soil geochemistry measurements. A.R.N., C.C.R. and M.J.W. wrote the manuscript, with assistance and input from all co-authors.

## Competing interests

The authors declare no competing interests.

## Additional information

**Extended data** is available for this paper at https://doi.org/10.1038/s41564-022-01203-y.

**Correspondence and requests for materials** should be addressed to Michael J. Wilkins.

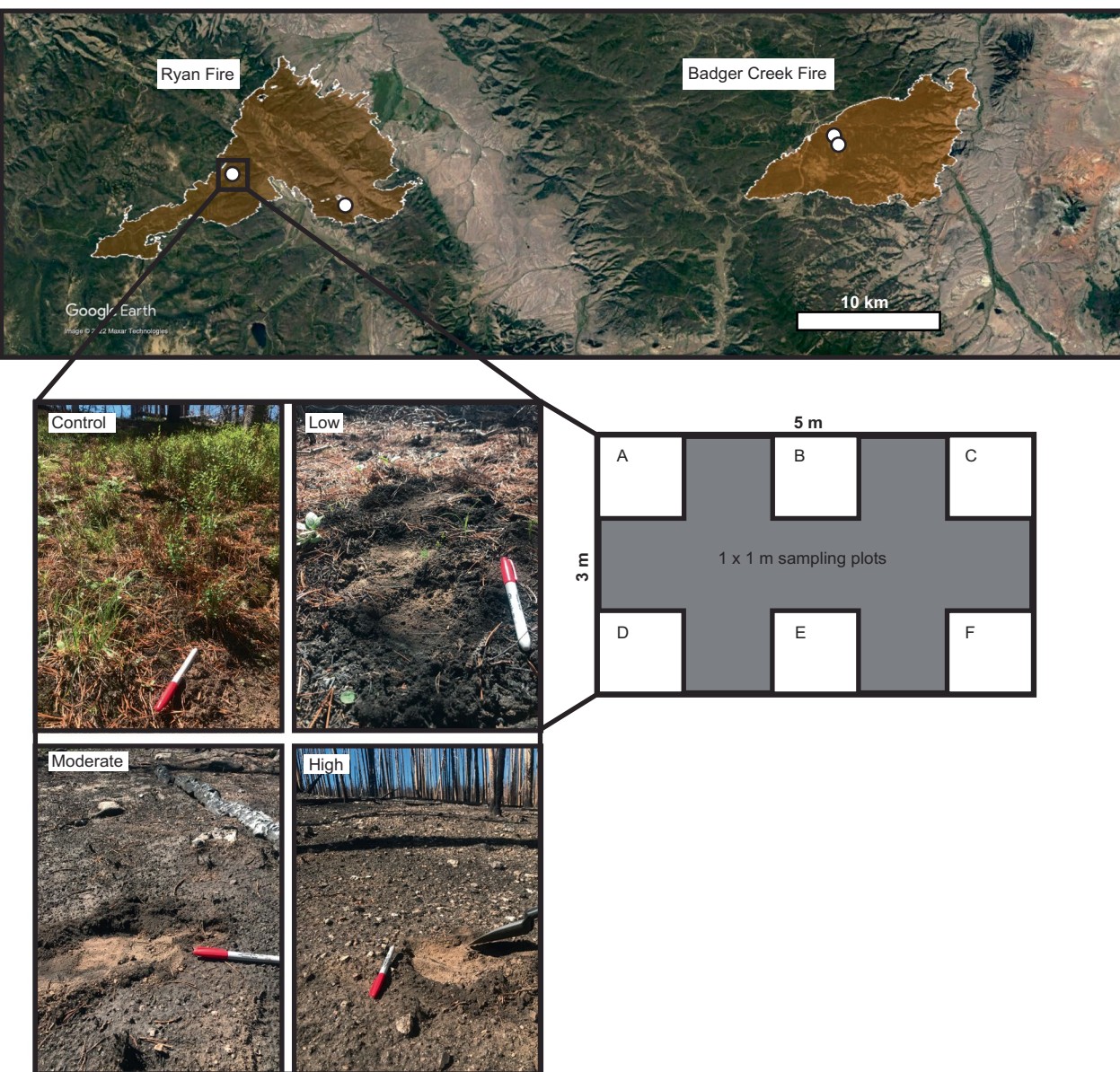

**Extended Data Fig. 1 | Overview of field sampling design.** There were four replicate burn severity gradients (two at Ryan Fire and two at Badger Creek Fire); six subsamples were collected in each burn condition at each gradient.

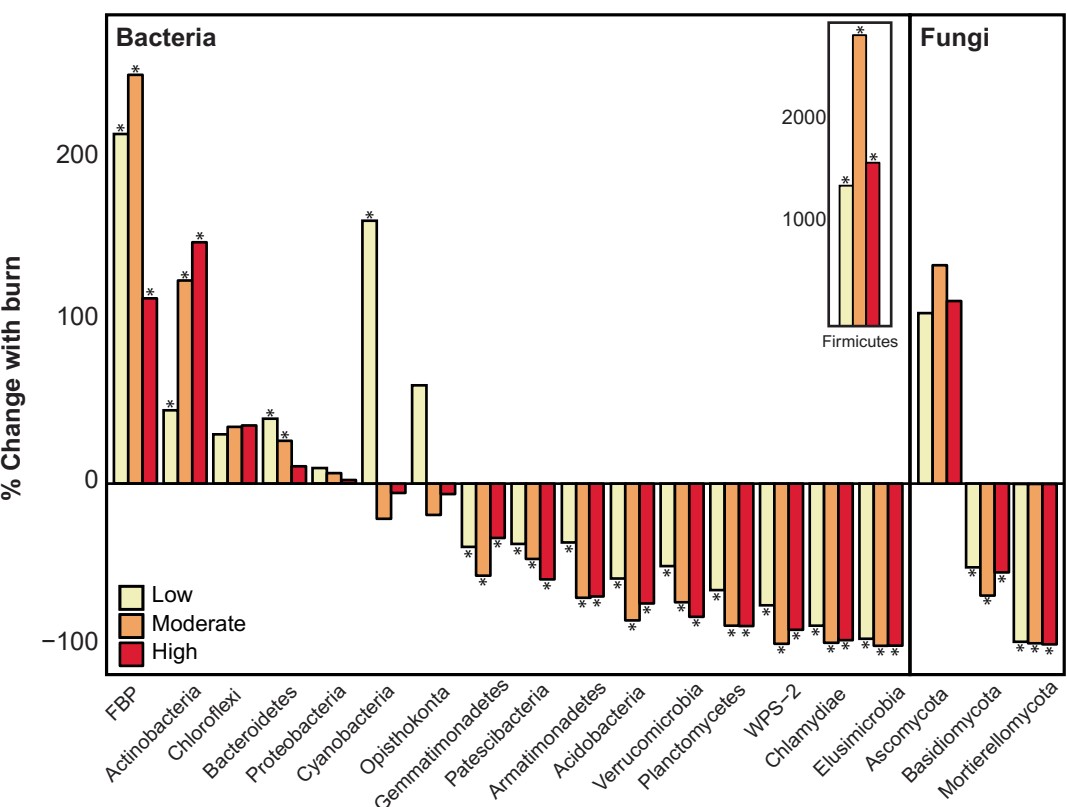

**Extended Data Fig. 2 | Shifting soil microbiome composition with wildfire burn severity.** The percent change in relative abundance from control to low, moderate, and high severity in surface soil of each main bacterial and fungal phylum. Phyla with relative abundance less than 0.5% were discarded for this analysis. Note that although the *Firmicutes* have the largest increase with burn (inset) their overall relative abundance in burned samples is still low relative to *Actinobacteria* (1.21% vs 25.6% relative abundance). Phyla with significant (one-sided pairwise t-test; p < 0.05) differences in relative abundance between the unburned and burned conditions are denoted with an asterisk.

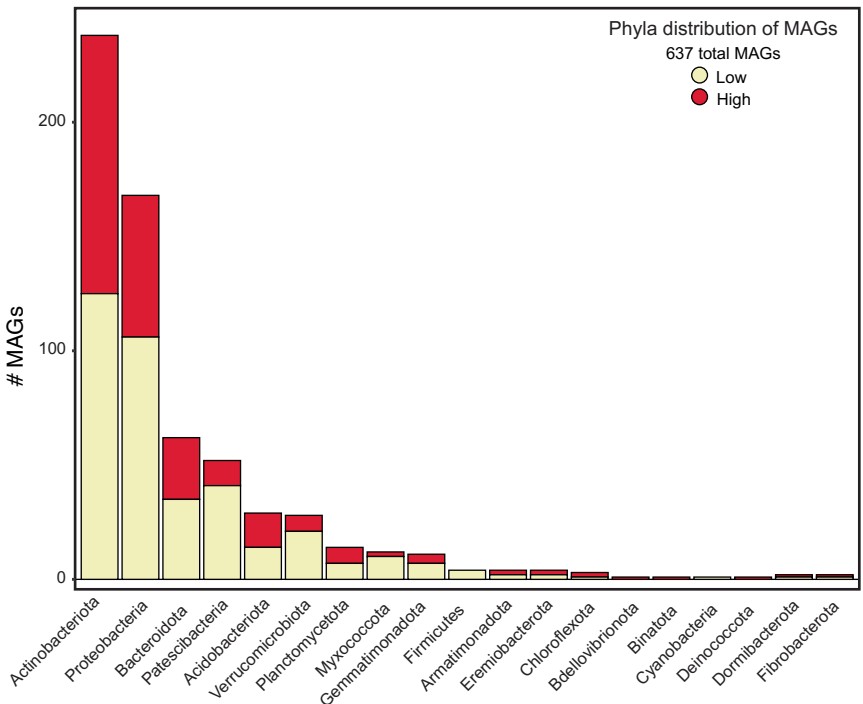

**Extended Data Fig. 3 | Phyla distribution of bacterial metagenome-assembled genomes (MAGs).** Phyla distribution of the 637 medium- and high-quality MAGs (> 50% completion, <10% contamination) from burned surface and deep soils.

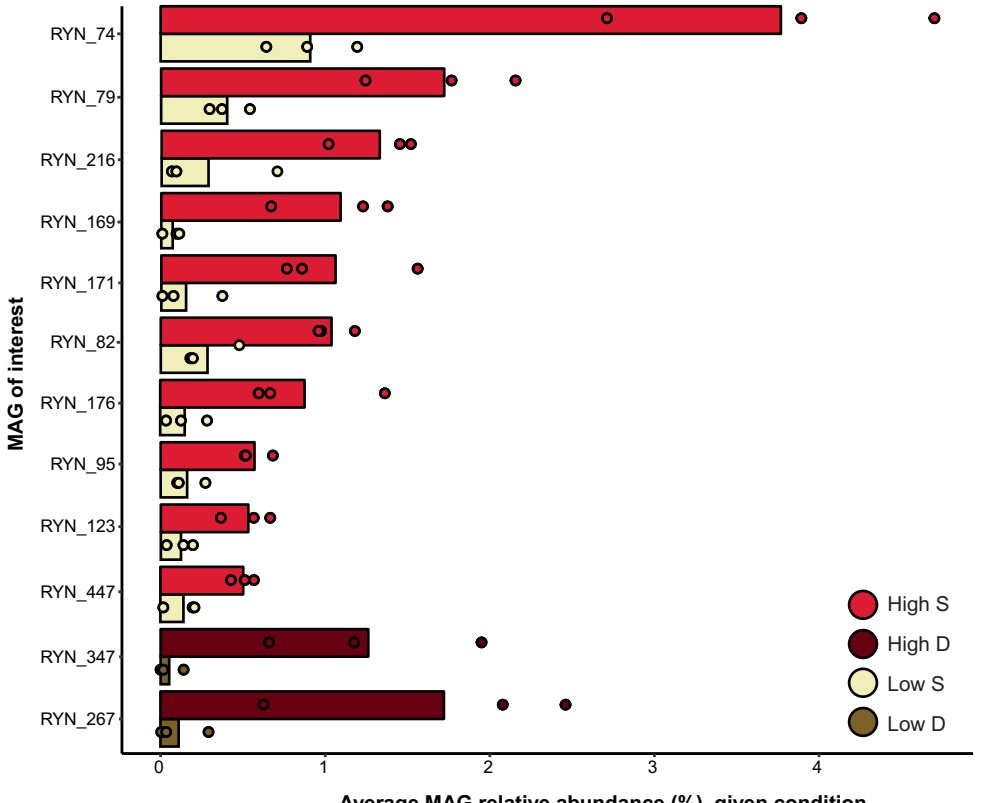

**Extended Data Fig. 4 | Average relative abundance of MAGs of interest across severities.** The relative abundances in Low and high severity of MAGs that are significantly enriched (one-sided pairwise t-test; $p < 0.05$; p-values in Supplementary Data 2) in High S vs. Low S (7 MAGs) or High D vs. Low D (2 MAGs) that match the designated criteria for discussion in the text (average relative abundance $> 0.05\%$ and standard deviation less than average relative abundance). Overlayed points show relative abundance of MAGs in each samples (n = 3 for each condition).

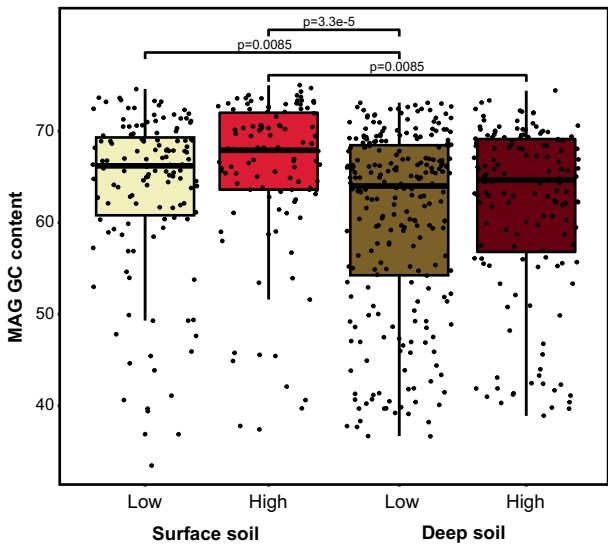

**Extended Data Fig. 5 | MAG GC content higher in MAGs reconstructed from high severity-impacted surface soils.** GC content of MAGs reconstructed from each condition. P-values are indicated between conditions if there are significant differences (one-sided pairwise t-test, p < 0.05). The lower and upper hinges of the boxplots represent the 25th and 75th percentile and the middle line is the median. The upper whisker extends to the median plus 1.5x interquartile range and the lower whisker extends to the median minus 1.5x interquartile range. Jittered points represent individual MAGs (n=131 reconstructed from Low S samples, n=111 from High S, n=247 from Low D, n=148 from High D).

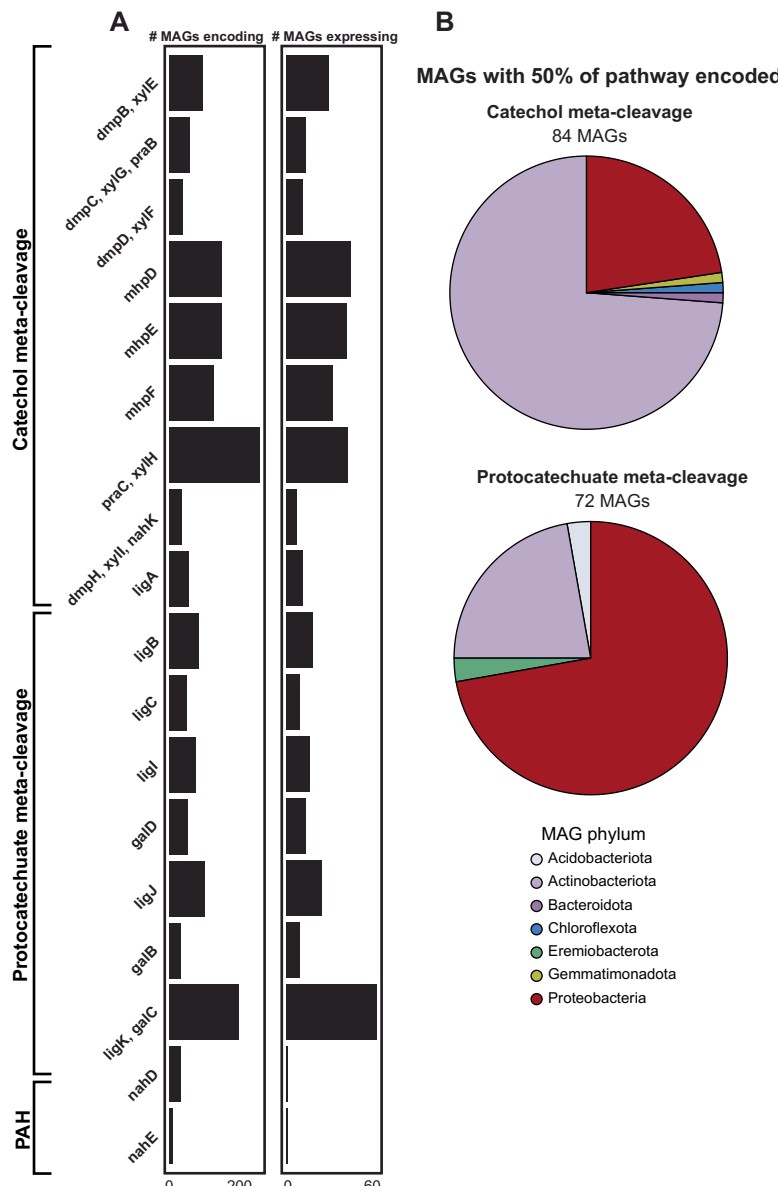

**Extended Data Fig. 6 | Widespread potential and expression of catechol and protocatechuate meta-cleavage in MAG dataset.** (**a**) Number of MAGs encoding and expressing each gene of the catechol and protocatechuate meta-cleavage pathways. (**b**) Phyla distribution of MAGs encoding 50% of either pathway.

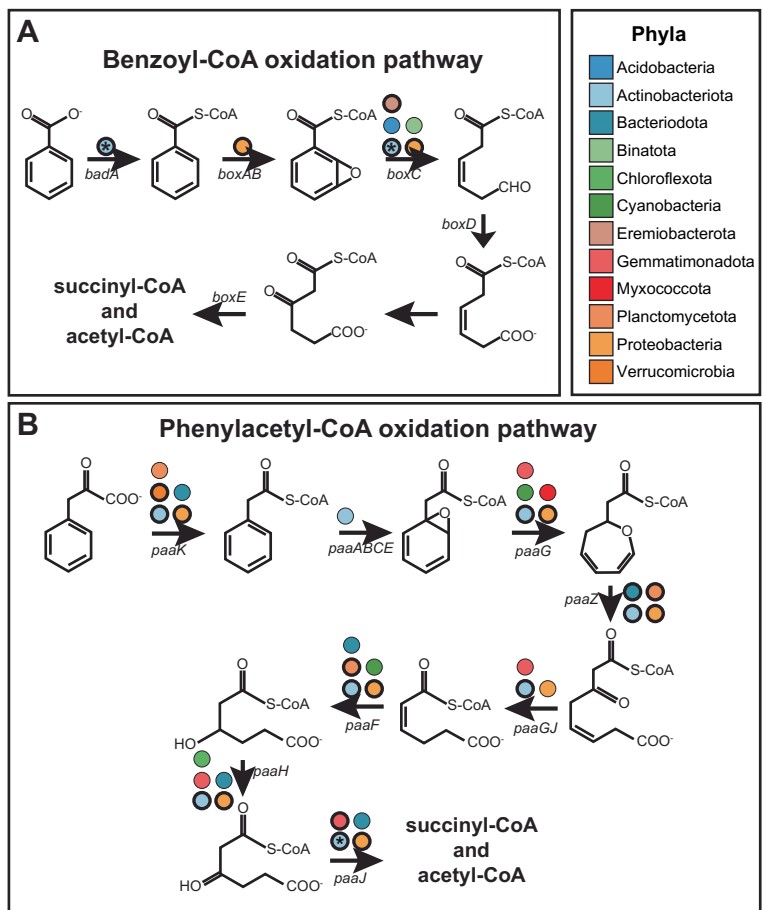

**Extended Data Fig. 7 | Genomic evidence of other aerobic aromatic C degradation pathways.** The benzoyl-CoA oxidation pathway (**a**) and phenylacetyl-CoA oxidation pathway (**b**) with overlaid gene names and whether there was encoded or expressed evidence of these genes. Colored circles indicate whether a MAG from that phyla encoded that gene, bolded circles indicate that metatranscriptomic reads mapped to the gene, and asterisks indicate the gene was being highly expressed in any given condition.

# Reporting Summary

## Statistics

For all statistical analyses, confirm that the following items are present in the figure legend, table legend, main text, or Methods section.

| n/a | Confirmed | |
|---|---|---|
| ☐ | ☒ | The exact sample size (*n*) for each experimental group/condition, given as a discrete number and unit of measurement |
| ☐ | ☒ | A statement on whether measurements were taken from distinct samples or whether the same sample was measured repeatedly |
| ☐ | ☒ | The statistical test(s) used AND whether they are one- or two-sided<br>*Only common tests should be described solely by name; describe more complex techniques in the Methods section.* |
| ☒ | ☐ | A description of all covariates tested |
| ☐ | ☒ | A description of any assumptions or corrections, such as tests of normality and adjustment for multiple comparisons |
| ☐ | ☒ | A full description of the statistical parameters including central tendency (e.g. means) or other basic estimates (e.g. regression coefficient) AND variation (e.g. standard deviation) or associated estimates of uncertainty (e.g. confidence intervals) |
| ☐ | ☒ | For null hypothesis testing, the test statistic (e.g. *F*, *t*, *r*) with confidence intervals, effect sizes, degrees of freedom and *P* value noted<br>*Give P values as exact values whenever suitable.* |
| ☒ | ☐ | For Bayesian analysis, information on the choice of priors and Markov chain Monte Carlo settings |
| ☒ | ☐ | For hierarchical and complex designs, identification of the appropriate level for tests and full reporting of outcomes |
| ☐ | ☒ | Estimates of effect sizes (e.g. Cohen's *d*, Pearson's *r*), indicating how they were calculated |

*Our web collection on statistics for biologists contains articles on many of the points above.*

## Software and code

Policy information about availability of computer code

| Data collection | No software was used for the collection of data (e.g., chemistry, raw sequencing reads). |
|---|---|
| Data analysis | Bioinformatic tools used for sequencing data processing and analysis: QIIME2 (2018.11), DADA2, FUNGuild (v1.2), BBduk, Sickle (v1.33), MEGAHIT (v1.2.9), MetaBAT2 (v2.12), checkM (v1.1.2), GTDB-Tk (V1.3.0), dRep (v2.2.3), CoverM (v0.6.0), gRodon, OrthoFinder (v2.5.4), MAFFT, TrimA1 (v1.4.rev22), iqtree (v1.6.9), FigTree (1.4.4), BUSCO (v4.0.6), CEGMA, RnaSPAdes (v3.13.0), DRAM (v1.0), HMMER, BBMap (v38.70), HTSeq, DESeq2, edgeR, VIRSorter2 (v2.2.2), CyVerse ClusterGenomes (v1.1.3), vConTACT2 (v0.9.8), Dram-v (v1.2.0), Geneious (v2020.0.3), CRisprASSembler (v1.0.1), VirHostMatcher (v1.0.0). The following tools were used for data processing and visualization: RStudio (v3.6.1), vegan (v2.5-7), phyloseq (v.1.28.0), Adobe Illustrator 2020 (v25.2). The following tools were used to process FTICR-MS data: ftmsRanalysis, Formularity, Bruker Daltonics DataAnalysis (version 4.2). |

For manuscripts utilizing custom algorithms or software that are central to the research but not yet described in published literature, software must be made available to editors and reviewers. We strongly encourage code deposition in a community repository (e.g. GitHub). See the Nature Portfolio guidelines for submitting code & software for further information.

## Data

Policy information about [availability of data](availability of data)

All manuscripts must include a [data availability statement](data availability statement). This statement should provide the following information, where applicable:
- Accession codes, unique identifiers, or web links for publicly available datasets
- A description of any restrictions on data availability
- For clinical datasets or third party data, please ensure that the statement adheres to our [policy](policy)

The data availability statement includes the NCBI Bioproject to assess the metagenomic and metatranscriptomic reads, bacterial and viral MAGs, 16S rRNA gene sequencing reads, and ITS amplicon reads (PRJNA682830). The two fungal MAGs are deposited in the JGI MycoCosm portal and can be assessed at https://mycocosm.jgi.doe.gov/ColoR113_1 and https://mycocosm.jgi.doe.gov/ColoR110_1. The raw FTICR-MS data is deposited on Zenodo with identifier doi:10.5281/zenodo.5182305. All data is publicly accessible at time of initial submission. The following databases were also used: Silva (release 132), UNITE (v8.3), GTDB-Tk (v1.3.0).

## Human research participants

Policy information about [studies involving human research participants and Sex and Gender in Research.](studies involving human research participants and Sex and Gender in Research.)

| | |
|---|---|
| Reporting on sex and gender | N/A to this study. |
| Population characteristics | N/A to this study. |
| Recruitment | N/A to this study. |
| Ethics oversight | N/A to this study. |

Note that full information on the approval of the study protocol must also be provided in the manuscript.

# Field-specific reporting

Please select the one below that is the best fit for your research. If you are not sure, read the appropriate sections before making your selection.

☐ Life sciences   ☐ Behavioural & social sciences   ☒ Ecological, evolutionary & environmental sciences

For a reference copy of the document with all sections, see [nature.com/documents/nr-reporting-summary-flat.pdf](nature.com/documents/nr-reporting-summary-flat.pdf)

# Ecological, evolutionary & environmental sciences study design

All studies must disclose on these points even when the disclosure is negative.

| | |
|---|---|
| Study description | To assess the influence of wildfire burn severity on the soil microbiome, we used coupled microbial approaches to comprehensively characterize the soil microbiome across a burn severity gradient (unburned control, low, moderate, and high severity). With marker gene approaches (16S rRNA gene sequencing, ITS amplicon sequencing), we showed that deeper mineral horizon soils were more insulated from the impacts of wildfire and that wildfire exerted a homogenizing influence on the organic horizon microbiome, with Actinobacteria dominating the post-fire soils. Metagenomic and metatranscriptomic sequencing coupled with FTICR-MS analyses within low and high severity-impacted soils showed the active degradation of fire-derived aromatic soil organic matter compounds in burned soils. This approach also allowed us to identify DNA and RNA viral genomes (vMAGs) within the post-fire system, with evidence of active viral predation of dominant and active MAGs. This publicly-available dataset offers a myriad of opportunities for future research into the impact of wildfire on the soil microbiome, which is becoming increasingly important to understand as wildfires increase in frequency, severity, and duration across the globe. |
| Research sample | Depth-resolved soil samples (organic and mineral horizon) from unburned, low, moderate, and high severity-impacted conditions were used for all data collection. A total of 176 samples were collected with a large amount of replication to fully capture the heterogeneity inherent in samples. For metagenomics and metatranscriptomics analyses, we used triplicate of each condition. |
| Sampling strategy | Four candidate burn severity gradients were selected based on US Forest Service, Burned Area Emergency Response program (BAER) remotely sensed imagery and maps, and subsequently field validated. Aspect, slope, and elevation were recorded at each sampling plot and was kept generally consistent across all plots. Each gradient comprised low, moderate, and high severity sites and an unburned control. Low, moderate, and high severity sites had >85%, 20-85%, and <20% surficial organic matter cover, respectively. Samples were collected on August 16 and 19 of 2019, approximately one year following containment of both fires. At each sampling site, a 3 m x 5 m sampling grid with six m2 subplots was laid out perpendicular to the dominant slope (Figure S1). Subsamples of the organic soil horizon (i.e., litter and duff; O-horizon) and upper mineral soil horizon (0-5 cm; A-horizon) were collected with a sterilized trowel in each subplot for DNA and RNA extractions and subsequent microbial analyses. In three subplots, additional material was collected for chemical analyses. Samples for RNA analyses were immediately flash-frozen using an ethanol-dry ice bath and subsequently placed on ice to remain frozen in the field. Samples for DNA extractions and chemical analyses were immediately |

| | |
|---|---|
| | placed on ice and all samples were transported to the laboratory at Colorado State University (CSU). Soils for DNA and RNA extractions were stored at -80°C in the laboratory until processing. |
| Data collection | ARN led the majority of data collection and completed all DNA and RNA extractions, and water extractions and SPE for FTICR-MS. 16S rRNA genes in extracted DNA were amplified and sequenced at Argonne National Laboratory on the Illumina MiSeq. ITS amplicon amplification and sequencing was completed at the University of Colorado BioFrontiers Institute Next-Gen Sequencing Core Facility on the Illumina MiSeq platform. Data was processed by ARN using QIIME and the Silva and Unite databases. A FTICR-MS located at the Environmental Molecular Sciences Laboratory in Richland, WA, was used to collect DOM high-resolution mass spectra of all DOM extracts and the Formularity software was used to assign formulas to peaks. The FTICR-MS data collection was completed by RKC. Metagenomics and metatranscriptomics sequencing on the 12 subsetted samples was completed at Genomics Shared Research at the Colorado Cancer Center. See methods for sequencing details. Bioinformatics tools were utilized by ARN to process the raw reads. Fungal MAGs were annotated by ASS, SJM, IVG, and AS. |
| Timing and spatial scale | Soils were collected at one time point one year following the containment of the Ryan and Badger Creek Wildfires. To characterize spatial heterogeneity, we collected samples across two ~200 m burn severity gradient transects within each fire. |
| Data exclusions | None of the aforementioned collected data was excluded from this study. |
| Reproducibility | Analyses were performed across biological replicate samples. Furthermore, biological trends (i.e., shifts in community composition) are inferred from several different analytical methods |
| Randomization | Sample allocation was not random and all samples were treated as equal as possible. Metagenome library construction was randomized, although samples were not blinded. FTICR MS analyses were randomized at EMSL |
| Blinding | DOM analyses via FTICR MS were performed using blinding. |

Did the study involve field work? ☐ Yes ☐ No

## Field work, collection and transport

| | |
|---|---|
| Field conditions | Sampling was conducted in lodgepole pine (Pinus contorta) forests burned by Badger Creek (8215 ha) and Ryan (11567 ha) fires during 2018 in the Medicine Bow National Forest. There were no precipitation events during the time of sampling. |
| Location | Coordinates for all sampling plots can be found in Supplementary Data 1. |
| Access & import/export | We did not have to obtain any permits or import/export any sample material. |
| Disturbance | Small volumes of soil were collected within burn scars, and as such any disturbances were extremely minimal. |

# Reporting for specific materials, systems and methods

We require information from authors about some types of materials, experimental systems and methods used in many studies. Here, indicate whether each material, system or method listed is relevant to your study. If you are not sure if a list item applies to your research, read the appropriate section before selecting a response.

### Materials & experimental systems

| n/a | Involved in the study |
|---|---|
| ☒ | ☐ Antibodies |
| ☒ | ☐ Eukaryotic cell lines |
| ☒ | ☐ Palaeontology and archaeology |
| ☒ | ☐ Animals and other organisms |
| ☒ | ☐ Clinical data |
| ☒ | ☐ Dual use research of concern |

### Methods

| n/a | Involved in the study |
|---|---|
| ☒ | ☐ ChIP-seq |
| ☒ | ☐ Flow cytometry |
| ☒ | ☐ MRI-based neuroimaging |

