## [Peer Review File · Nature Microbiology]

Peer Review Information

Journal: Nature Microbiology

Manuscript Title: Wildfire-dependent changes in soil microbiome diversity and function

Corresponding author name(s): Mike Wilkins

Reviewer Comments & Decisions:

Decision Letter, initial version:

Dear Dr. Wilkins,

Thank you very much for your enquiry about submitting your manuscript "Playing with FiRE: A genome resolved view of the soil microbiome responses to high severity forest wildfire" to Nature Microbiology. It certainly sounds interesting, and we would be happy to consider it for publication. However, I'm sure you'll understand that we cannot make a firm decision about whether to send the paper out to review until we have carefully read the full paper (and appropriate background literature). We also have some concerns with the level of insight provided into how these data would guide ecological restoration.

In order to submit your complete manuscript to Nature Microbiology, please use the link below:

{redacted}

If you have any questions, please feel free to contact me.

Yours sincerely,

{redacted}

Decision Letter, first revision:

Dear Dr Wilkins,

Thank you for your patience while your manuscript "Playing with FiRE: A genome resolved view of the soil microbiome responses to high severity forest wildfire" was under peer-review at Nature Microbiology. It has now been seen by 4 referees, whose expertise and comments you will find at the end of this email. Although they find your work of some potential interest, they have raised a number of concerns that will need to be addressed before we can consider publication of the work in Nature Microbiology.

In particular, the referees are concerned that the study doesn't currently focus enough on the novel aspects of the work and instead also has a lot of already known aspects in it. Previous literature needs to be better cited, some of the conclusions toned down, and/or better supported by experiments, and

2there are also some technical concerns regarding the comparative genomic analyses. Further, the statistics need improvement. Editorially, we suggest to reframe the study and not present it as a database, but instead as a multi-omics overview and to focus on the novel insights.

Should further experimental data allow you to address these criticisms, we would be happy to look at a revised manuscript.

Please include a data availability statement as a separate section after Methods but before references, under the heading "Data Availability". This section should inform readers about the availability of the data used to support the conclusions of your study. This information includes accession codes to public repositories (data banks for protein, DNA or RNA sequences, microarray, proteomics data etc...), references to source data published alongside the paper, unique identifiers such as URLs to data repository entries, or data set DOIs, and any other statement about data availability. At a minimum, you should include the following statement: "The data that support the findings of this study are available from the corresponding author upon request", mentioning any restrictions on availability. If DOIs are provided, we also strongly encourage including these in the Reference list (authors, title, publisher (repository name), identifier, year). For more guidance on how to write this section please see: <http://www.nature.com/authors/policies/data/data-availability-statements-data-citations.pdf>

- * Include a "Response to referees" document detailing, point-by-point, how you addressed each referee comment. If no action was taken to address a point, you must provide a compelling argument. This response will be sent back to the referees along with the revised manuscript.
- * If you have not done so already we suggest that you begin to revise your manuscript so that it conforms to our Letter format instructions at <http://www.nature.com/nmicrobiol/info/final-submission>. Refer also to any guidelines provided in this letter.
- * Include a revised version of any required reporting checklist. It will be available to referees (and, potentially, statisticians) to aid in their evaluation if the manuscript goes back for peer review. A revised checklist is essential for re-review of the paper.

2When submitting the revised version of your manuscript, please pay close attention to our [Digital Image Integrity Guidelines](https://www.nature.com/nature-research/editorial-policies/image-integrity) and to the following points below:

{redacted}

Note: This url links to your confidential homepage and associated information about manuscripts you may have submitted or be reviewing for us. If you wish to forward this e-mail to co-authors, please delete this link to your homepage first.

Nature Microbiology is committed to improving transparency in authorship. As part of our efforts in this direction, we are now requesting that all authors identified as 'corresponding author' on published papers create and link their Open Researcher and Contributor Identifier (ORCID) with their account on the Manuscript Tracking System (MTS), prior to acceptance. This applies to primary research papers only. ORCID helps the scientific community achieve unambiguous attribution of all scholarly contributions. You can create and link your ORCID from the home page of the MTS by clicking on 'Modify my Springer Nature account'. For more information please visit www.springernature.com/orcid.

If you wish to submit a suitably revised manuscript we would hope to receive it within 6 months. If you cannot send it within this time, please let us know. We will be happy to consider your revision, even if a similar study has been accepted for publication at Nature Microbiology or published elsewhere (up to a maximum of 6 months).

{redacted}

Reviewer Expertise:Referee #1: Fire ecology, soil carbon and nitrogen dynamics

Referee #2: Soil microbiomes, metagenomics

Referee #3: Multi-omics, soil ecology, environmental microbiology

Referee #4: Soil microbiomes, metagenomics

Reviewer Comments:

Reviewer #1 (Remarks to the Author):

General comments

This paper from Nelson et al. is an impressive study that effectively leverages complex datasets to investigate differences in the soil microbial community composition and function in wildfire sites and across contrasting levels of burn severity. The manuscript includes well-focused objectives and addresses multiple current gaps in scientific knowledge, specifically (1) relationships between microbial community composition and metabolic functions and (2) soil functional response to wildfire and across contrasting levels of burn severity. The manuscript is well-written and information is presented in a logical, well-organized, and convincing way. Figure content and quality are excellent. I focused my review primarily on the fire ecology and biogeochemical aspects of the manuscript. The sampling approach is appropriate and the suite of detailed microbial and metabolic analyses provides powerful insight to post-fire soils. This study will be an important contribution to the microbial and fire ecology literature. My main critique is that the manuscript will be strengthened via minor adjustments to language that better place the study into context. First, this applies to considering in more depth how a one year time period between the wildfire and the soil sampling might influence the observed results. Second, this applies to ensuring results are presented within the constraints of the experiment: i.e., as a comparison between burned vs unburned and across a burn severity gradient, at 1 year post-fire, rather than over-reaching the context of your study via statements that fire "caused" a given response. Both soils and fire are very variable. Without being able to compare differences between paired prefire and postfire samples, we are limited to describing differences among areas which represent unburned versus burned conditions. Tightening this language throughout the manuscript (text, figure captions, supporting information) is important.

Additional references that may be relevant for context:

- Adkins, 2021. Post-fire effects of soil heating intensity and pyrogenic organic matter on microbial anabolism. *Biogeochemistry* 154:555-571. <https://doi.org/10.1007/s10533-021-00807-6>
- Dove et al., 2017. Fire reduces fungal species richness and in situ mycorrhizal colonization: A meta-analysis. *Fire Ecology* 13: 37–65
- Dove et al., 2020. High-severity wildfire leads to multi-decadal impacts on soil biogeochemistry in mixed-conifer forests. *Ecological Applications* 30(4):e02072. [10.1002/eap.2072](https://doi.org/10.1002/eap.2072)

Finally, because your study used low-severity and high-severity sites, I recommend expanding the title to end with "...microbiome responses to wildfire."

Specific comments

Introduction

4L80: revise to "extent of vegetation..." to avoid any confusion with temperature degrees.

Methods

L99-101: here I miss a description of the forest age and/or structure and how much variability existed within and between wildfire sites. How did forest age or structure influence the fire characteristics and correspond with the observed burn severity? Was the forest even-age? Was the forest planted or did it regenerate naturally from a previous fire? What are the previous known dates for past fires at these sites?

L101: Here we miss information about soil classification. Was soil taxonomy and texture similar or variable across the study sites?

L101: specify when after each fire was field validation performed. What other characteristics differed among severity levels that have potential to influence your one year post-fire results? For example, abundance or cover by live vegetation (herbaceous species or shrubs responding to an opened canopy? Live trees remaining? Thickness of any post-fire new litter layer? Etc.)

L102: give reference for source of the BAER maps

L103: how were slope, aspect, elevation used in the statistical analysis? To what extent do these characteristics help explain results?

L106: here it would be helpful also to report the mean depth of organic horizon or ash layer in each severity. I appreciate that the authors provide photos of their study site and severity levels.

Results:

L307 consider revising to "Bacterial and fungal communities were significantly different between burned (n=144) and unburned (n=32) soils (bacterial...)"

L321: why "organic layers" (plural)? In methods sampling the O-horizon, i.e. litter and duff, was described as a single layer collected. Assuming each sample collection location provided a single sample of the organic horizon, then in L321 and throughout the manuscript I recommend using "organic layer" or "organic layer samples" (singular layer).

L334 & elsewhere, consider just "surficial organic horizons" for simplicity & to avoid any potential for confusion with an organic soil.

L335: this depends on the moisture content of the organic horizon, which when dry can contribute to sustained soil heating. See Hartford, R., and W. Frandsen. 1992. When it's hot, It's hot... Or maybe it's not! (Surface flaming may not portend extensive soil heating). *International Journal of Wildland Fire* 2:139-144.

L396: "surficial organic horizons in forest soils"

L396: because (1) severity is a consequence of the fire, not a descriptor of its behavior while active, and because (2) any level of heating during fire is a pulse disturbance (of varying magnitude and impact) and intensity of heating was not recorded for your study, and (3) pulse disturbance from heat during wildfire is not just limited to organic horizons but may also be experienced into the mineral soil, and because (4) the organic horizon is thermochemically fundamentally transformed by fire into ash and char, I recommend revision to "The heat produced during wildfire exerts a pulse disturbance on soils."

This revision opens up the statement to be more accurate, and to avoid limiting the following statement only to organic horizons. If you wish to keep the paragraph focused on O horizons, then revise to limit the statement to apply to your study system rather than seem like a universal truth; some high intensity wildfires cause very minimal soil heating depending on ecosystem type, fire rate of spread, etc.

L409: are you sure this is just in organic horizons, which typically are combusted or pyrolyzed during fires in western forests? I recommend avoiding these types of universal statements which are also noticeably uncited (here and as noted previously). A more accurate statement is: "Soil heating during wildfire can lyse heat-sensitive..." Where organic matter is combusted, it is unlikely that labile C and N influx to the horizon rather than become gasified/volatized, pyrolyzed, combusted, convected away in smoke etc. Also keep in mind that many inputs & outputs have occurred in these soils between the time of fire and time of sampling. The influx of labile org C & N associated with necromass may come from adjacent areas in the soil, from downslope translocation, and so on. In general, align your interpretation with the constraints of your study: you are comparing unburned to burned sites of contrasting severity levels, one year after fire. It is important to avoid over-attributing differences as being directly caused by fire when many confounding factors occur over a one-year post-fire time period.

L420: colonizing or recovering in

L432-435: move to Methods

L439: this study did not assess characteristics or microbial composition immediately after fire. Revise to avoid misleading statement.

L515: consider "unique organic compounds"?

L517-518: because you collected one depth increment of mineral soil, revise to "These aromaticity index trends were not identified in DOM released from mineral horizons, which are more insulated from heat than are organic horizons during fire."

L605: To what extent might the timing of sampling influence these results, i.e., recovery during the one year lag between fire and sampling.

L685: Your study can assess differences among sites with contrasting post-fire characteristics/burn severities, but cannot assess change due to fire without a true pre-fire/post-fire comparison. Please revise to "we observed short-term (one year post-fire) differences in..."

L687-88: revise to clarify that the community diversity is what is being cited by ref #99 for ability to influence ecosystem function. (We think burn severity will, but data on this are still limited).

L690: ammonium concentration

L698: "to assess how C and N function changes as a function of time since fire" or similar in line with previous comment about framing results when you have unburned reference areas rather than pre-fire and post fire samples from the same location. Soil is variable, as is fire and its direct effects such as soil heating, extent of OM combustion or thermochemical conversion, etc. You have an excellent study with results that suggests how fire might change soil characteristics, and constraining results & discussion statements to language about "differences" instead of "changes" does not weaken the value of your impressive study.

L706: Same comment as above, because your study can infer (but did not measure) fungal or host plant mortality, revise to "... (Table S3), which could result from direct heat-induced fungal mortality or plant host death."

L715 as above, "we show that wildfire and the post-fire soil chemical environment contributes to a microbiome that..." or "is associated with a microbiome that..."

L715-720 as a key take-away, this seems to belong in a Conclusions paragraph. Otherwise, move up to where other Fig 5 information is presented.

L718 & L729: specify what these implications are

Conclusion

L731: omit "across fires" as your study was limited to two fires and most analysis were performed on

only one.

L732-733: in general this final statement feels like an unsupported throw-away. This detailed data on microbial community composition and function is very important but is unlikely to be used to inform management unless there is a clear connection to specific actions managers can feasibly take at the scale of a wildfire. Unless you specify how this would be accomplished, leave it out to avoid detracting from the impressive basic science knowledge advancement your study provides, which is its strength. Keep the Conclusions focused on 3-5 clear take-away messages that are clearly supported by this study's results.

Figures & tables

Figure S3: the colored points to indicate significant differences between conditions is a bit confusing. I think the standard letters to indicate statistical differences would be more clear to readers. Panel letters would be helpful, as in main text figures.

Fig S6 & S7: revise "High A soils" to "High A samples" and similar throughout the manuscript: discuss differences between types of samples. The sample represent a soil type (i.e. "a soil") from which O horizon and A horizon samples were collected.

Fig S8: "expressed in the burned mineral horizon" or "expressed in mineral horizon samples from burned areas"

Fig S11: here the four types of samples are called "conditions" but really they are two horizons (O, A) collected from each of two conditions (low severity, high severity), yes? Please check for consistency in language throughout, including in Fig S12 caption. In contrast, Fig S14 caption refers to two conditions. Perhaps define "condition" in Methods then use it consistently through the text.

Reviewer #2 (Remarks to the Author):

The manuscript by Nelson and colleagues describes the use of different 'omics' approaches (metagenomics, metatranscriptomics and FT-ICR metabolomics) to assess how the soil microbiome responds to fires with different degrees of severity and at different soil depths.

General comments:

Some of the results are confirmatory of previously published work, whereas other results are more novel. I suggest that the authors reconsider what to emphasize in this study and to move more confirmatory results to supplemental files. I provide detailed comments and suggestions below.

An additional general comment is that much of the results are speculative. The authors should avoid speculation and over interpretation of their data to fit their hypotheses.

7Specific comments:

1. Abstract: Carbon cycling 'likely' influenced by processing of pyrogenic compounds and turnover by abundant viruses (is this supported in results).
2. Abstract: How specific could results from this study inform restoration efforts?
3. Intro: Note that as the authors mention: post fire shifts in soil microbiome composition are already relatively well characterized. Therefore, this study should not focus as much effort on this aspect.
4. Minor question: The samples for RNA were immediately flash frozen in ethanol dry ice and frozen in field. Why not also for DNA if the possibility existed?
5. Used FTICR-MS to analyze DOM. Note that FTICR cannot assign metabolite identities.
6. Used FUNGuild to classify fungal guilds using fungal ASVs. The retaining of ASVs seems to be subjective on lines 182-185.
7. MGs – 3 per condition, 2 sampling depths. What was the sequencing depth? Total sequence per MG?
8. Line 240: This was confusing to me. Why would the metabolic pathway be considered complete if it is more than 50% complete, because MAGS are commonly not 100% complete?
9. Paragraph starting line 309: The 16S and ITS results were confirmatory of prior studies
10. Figure 1 and other graphs: Recommend different colors to distinguish moderate from high fire intensity.
11. Fig. 1. What is the relative abundance of Firmicutes. Less than 0.5%?
12. Line 366: If myriads of studies have shown community changes after wildfire are the results mainly confirmatory? Maybe could instead highlight the more novel functional results and show less about the community shifts. Lines 308-366: recommend condensing and moving figure 1 and figure 2 to supplemental data if confirmatory only. Could start results from line 368: Here we used genome-resolved metagenomics... Then the Supplemental Figures: S4, S5 and S6 could be moved to main text.
13. Line 390: Provide a short explanation in results for how metatranscriptomic analyses was used to indicate that MAGS were active. E.g. read mapping...
14. Line 393: In addition to MAGs, could use the metatranscriptome information to assess active taxa that were not binned.
15. Section starting line 396: finding of traits in MAGs may or may not indicate that they are involved in

thermal resistance. This is speculation. Lots of 'may' and 'likely' and 'suggesting' in this section. The authors should concede that finding these genes is not proof, but provides some interesting hypotheses to further investigate. By contrast, the peptidase gene expression in High O samples has some validation (Line 414). Similar for section on doubling times: 'suggest' and 'likely'. Try to avoid speculation.

16. Line 466 and elsewhere: I worry that using 637 MAGs to determine microbial composition and enrichment of specific taxa is not a valid approach. Not all populations are binned into MAGs. By contrast, the amplicon data is better for this purpose.

17. Line 484: Not convinced that finding that *Streptomyces* have sporulation genes and ability to metabolize diverse substrates is specific to fire or soil layers....These might be common properties of members of *Streptomyces*.

18. Line 489: Expression data is interesting. Cobalamin... Line 503: increased transcription of cobalamin synthesis is 'likely' a beneficial consequence of wildfire enriching taxa that encode this trait (speculative).

19. Line 506: with all of the benefits of cobalamin – outside of scope of results and speculative.

20. Line 526: 'likely' - avoid speculation.

21. Figure 4: Isn't it known that pyrogenic DOM becomes more aromatic with wildfire burn severity? If so, provide reference. Or is this the first time reported?

22. Section starting line 596 is more novel...

23. Figure 6: Very hard to distinguish phyla based on color scheme.

24. Line 680: Speculation that MAGs can potentially degrade necromass..

25. The emphasis of this study seems to be on the database, which is confusing.

26. Line 725; change "indicate" to "suggest".

Reviewer #3 (Remarks to the Author):

Nelson et al. studies the impact of wildfire burn severity gradients in soil microbiomes of coniferous forests in Colorado and Wyoming, USA. Working hypothesis is that higher severity wildfire would drastically alter the soil microbiome and microbes colonizing burned soils would need to specific metabolic potential and adaptations to thrive in post fire. Author's use a wide range of analysis tools

9(sequencing amplicon, DNA & RNA), FTIR-MS and aim to provide a multi-faceted description of microbial (bacterial, fungal, and viral) potential, activity at sampling time (apprx. one year after fire) & interactions. Manuscript (MS) provides a rich bacterial MAG and viral MAG sets paired with meta transcriptomes to describe post-fire processes that of relevance to long term biogeochemistry and potential vegetation recovery.

While containing some new information mainly tied to exploration of bacterial and viral interactions remainder of the data is mainly reiterate on existing knowledge and does not necessarily explain it further. Burn severity significantly moderates the microbial response to wildfire. Actinobacteria has been identified to be dominant across burned soils, and many soil bacteria can degrade phenolic compounds, such as Arthrobacter, through aerobic metabolism. Data generated for the manuscript needs better connection in between and to the literature. MS is also short on necessary details to understand this work. Materials and Methods (M&M) lack several descriptions that should be included as methods or SI M&M, and analysis methods and statistical reporting (averages, tests and std dev) are limited. There are also several mismatches between writing in MS and data presented. When looked closely some key results may lack sufficient evidence (highlighted Actinobacterial MAGs for example).

In NMDS analysis stress values equal to or below 0.05 indicate good fit; which is not the case for any of the following figures generated outlined findings in L306-L315. Most values reported in Fig.2 & Fig S2 are between 0.1-0.2 which are considered fair. Even with the recognition that there are many studies using this metric in relatively high stress values (0.05-0.10), the calculations (i.e. ranking other than measuring the dissimilarity) in this analysis is robust for data which do not have an identifiable distribution. Preferably authors should have explored other approaches that could identify the gradients that they are seeking. But looks like this representation is a result of another and rather big assumption; in line L191 authors write "16S rRNA gene data confirmed that the soil microbial communities were not significantly different." referring to the two study sites which are more than 20 km. Data is not shown to describe the similarities among these two sites but based on this assumption authors continue to analyze both fire locations together. This is very unexpected as sites that are geographically apart tend to contain significantly different soil microbes, so some difference is expected. It would be beneficial here to set the stage by identifying the appropriate gradients, but neither analysis parameters used in Qiime nor the distance metric (which is has a major impact on the results) are identified. There is enough replication in amplicon dataset to be able to explore a better way forward. Results presented in L306-L322 requires careful reconsideration and appropriate evidence.

Database statement (L365): Depositing sequencing data and its products into NCBI doesn't make it a database but a data entry. Not this study but NCBI is the database as they are the ones providing aggregated data and interphase to interact with it. MS lacks reporting standards for metagenomes, metatranscriptomes and MAGs.

Supplementary Data 2 - Bacterial MAG relative abundance across all 12 samples sequenced for metagenomics calculated with mapping.: There is an inconsistency in the MS and results reported in this table. Per Supp Data 2 all MAGs together map to 81-95% of the reads per sample; which is a substantial number as most soil studies report somewhere between 10-20% of the available reads to be assembled and binned into MAGs. Additionally, looking at the reported relative abundances per MAG for eight Actinobacteria MAGs, that are deemed important, their distribution in different horizons and burn states

10are different than those reported in MS. Authors write (L389-390) " Combined, these MAGs accounted for an average relative abundance of 19% in High O soils and 12% in O-horizon soils impacted by low severity wildfire ('Low O' soils)." Supplementary Data 2 shows High Burn O horizon: 18.1 +/- 7.1 %; Low Burn O horizon: 8.0 +/- 6.5 %. Standard deviations are quite high suggesting that not each sample has these MAGs. Same set of eight MAGs has following levels in mineral soils: High Burn A horizon: 0.8 +/- 0.4 %; Low Burn A horizon: 5.2 +/- 6.9 %. Comparing samples by testing in High Burn O vs A t-test(welch) = 4.23, p=0.051 n=6 and in O-Horizon High vs Low t-test(welch) = 3.97, p=0.142 n=6 both show no significance. Same goes for Low and A horizon comparisons. It would be interesting to take a look at transcription data, but this is not provided. But strong findings written about it by L391-394. All together these results only show that the importance of these MAGs is yet to be clarified.

Doubling time estimations and Fig 3B: Firstly, authors might be missing an opportunity here by just adhering to removing CUB with > 5hrs values. gRodon cannot accurately differentiate between long doubling times but 5 h is a pragmatic default, maybe not a biologically meaningful one. Infact Weissman et al. writes "5 h threshold suggests a natural definition of an oligotroph as an organism ..." They suggest another important filtering (not clear if it is applied in this MS) for data/result QC by making sure that each MAG has >10 ribosomal proteins. Secondly, analysis shown in here is not necessarily informative. Figure itself show it that many (predicted) fast and slow growers are very low (bw 0 to < 0.5%) in abundance. Even looking MAGs with relative abundance > 0.5% a good number still falls above the calculated raw average. It would be useful to have a correlation analysis with appropriate statistics showing that predicted fast growth is indeed can be associated with high MAG abundance (or is the shaded area in Fig3B is showing the boundaries of significance?).

"Soil chemistry" section only covers Nitrogen species so better titled as what it is. Soil moisture and pH are very important parameters but either not measured or reported. Soil pH increases after fire and shown to impact microbial communities. Soil moisture is not reported (but recorded in MS as measured). Nitrogen availability within a few years after fire is generally high mainly due to fire-induced mineralization. Testing Control vs Burn Severity in mineral A horizons result reported (Supp Data1) only ammonia concentrations in High and Moderate burn sites seem to be statistically different [t-test(welch)=-2.84, p=0.035, n=11 Ryan-Mod; t-test(welch)=-3.11, p=0.023 n=11 Ryan-High] than the control sites with no significant changes in nitrate concentrations. As no Control samples were sequenced it is unclear that which ammonia oxidizing bacteria and archaea reside in these soils. Study is likely having a sample sequence coverage issue where the sequencing effort is not sufficient to cover all high/low abundant & rare populations, including those carrying amoA. Other potential N-related processes such as use of urea or protein degradation is not covered so source of N is unknown. This is critical to the arguments of the paper since fire is argued to open up niches for fast growing organisms which require high C and N. In theory, increases in inorganic N in post-fire soils should be favorable to copiotrophic microorganisms but potential copiotrophs are assumed to be in O-horizon not in A-horizon. All together this is a mix of results that require better framing and rethinking. In parallel, statements on PyC lability can use better integration with MAG and transcriptomic data. It is more thermodynamically favorable to oxidize compounds with a high NOSC in reducing (anoxic) environments. More oxidized DOM, in having been broken down by aerobic microbes should exhibit lower aromaticity (and maybe have a lower N content). That's show in low-NOSC index and high catA activity in Low severity burn site relative to High burn severity. There is no info (or evidence) for presence of anoxic conditions in high burn severity site, NOSC index is high (L518 is not particularly useful here) and overall transcriptome

activity on aromatic compounds is low (relative to low burn site). Looks like fire created substrates are yet to spark microbial activity in high burn severity sites. It is too early to make statements like L705 & L715.

General Comments:

Please provide a table describing the results of metagenome and metatranscriptome sequencing, assembly processes. It would be advisable to use a metric such as Nonpareil to determine the coverage and estimated number of species in these samples to show that indeed this MAG dataset corresponds to 81-95% of the population.

Reporting statistics: When calling significant one must report the test and p-value

MS uses active vs abundant interchangeably which is difficult to decipher from the writing at times. It would be better if activity can be strictly designated to metatranscriptomes.

Author's should consider moving FTIR-MS analysis to earlier to describe the changes in DOM and measured N-species together.

Description of post fire soil conditions is limited. MS would benefit from explaining difference in high -to-low severities shown in Fig.S1 and some description of surviving plant cover at the time of sampling. Likewise, climate and soil conditions at the time of sampling warrant description.

Point-by-point Comments:

L75-76: Statement suggesting the studies focused on post-fire soil microbiome is mostly descriptive is not accurate – there are some interesting ecological theory applications taking advantage of fire disturbance. Here are few examples:

Zhang L, Ma B, Tang C, Yu H, Lv X, Rodrigues JL, Dahlgren RA, Xu J. Habitat heterogeneity induced by pyrogenic organic matter in wildfire-perturbed soils mediates bacterial community assembly processes. *The ISME Journal*. 2021 Jan 29:1-3.

Yang S, Zheng Q, Yang Y, Yuan M, Ma X, Chiariello NR, Docherty KM, Field CB, Gutknecht JL, Hungate BA, Niboyet A. Fire affects the taxonomic and functional composition of soil microbial communities, with cascading effects on grassland ecosystem functioning. *Global change biology*. 2020 Feb;26(2):431-42.

Knelman JE, Schmidt SK, Garayburu-Caruso V, Kumar S, Graham EB. Multiple, compounding disturbances in a forest ecosystem: fire increases susceptibility of soil edaphic properties, bacterial community structure, and function to change with extreme precipitation event. *Soil Systems*. 2019 Jun;3(2):40.

L104: Nitrogen species are not reported for mineral soils in Supp1 as stated

L167: Parameters used for analysis should be included as M&M or SI which ever is appropriate

12L191: Author's write "16S rRNA gene data confirmed that the soil microbial communities were not significantly different.". This is very unexpected as sites that are apart tends to look significantly different. Per Fig S1 distance between sites are more than 20 km.

L221: How eukaryotic MAGs were identified is not described. M&M doesn't outline if any effort was put into binning and gives the impression that contigs identified as eukaryotic was treated as a MAG if annotated to the same species. This is just an impression, as it is not described.

L226-228: Author's write "A metabolic pathway within a MAG is considered complete if it is $\geq 50\%$ complete because MAGs are commonly not 100% complete." There are many reasons why a MAG is not complete: reading depth, sample GC distribution, performance of the assembly and binning process, ability to annotate hypothetical pathways. This statement is out of bounds.

L234: Please provide details on how ribosomal RNA was removed from total RNA. No kit or methodology is cited.

L374: Fig 1 and Figure S4 are drawn at phylum scale. It is not possible to one on one compare, please rephrase the sentence

L382-386: Authors write: "MAGs affiliated with the Actinobacteria genera *Blastococcus*, *Mycetocola*, *SISG01*, *SCTD01*, *Nocardiodes*, and *Arthrobacter* were all enriched (relative to control soils) in surface organic soil horizons (O-horizon) impacted by high severity wildfire (hereafter referred to as 'High O') that in most instances had been combusted to an ash layer." But per M&M they did not analyze a control sample via metagenome sequencing. Please revise the statement.

L387: Here please refer to figure showing the enrichment of Actinobacteria MAGs (MAGs RYN_93, RYN_94, RYN_101, RYN_124, RYN_147, RYN_169, RYN_175, RYN_216)

L398: Spore forming is a common Actinobacterial trait that can be found with or without fire. Similarly, heat shock proteins (L401) do exist in almost all genomes. Fig.3A should include standard deviations and appropriate statistics to show the importance of these genes post-fire.

L434-438: Another transcriptome statement with no figure or table to support it

L469: This data needs to better representation. RYN_342 has an abundance of 1.16 +/- 1.32 % so it is abundant in one sample, may not be representative of any other.

L617: VirHostMatcher is not mentioned in M&M or SI M&M

L 685: Please edit for meaning "...in soil mineral soil ammonium.."

L696: Figure S6: Higher activity of CAZY genes in A-horizon than O-horizon is surprising as they are indicative of low C availability & investment in C acquisition. So in the balance of simple(r) C substrates this figure shows that O-horizon is less C limited post-fire than A-horizon it does not "... suggesting that certain C cycling processes may be absent in High O samples."

L721: "... offering opportunities to leverage these results for more effective management of other wildfire-disturbed environments." authors should state what do they mean by leveraging these results as this is

Figure1: no standard deviations or statistics presented

Figure S7: Non Actinobacterial MAGs are circled as MAGs of interest but there is no explanation to why

Figure S8: What does the asterisk donate, significance?

Supp Data 2: RYN_1 has no tax assignment – is this even a Bacteria?

Reviewer #4 (Remarks to the Author):

This study is conceptually straightforward, but the analyses that went into this study are comprehensive. Essentially, the authors wanted to determine how high intensity wildfires influence the microbial communities found in soil post-fire. While there have been many studies investigating wildfire effects on soil microbial communities and microbial processes, this study is unique in that it generated and analyzed metagenome-assembled genomes that were integrated with metabolomic, metatranscriptomic, and marker gene sequencing data. While the techniques used here are clearly 'cutting edge', I do think the authors oversell their work a bit and some of the conclusions are not well-supported by the data presented. The authors could benefit by being more cautious in the presentation of their results and acknowledge some of the uncertainties associated with the chosen study design and the limitations thereof. More detailed comments follow below:

In the Abstract and Introduction, it is frequently mentioned that 'little is known regarding the the impact of high severity fire on microbially-mediated processes' (or equivalent). This is not true – there are many studies looking at how soil trace gas emissions and N dynamics (for example) are affected by wildfire. Perhaps more importantly, this study did not include any information on microbially-mediated process rates (obviously genes and transcripts do not equate with process rates) so it is a bit disingenuous to frame the Introduction in this manner.

It is a bit unclear to me why the authors combined results from the two sites (Ryan and Badger Creek) for many of the analyses, especially since the ANOSIM analyses show that the soil microbial communities are significantly different (lines 201-204). This would seem like a small detail, but combining results from the 2 sites is likely to influence the patterns observed. For example, the differences in beta dispersion evident in Figure 2 could simply be a product of site-level variability in community composition (particularly across the unburned samples).

There is clearly a lot of data that went into this study, but I was surprised there was no data on soil bacterial/fungal biomass as that would have allowed the authors to determine if the 'fire responding' taxa are actually growing after the fire event, or are merely increasing in relative abundance because 'fire sensitive' taxa were removed by the fire event. Even qPCR-based estimates of total bacterial/fungal DNA amounts would have been useful to include. Any conclusions about 'fire responders' being able to grow quickly would need to be supported by some sort of actual biomass estimates.

One of their main conclusions is that taxa found in soils post-fire tend to have more genes for thermal resistance (like heat shock proteins). However, such genes are likely common across a broad diversity of taxa. Likewise, it seems like a bit of a stretch to claim that 'higher GC content may be another heat resistance trait' when many of the 'fire responsive' taxa were Actinobacteria which are known to have high GC content. More generally, identifying the key traits of the 'fire responsive' MAGs would require a comparison against MAGs from 'non-fire responders' and it is not clear if this was done (maybe I missed this important information).

Is it reasonable to claim that MAG abundance-normalized transcripts are a measure of growth rate? (lines 432-440).

The discussion of cobalamin production seems like a bit of a stretch (lines 489-508) – just because the genes/transcripts are there, does not mean cobalamin production is happening at higher rates in fire-affected soils (I know the authors know this, but some caveats would be useful here).

Author Rebuttal, first revision:

Reviewer #1 (Remarks to the

Author): General comments

This paper from Nelson et al. is an impressive study that effectively leverages complex datasets to investigate differences in the soil microbial community composition and function in wildfire sites and across contrasting levels of burn severity. The manuscript includes well-focused objectives and addresses multiple current gaps in scientific knowledge, specifically (1) relationships between microbial community composition and metabolic functions and (2) soil functional response to wildfire and across contrasting levels of burn severity. The manuscript is well-written and information is presented in a logical, well-organized, and convincing way. Figure content and quality are excellent. I focused my review primarily on the fire ecology and biogeochemical aspects of the manuscript. The sampling approach is appropriate and the suite of detailed microbial and metabolic analyses provides powerful insight to post-fire soils. This study will be an important contribution to the microbial and fire ecology literature. My main critique is that the manuscript will be strengthened via minor adjustments to language that better place the study into context. First, this applies to considering in more depth how a one year time period between the wildfire and the soil sampling might influence the observed results. Second, this applies to ensuring results are presented within the constraints of the experiment: i.e., as a comparison between burned vs unburned and across a burn severity gradient, at 1 year post-fire, rather than over-reaching the context of your study via statements that fire "caused" a given response. Both soils and fire are very variable. Without being able to compare differences between paired prefire and postfire samples, we are limited to describing differences among areas which represent

15unburned versus burned conditions. Tightening this language throughout the manuscript (text, figure captions, supporting information) is important.

We thank the reviewer for their extremely constructive comments. Throughout the manuscript, we have removed language so that our discussion follows the caveats of the experimental design. For example, we removed language calling MAGs “fire-responding” and have noted throughout the discussion that these samples were collected one year post-fire to reorient the readers (i.e., L160, L245, L413). We agree with the reviewer that this was an incredibly important change to the manuscript to describe the dataset more accurately. We’ve also responded to all your comments below:

Additional references that may be relevant for context:

- Adkins, 2021. Post-fire effects of soil heating intensity and pyrogenic organic matter on microbial anabolism. *Biogeochemistry* 154:555-571. <https://doi.org/10.1007/s10533-021-00807-6>
- Dove et al., 2017. Fire reduces fungal species richness and in situ mycorrhizal colonization: A meta-analysis. *Fire Ecology* 13:37–65
- Dove et al., 2020. High-severity wildfire leads to multi-decadal impacts on soil biogeochemistry in mixed-conifer forests. *Ecological Applications* 30(4):e02072. [10.1002/eap.2072](https://doi.org/10.1002/eap.2072)

These have all been added to the main text for contextualization and acknowledgement of previous work.

Finally, because your study used low-severity and high-severity sites, I recommend expanding the title to end with “...microbiome responses to wildfire.”

We agree with the reviewer and, for clarification to readers, have changed the title.

Specific comments

Introduction

1.1 - L80: revise to “extent of vegetation...” to avoid any confusion with temperature degrees.

This sentence was revised.

Methods

1.2 - L99-101: here I miss a description of the forest age and/or structure and how much variability existed within and between wildfire sites. How did forest age or structure influence the fire characteristics and correspond with the observed burn severity? Was the forest even-age? Was the forest planted or did it regenerate naturally from a previous fire? What are the previous known dates for past fires at these sites?

We agree with the reviewer that these are important metadata details to include and have added them into the methods section (L476-480). The forest was even-age, old-growth, and dominated by lodgepole pine. There is no record of previous fire at these sites, but it is noted that a previous study found the average fire rotation here to be ~182 years¹.

1.3 - L101: Here we miss information about soil classification. Was soil taxonomy and texture similar or variable across the study sites?

Soil taxonomy and classification information was added to Supplementary Data 1 and to the methods section (L482-484). Soil taxonomy and texture did not vary greatly among sites.

1.4 - L101: specify when after each fire was field validation performed. What other characteristics differed among severity levels that have potential to

influence your one year post-fire results? For example, abundance or cover by live vegetation (herbaceous species or shrubs responding to an opened canopy? Live trees remaining? Thickness of any post-fire new litter layer? Etc.)

We thank the reviewer for pointing out this missing information and agree that it is important to include. We've added in more information about conditions at time of sampling and how severity levels differed (L491-497; also see comment 3.10).

1. 5 -L102: give reference for source of the BAER maps

The reference for where we retrieved the BAER maps has been added (L491).

1.6 - L103: how were slope, aspect, elevation used in the statistical analysis? To what extent do these characteristics help explain results?

There was little variation in elevation between plots (2480-2760 m) and the slopes ranged from gentle to moderate (1.7° to 15.3°). The aspects did differ, so I ran ANOSIM analyses on community compositions to ensure that microbial communities did not significantly differ between north and south facing slopes on moderate sloping plots (L485-488)

1.7- L106: here it would be helpful also to report the mean depth of organic horizon or ash layer in each severity. I appreciate that the authors provide photos of their study site and severity levels.

Unfortunately, we did not take these measurements in the field and cannot provide them but agree they would be useful to report and will do so in future studies.

Results:

1.8 - L307 consider revising to “Bacterial and fungal communities were significantly different between burned (n=144) and unburned (n=32) soils(bacterial....)”

This sentence was revised for clarification.

1.9- L321: why “organic layers” (plural)? In methods sampling the O-horizon, i.e. litter and duff, was described as a single layer collected. Assuming each sample collection location provided a single sample of the organic horizon, then in L321 and throughout the manuscript I recommend using “organic layer” or “organic layer samples” (singular layer).

Throughout the text, we’ve changed the labeling of our different soil samples from organic (O-horizon) and mineral (A-horizon) to surface and deep. We’ve made this change because in control and low severity plots we were sampling at least some O-horizon material at the surface, whereas in moderate and high severity plots it was some combusted O-horizon material (i.e., ash) mixed with mineral soil. We’ve clarified this in the methods (L500) and have changed all labeling to ‘surface’ and ‘deep’ soils that represent 0-5 and 5-10 cm depths.

1.10- L334 & elsewhere, consider just “surficial organic horizons” for simplicity & to avoid any potential for confusion with an organic soil.

See above comment (1.9) about naming changes.

1.11- L335: this depends on the moisture content of the organic horizon, which when dry can contribute to sustained soil heating. See Hartford, R., and W. Frandsen. 1992. When it's hot, It's hot... Or maybe it's not! (Surface flaming may not portend extensive soil heating). International Journal of Wildland Fire 2:139-144.

Thank you for this clarification, we’ve re-worded this sentence to acknowledge this caveat

(L125). Unfortunately, we don't have thermocouple data or similar to inform whether the mineral soil was indeed insulated from wildfire-induced heating but think it's important to note in the text that the sequencing data suggests this is the case.

1.12- L396: "surficial organic horizons in forest soils"

See comment 1.9 about naming changes throughout text.

1.13- L396: because (1) severity is a consequence of the fire, not a descriptor of its behavior while active, and because (2) any level of heating during fire is a pulse disturbance (of varying magnitude and impact) and intensity of heating was not recorded for your study, and (3) pulse disturbance from heat during wildfire is not just limited to organic horizons but may also be experienced into the mineral soil, and because (4) the organic horizon is thermochemically fundamentally transformed by fire into ash and char, I recommend revision to "The heat produced during wildfire exerts a pulse disturbance on soils." This revision opens up the statement to be more accurate, and to avoid limiting the following statement only to organic

horizons. If you wish to keep the paragraph focused on O horizons, then revise to limit the statement to apply to your study system rather than seem like a universal truth; some high intensity wildfires cause very minimal soil heating depending on ecosystem type, fire rate of spread, etc.

We agree with your statement that this statement was not accurate across systems and fires and have revised to the recommended sentence to broaden and increase accuracy. This also works well because the next sentence encompasses both surface and deep soils, after which we focus on only surface soils.

1.14- L409: are you sure this is just in organic horizons, which typically are combusted or pyrolyzed during fires in western forests? I recommend avoiding these types of universal statements which are also noticeably uncited (here and as noted previously). A more accurate statement is: “Soil heating during wildfire can lyse heat- sensitive...” Where organic matter is combusted, it is unlikely that labile C and N influx to the horizon rather than become gasified/volatized, pyrolyzed, combusted, convected away in smoke etc. Also keep in mind that many inputs & outputs have occurred in these soils between the time of fire and time of sampling. The influx of labile org C & N associated with necromass may come from adjacent areas in the soil, from downslope translocation, and so on. In general, align your interpretation with the constraints of your study: you are comparing unburned to burned sites of contrasting severity levels, one year after fire. It is important to avoid over-attributing differences as being directly caused by fire when many confounding factors occur over a one-year post-fire time period.

We have changed this statement to the recommended wording and have also gone through the manuscript to avoid overinterpretation of the data within the constraints of our study.

1.15- L420: colonizing or recovering in

We've changed to align with the constraints of our study.

1.16- L432-435: move to Methods

We've decided to keep this statement (using MAG abundance-normalized transcripts for ribosomal proteins and central metabolisms) because we think it helps explain to the reader how we addressed whether the MAGs with the fastest maximum growth rate (via gRodon

using codon usage bias) were potentially the fastest current growers in the system. See comment 4.5 for more justification on the utility of looking at ribosomal and central metabolism transcripts as a proxy of growth rate.

1.17- L439: this study did not assess characteristics or microbial composition immediately after fire. Revise to avoid misleading statement.

This statement was revised to *'Together, these analyses indicate that potential rapid growth rates could enable 'fast-responders' to occupy free niche space in soil immediately following wildfire, but this may not maintained once those niches are filled'* so that it is worded as more of a hypothesis from our dataset instead of a conclusion we can make from the dataset (L231).

1.18- L515: consider "unique organic compounds"?

This was changed for clarity.

1.19- L517-518: because you collected one depth increment of mineral soil, revise to

“These aromaticity index trends were not identified in DOM released from mineral horizons, which are more insulated from heat than are organic horizons during fire.” This sentence was revised to avoid any confusion regarding the soil sampling.

1.20- L605: To what extent might the timing of sampling influence these results, i.e., recovery during the one year lag between fire and sampling.

-RNA and DNA viruses not significantly different between low/high severity fire but differed between the two soil horizons

We agree that the RNA and DNA viruses are not significantly different between low/high severity fire likely due to recovery over the one year post-fire and have added this into the text (L339).

1.21- L685: Your study can assess differences among sites with contrasting post-fire characteristics/burn severities, but cannot assess change due to fire without a true pre-fire/post-fire comparison. Please revise to “we observed short-term (one year post-fire) differences in...”

We revised this statement to avoid any overinterpretation of the dataset.

1.22- L687-88: revise to clarify that the community diversity is what is being cited by ref #99 for ability to influence ecosystem function. (We think burn severity will, but data on this are still limited).

The sentence was reworded to clarify that reduced community diversity is expected to influence ecosystem function.

1.23- L690: ammonium concentration

‘Concentration’ was added for clarification.

1.24- L698: “to assess how C and N function changes as a function of time since fire” or similar in line with previous comment about framing results when you have unburned reference areas rather than pre-fire and post fire samples from the same location. Soil is variable, as is fire and its direct effects such as soil heating, extent of OM combustion or thermochemical conversion, etc. You have an excellent study with results that suggests how fire might change soil

characteristics, and constraining results & discussion statements to language about “differences” instead of “changes” does not weaken the value of your impressive study.

We've reframed this section of the manuscript and this statement is no longer included.

1.25- L706: Same comment as above, because your study can infer (but did not measure) fungal or host plant mortality, revise to “...(Table S3), which could result from direct heat-induced fungal mortality or plant host death.”

This statement was revised to reduce overinterpretation of results.

1.26- L715 as above, “we show that wildfire and the post-fire soil chemical environment contributes to a microbiome that...” or “is associated with a microbiome that...”

This statement was revised to ‘...contributes to a microbiome that...’ to reduce overinterpretation of results.

1.27- L715-720 as a key take-away, this seems to belong in a Conclusions paragraph. Otherwise, move up to where other Fig 5 information is presented.
We agree that this idea fits better in the pyrogenic C section and have moved it to L281.

1.28- L718 & L729: specify what these implications are
The implications are elaborated on here in L281-L283.

1.29 - Conclusion

L731: omit “across fires” as your study was limited to two fires and most analysis were performed on only one.

“Across fires” was removed to eliminate overinterpretation of results.

1.30- L732-733: in general this final statement feels like an unsupported throw-away. This detailed data on microbial community composition and function is very important but is unlikely to be used to inform management unless there is a clear connection to specific actions managers can feasibly take at the scale of a wildfire. Unless you specify how this would be accomplished, leave it out to avoid detracting from the impressive basic science knowledge advancement your study provides, which is its strength. Keep the Conclusions focused on 3-5 clear take-away messages that are clearly supported by this study’s results.
We have re-worded the conclusion paragraph and have removed any connection to restoration strategies while focusing on the key novel scientific findings of the study.

Figures & tables

1.31 - Figure S3: the colored points to indicate significant differences between conditions is a bit confusing. I think the standard letters to indicate statistical differences would be more clear to readers. Panel letters would be helpful, as in main text figures.

Panel letters were added, and the colored points were changed to colored asterisks (with description in figure caption) to match how significance was represented in other figures.

251.32 - Fig S6 & S7: revise “High A soils” to “High A samples” and similar throughout the manuscript: discuss differences between types of samples. The sample represent a soil type (i.e. “a soil”) from which O horizon and A horizon samples were collected. See comment 1.9 for description of naming changes of sample conditions throughout text. We have changed all instances of “High/Low X (S/D) soils” to “High/Low X (S/D) samples”.

1.33- Fig S8: “expressed in the burned mineral horizon” or “expressed in mineral horizon samples from burned areas”

This was revised for simplicity.

1.34- Fig S11: here the four types of samples are called “conditions” but really they are two horizons (O, A) collected from each of two conditions (low severity, high severity), yes? Please check for consistency in language throughout, including in Fig

S12 caption. In contrast, Fig S14 caption refers to two conditions. Perhaps define “condition” in Methods then use it consistently through the text.

We've defined the four separate 'conditions' for metagenomic and metatranscriptomic sequencing samples as 'Low S', 'Low D', 'High S', and 'High D' (S - shallow; D - deep). This was changed from O/A (see response to comment 1.9 and L500). These conditions are defined in the methods in the 'Metagenomic assembly and binning' section (L619).

Reviewer #2 (Remarks to the Author):

The manuscript by Nelson and colleagues describes the use of different 'omics' approaches (metagenomics, metatranscriptomics and FT-ICR metabolomics) to assess how the soil microbiome responds to fires with different degrees of severity and at different soil depths.

General comments:

Some of the results are confirmatory of previously published work, whereas other results are more novel. I suggest that the authors reconsider what to emphasize in this study and to move more confirmatory results to supplemental files. I provide detailed comments and suggestions below.

An additional general comment is that much of the results are speculative. The authors should avoid speculation and over interpretation of their data to fit their hypotheses.

Thank you to the reviewer for their constructive comments. We agree that the compositional results are confirmatory of previous studies and have moved these results to the supplementary information (Supplementary Note 1) to focus on more novel findings from our study (see comments 2.3, 2.12). We use the speculative language ('likely', 'suggest') due to the speculative nature of metagenomics and metatranscriptomics datasets and have retained some of this language throughout the manuscript (comments 2.15, 2.18). We like to use these terms to avoid overselling what the data can tell us but have removed these terms where appropriate.

Specific comments:

2.1 - 1. Abstract: Carbon cycling 'likely' influenced by processing of pyrogenic compounds and turnover by abundant viruses (is this supported in results).

Yes, we believe this statement is supported in the data via both abundant genomic evidence of aromatic C degradation pathways and the presence of transcripts that map to these pathways indicating activity (see '*Actinobacteria catalyze degradation of pyrogenic organic matter*' section). Regarding viral roles in C cycling, we also detect metatranscriptomic reads mapping to viral AMGs related to SOM degradation (see '*Viral dynamics impact community structure and function of the burned soil microbiome*' section).

2.2 - 2. Abstract: How specific could results from this study inform restoration efforts?

We have removed the connection of this dataset to informing restoration efforts in the abstract, introduction, and conclusion, because of this comment and related comments by other reviewers.

2.3 - 3. Intro: Note that as the authors mention: post fire shifts in soil microbiome composition are already relatively well characterized.

Therefore, this study should not focus as much effort on this aspect.

We moved most of the compositional details to the supplementary information and moved Figure 1 to the SOM as Figure S1. Some of the results from the 16S and ITS sequencing were retained in the main text (e.g., the depth-resolved differences between organic and mineral soil samples) because we think they set up the next sections nicely.

2.4 - 4. Minor question: The samples for RNA were immediately flash frozen in ethanol dry ice and frozen in field. Why not also for DNA if the possibility existed? The flash freezing in the field was only used for samples for RNA analysis due to the short half-life of RNA. In contrast, DNA has a longer half-life and can be preserved well in the field on dry ice alone. The flash freezing method was not utilized for all DNA samples because it was not necessary and would be too time-intensive for the 176 samples collected.

2.5 - 5. Used FTICR-MS to analyze DOM. Note that FTICR cannot assign metabolite identities.

We are aware of this limitation of FTICR-MS data and did not attempt to assign metabolite identities to any peaks or assigned formulas, only used the data to identify general aromaticity trends.

2.6 - 6. Used FUNGuild to classify fungal guilds using fungal ASVs. The retaining of ASVs seems to be subjective on lines 182-185.

We changed which FUNGuild guild assignments we retained to follow recent literature using FUNGuild and the recommendations in the primary FUNGuild paper ². We accepted all guild assignments that were classified as 'probable' or 'highly probable' and only retained those that were classified as a single guild, to avoid any misinterpretation (L599-603).

2.7 - 7. MGs – 3 per condition, 2 sampling depths. What was the sequencing depth? Total sequence per MG?

Important metagenomics and metatranscriptomics sequencing information (e.g., sequencing depth, % reads assembled and binned) have been added to Supplementary Data 2 in Sheet A.

2.8 - 8. Line 240: This was confusing to me. Why would the metabolic pathway be considered complete if it is more than 50% complete, because MAGS are

commonly not 100% complete?

This statement was removed to eliminate confusion.

2.9 - 9. Paragraph starting line 309: The 16S and ITS results were confirmatory of prior studies

We moved most of the compositional findings and details to the supplementary information.

2.10 - 10. Figure 1 and other graphs: Recommend different colors to distinguish moderate from high fire intensity.

All figure colors were changed to better distinguish the different conditions. These new colors are still color blind friendly.

2.11 - 11. Fig. 1. What is the relative abundance of Firmicutes. Less than 0.5%?

The average overall relative abundance of Firmicutes is 1.21% and was added to the figure caption (now Figure S4). It is in the inset because its percent change with burn is so much larger than any of the other phyla and would make it difficult for the reader to see other trends. The note of the relatively low Firmicutes relative abundance was added to the figure caption to explain why we didn't discuss the phyla in the main text.

2.12 - 12. Line 366: If myriads of studies have shown community changes after wildfire are the results mainly confirmatory? Maybe could instead highlight the more novel functional results and show less about the community shifts. Lines 308-366: recommend condensing and moving figure 1 and figure 2 to supplemental data if confirmatory only. Could start results from line 368: Here we used genome-resolved metagenomics... Then the Supplemental Figures: S4, S5 and S6 could be moved to main text.

We moved most of the compositional findings and details to the supplementary information (Supplementary Note 1).

2.13 - 13. Line 390: Provide a short explanation in results for how metatranscriptomic analyses was used to indicate that MAGS were active. E.g. read mapping...

We've changed 'metatranscriptomic analyses...' to 'the mapping of metatranscriptomic reads...' to clarify which metatranscriptomic analyses were done.

2.14 - 14. Line 393: In addition to MAGs, could use the metatranscriptome information to assess active taxa that were not binned.

We did do this to specifically look at inorganic N cycling genes; we annotated the assemblies using DRAM and mapped metatranscriptomic reads to them to quantify the expression of inorganic N genes that were not binned. This analysis is included in the 'Ecosystem implications' section. We did not discuss the assembly-based analysis in this section because we decided to focus on the genome-resolved methodology.

2.15 - 15. Section starting line 396: finding of traits in MAGs may or may not indicate that they are involved in thermal resistance. This is speculation. Lots of 'may' and 'likely' and 'suggesting' in this section. The authors should concede that finding these genes is not proof, but provides some interesting hypotheses to further investigate. By contrast, the peptidase gene expression in High O

samples has some validation (Line 414). Similar for section on doubling times: 'suggest' and 'likely'. Try to avoid speculation.

We agree that metagenomic and metatranscriptomic data can yield speculative insights. Where possible, we have tried to describe traits and trends that are supported by the coupled omics datasets. It is important to note that we do not expect that all of these traits (e.g., heat resistance) to be expressed one year post-fire, but anticipate that these encoded traits enabled persistence during the disturbance. Additionally, many of these consistent observations come from analyses across multiple MAGs that are dominant in burned soils via both metagenomic and marker gene analysis, thus increasing the confidence of our inferences. However, where we discuss genomic inferences alone we have tried to provide appropriate caveats and note to the reader that this is functional potential and not definitive evidence of the importance of a given trait or pathway. Regarding terms like 'suggest' or 'likely', we use these words to avoid overselling the data, we have also removed overuse of these terms where appropriate.

2.16 - 16. Line 466 and elsewhere: I worry that using 637 MAGs to determine microbial composition and enrichment of specific taxa is not a valid approach. Not all populations are binned into MAGs. By contrast, the amplicon data is better for this purpose.

We agree with this comment and, although we have to use a consistent MAG relative abundance cutoff to select specific 'MAGs of interest' to focus on, we note that many of these MAGs represent taxonomies that were also enriched in high severity-impacted soils in the 16S rRNA gene sequencing data (e.g., L156). We also try to use the marker gene data to inform which MAGs are discussed throughout the text (e.g., paragraph beginning at L433). We do not use the metagenomic data to inform any broader compositional changes and rely solely on the 16S data to understand compositional shifts with burn.

2.17 - 17. Line 484: Not convinced that finding that *Streptomyces* have sporulation genes and ability to metabolize diverse substrates is specific to fire or soil layers....These might be common properties of members of *Streptomyces*.

The metagenomic and 16S data (relative abundance increase of 155% from control to high A soils) shows the enrichment of these groups in High A, so while these properties are likely common within the *Streptomyces* genera, this group is enriched in fire-impacted soils and these commonly-encoded traits likely mediate this. We understand that the text implies that these traits are specific to fire and have added language to clarify this misunderstanding. During the revision process, we decided that this section is less important than the other main findings and have moved it to the SOM (Supplementary Note 2).

2.18 - 18. Line 489: Expression data is interesting. Cobalamin... Line 503: increased transcription of cobalamin synthesis is 'likely' a beneficial consequence of wildfire enriching taxa that encode this trait (speculative).

As mentioned in response to comment 2.15, unfortunately the nature of the dataset is speculative and we use modifiers "likely", "may", etc. to not oversell the dataset to readers. The note that increased cobalamin synthesis is likely a indirect beneficial consequence of wildfire because the production of cobalamin is taxonomically constrained in soils³ and taxa known to encode this trait (*Streptomyces*) are enriched in high severity-impacted mineral soils. Note here that this section was moved to the SOM.

2.19 - 19. Line 506: with all of the benefits of cobalamin – outside of scope of

results and speculative.

We've modified this sentence to remove speculation about benefits for restoration of post-fire landscapes. Note here that this section was moved to the SOM (Supplementary Note 2).

2.20 - 20. Line 526: 'likely' - avoid speculation.

This was removed because this statement is supported by the data presented in the paragraph thereafter.

2.21 - 21. Figure 4: Isn't it known that pyrogenic DOM becomes more aromatic with wildfire burn severity? If so, provide reference. Or is this the first time reported? This finding has been documented in the literature and we have added in references to support this text (L244).

2.22 - 22. Section starting line 596 is more novel...

Because of your previous comments (comments 2.3 and 2.9), we've moved the more confirmatory, less novel text (compositional results) to the SOM and have retained more novel aspects of the work in the main text. Remaining in the main text are the depth-resolved impacts of fire as revealed by amplicon sequencing, genome-resolved metagenome and metatranscriptome results, aromatic C degradation pathways and complementary C chemistry data, fungal MAG findings, viral-host dynamics, and ecosystem implications of fire.

2.23 - 23. Figure 6: Very hard to distinguish phyla based on color scheme.

We agree and have edited the figure so that the main message is clearer for the reader (Figure 5).

2.24 - 24. Line 680: Speculation that MAGs can potentially degrade necromass..

This was rephrased so that it sounds less speculative but we note that the metatranscriptome data (i.e., expression of peptidases) mentioned in the text supports our inferences (L408).

2.25 - 25. The emphasis of this study seems to be on the database, which is confusing. -move first section to methods and/or SOM

The emphasis on the database has been removed and it is now discussed as a new, publicly accessible dataset. We think the first section is important to keep because it frames the data we are discussing throughout the rest of the text, so it has been retained but reworded.

2.26 - 26. Line 725; change "indicate" to "suggest".

This wording has been changed (L464).

Reviewer #3 (Remarks to the Author):

Nelson et al. studies the impact of wildfire burn severity gradients in soil microbiomes of coniferous forests in Colorado and Wyoming, USA. Working hypothesis is that higher severity wildfire would drastically alter the soil microbiome and microbes colonizing burned soils would need to specific metabolic potential and adaptations to thrive in post fire. Author's use a wide range of analysis tools (sequencing amplicon, DNA & RNA), FTIR-MS and aim to provide a multi-faceted description of microbial (bacterial, fungal, and viral) potential, activity at sampling time (apprx. one year after fire) & interactions. Manuscript (MS) provides a rich bacterial MAG and viral MAG sets paired with meta transcriptomes to

35describe post-fire processes that of relevance to long term biogeochemistry and potential vegetation recovery.

While containing some new information mainly tied to exploration of bacterial and viral interactions remainder of the data is mainly reiterate on existing knowledge and does not necessarily explain it further. Burn severity significantly moderates the microbial response to wildfire. Actinobacteria has been identified to be dominant across burned soils, and many soil bacteria can degrade phenolic compounds, such as Arthrobacter, through aerobic metabolism. Data generated for the manuscript needs better connection in between and to the literature. MS is also short on necessary details to understand this work. Materials and

Methods (M&M) lack several descriptions that should be included as methods or SI M&M, and analysis methods and statistical reporting (averages, tests and std dev) are limited. There are also several mismatches between writing in MS and data presented. When looked closely some key results may lack sufficient evidence (highlighted Actinobacterial MAGs for example).

We thank the review for their thorough review and analysis of our manuscript and data and feel these comments have allowed us to strengthen the manuscript. To address your comments, we've re-done all the MAG analyses using a more robust and data-driven set of MAGs by considering both relative abundance of a MAG in the given condition and standard deviation of relative abundance across samples (see L645 and comment 3.3). We've highlighted the novelty of this study by providing functional context and linkages between compositional field studies and laboratory physiological studies of pyrophilous genera. We've included much more of the processed data in the supplementary data files (i.e., all processed metatranscriptomic data in Supplementary Data 4), have added in additional statistics when needed, put more detail in the methods section, and have added discussion regarding chemistry data (comment 3.5; Supplementary Note 3). To further analyze inorganic N cycling in our dataset, we mapped our metatranscriptomic reads to newly annotated assemblies to quantify transcription of unbinned inorganic N cycling genes (comment 3.5). See below for specific responses regarding your comments:

3.1 - In NMDS analysis stress values equal to or below 0.05 indicate good fit; which is not the case for any of the following figures generated outlined findings in L306- L315. Most values reported in Fig.2 & Fig S2 are between 0.1-0.2 which are considered fair. Even with the recognition that there are many studies using this metric in relatively high stress values (0.05-0.10), the calculations (i.e. ranking other than measuring the dissimilarity) in this analysis is robust for data which do not have an identifiable distribution. Preferably authors should have explored other approaches that could identify the gradients that they are seeking. But looks like this representation is a result of another and rather big assumption; in line L191 authors write "16S rRNA gene data confirmed that the soil microbial communities were not significantly different." referring to the two study sites which are more than 20 km. Data is not shown to describe the similarities among these two sites but

based on this assumption authors continue to analyze both fire locations together. This is very unexpected as sites that are geographically apart tend to contain significantly different soil microbes, so some difference is expected. It would be beneficial here to set the stage by identifying the appropriate gradients, but neither analysis parameters used in Qiime nor the distance metric (which is has a major impact on the results) are identified. There is enough replication in amplicon dataset to be able to explore a better way forward. Results presented in L306-L322 requires careful reconsideration and appropriate evidence.

NMDS response - As the reviewer mentioned, there are many studies that use NMDS to visualize trends in their marker gene datasets with stress values > 0.1 in high-impact journals like ISME, Global Change Biology, and Soil Biology and Biochemistry⁴⁻⁶. Additionally, the NMDS plots (Figure 1A-D) do mirror the trends in the diversity data (Figure 1E,F) and we believe that the NMDS analysis, along with the beta dispersion calculations (Figure S4) derived from the aforementioned NMDS, provide convincing evidence of the discussed depth-resolved impacts of wildfire on the soil microbiome. Because of this, we've chosen to retain the NMDS analyses in Figure 1 to discuss the 16S rRNA gene and ITS

amplicon sequencing. We have also added in which distance metric was used for the NMDS (Bray-Curtis dissimilarity; L607)

16S rRNA data different between Badger Creek and Ryan Fire - Unfortunately here the ANOSIM stat result was initially misinterpreted and incorrectly not changed in the text. Although it is unfortunate that the sites are not indistinct from one another, we initially chose the Ryan Fire transect for metagenomics and metatranscriptomics because the Ryan Fire was within the less managed forest of the two, making it more representative of natural forests in the western US. Additionally, although there are some differences between the soil microbial communities of the Ryan and Badger Creek fire sites, we note that the primary taxa we discuss in the manuscript follow similar trends in the two sites. For example, Actinobacteria have a % increase in relative abundance of ~70% & 155% from unburned to burned soils in the Badger Creek and Ryan samples, respectively (including both soil depths together). The Actinobacteria genera that are most discussed in the text as important in post-fire soils also follow similar trends at both sites: *Arthrobacter* (1553% and 237% relative abundance increase with burn in Badger Creek and Ryan), *Blastococcus* (4370% and 1866%), *Modestobacter* (2050% and 8828%), and *Streptomyces* (15% and 132%). Both the Verrucomicrobia and Acidobacteria, discussed as being lost in burned soils, also show similar trends in the 16S rRNA gene sequencing data from both sites: Verrucomicrobia had relative abundance decreases of -6% and -58% from unburned to burned conditions in Badger Creek and Ryan sites, and Acidobacteria had relative abundance decreases of -29% and -37%.

QIIME analysis parameters response - More detailed QIIME analysis parameters were added into the methods section (L591-599).

3.2 - Database statement (L365): Depositing sequencing data and its products into NCBI doesn't make it a database but a data entry. Not this study but NCBI is the database as they are the ones providing aggregated data and interphase to interact with it. MS lacks reporting standards for metagenomes, metatranscriptomes and MAGs.

The emphasis on the database has been removed and it is now discussed as a new, publicly accessible dataset. Reporting standards have been added to Supplementary Data 2.

3.3 Supplementary Data 2 - Bacterial MAG relative abundance across all 12 samples sequenced for metagenomics calculated with mapping.: There is an

39inconsistency in the MS and results reported in this table. Per Supp Data 2 all MAGs together map to 81-95% of the reads per sample; which is a substantial number as most soil studies report somewhere between 10-20% of the available reads to be assembled and binned into MAGs. Additionally, looking at the reported relative abundances per MAG for eight Actinobacteria MAGs, that are deemed important, their distribution in different horizons and burn states are different than those reported in MS. Authors write (L389-390) " Combined, these MAGs accounted for an average relative abundance of 19% in High O soils and 12% in O-horizon soils impacted by low severity wildfire ('Low O' soils)."

Supplementary Data 2 shows High Burn O horizon:

18.1 +/- 7.1 %; Low Burn O horizon: 8.0 +/- 6.5 %. Standard deviations are quite high suggesting that not each sample has these MAGs. Same set of eight MAGs has following levels in mineral soils: High Burn A horizon: 0.8 +/- 0.4 %; Low Burn A

horizon: 5.2 +/- 6.9 %. Comparing samples by testing in High Burn O vs A t- test(welch) = 4.23, p=0.051 n=6 and in O-Horizon High vs Low t-test(welch) = 3.97, p=0.142 n=6 both show no significance. Same goes for Low and A horizon comparisons. It would be interesting to take a look at transcription data, but this is not provided. But strong findings written about it bw L391-394. All together these results only show that the importance of these MAGs is yet to be clarified.

Mapping comment: In Supplementary Data 2, the MAGs initially added up to ~81-95% because this is relative abundance data from the reads that mapped to the MAGs. The summed value for each sample was <100% because we have discarded low quality MAGs. Note that now these values add to 1 (relative abundance of 100%; Supplementary Data 2, Sheet C) (1) because we have recalculated MAG relative abundance after discarding low-quality MAGs from the dataset. The percentage of reads that have been assembled and binned have been added to Supplementary Data 2 (Sheet A).

Chosen MAGs comment: We thank the reviewer for their thoroughness and agree that we should have accounted for standard deviation across the triplicates when choosing the MAGs of interest in High S and High D soils to profile in the text. To be more systematic in the choice of which MAGs to discuss, we chose to select MAGs that have an average relative abundance > 0.5% across the triplicate samples and a standard deviation that is less than the average relative abundance (methods L645). This method resulted in 34 MAGs of interest in High S and 11 MAGs in High D samples, which further reflects the homogenizing influence of wildfire on surface soils (i.e., more variation across samples in High D and higher standard deviation of MAG relative abundance resulted in more MAGs dropping out). Note that we chose to do this solely based on the metagenomic data and not metatranscriptomic, because we believe it better reflects the microbiome composition as mRNA is more transient than DNA. We use pairwise t-tests to identify the MAGs from these subsetted lists that are significantly enriched in high vs. low severity (S and D). Additionally, the metatranscriptomics data has now been provided in Supplementary Data 4.

3.4- Doubling time estimations and Fig 3B: Firstly, authors might be missing an opportunity here by just adhering to removing CUB with > 5hrs values. gRodon cannot accurately differentiate between long doubling times but 5 h is a pragmatic default, maybe not a biologically meaningful one. Infact Weissman et al. writes “5 h threshold suggests a natural definition of an oligotroph as an organism ...” They suggest another important filtering (not clear if it is applied in this MS) for data/result QC by making sure that each MAG has >10 ribosomal proteins. Secondly, analysis shown in here is not necessarily informative. Figure

41itself show it that many (predicted) fast and slow growers are very low (bw 0 to < 0.5%) in abundance. Even looking MAGs with relative abundance > 0.5% a good number still falls above the calculated raw average. It would be useful to have a correlation analysis with appropriate statistics showing that predicted fast growth is indeed can be associated with high MAG abundance (or is the shaded area in Fig3B showing the boundaries of significance?).

We have removed MAGs from this analysis that had <10 ribosomal proteins (as annotated by DRAM). This QC step removed 14 MAGs, resulting in 374 MAGs with <5 hours growth rate used for the correlation analysis. We thank the reviewer for noting that we needed statistics to bolster our conclusions and have added the results from a Spearman's correlation analysis onto the new Figure 2. These results agree with our previous conclusion that surficial soils impacted by high severity wildfire potentially favor MAGs with lower maximum doubling times (significant negative correlation between MAG maximum

doubling time and average MAG relative abundance in High O), whereas these trends are not present in the other three conditions. For the purposes of this correlation analysis, we think it is important to remove MAGs with >5 hours growth rate but agree with the reviewer that it is interesting to note that Weissman states that the threshold “suggests a natural definition of an oligotroph as an organism for which selection for rapid maximal growth is low enough so that no signal of growth optimization is observed.”

3.5- “Soil chemistry” section only covers Nitrogen species so better titled as what it is. Soil moisture and pH are very important parameters but either not measured or reported. Soil pH increases after fire and shown to impact microbial communities. Soil moisture is not reported (but recorded in MS as measured). Nitrogen availability within a few years after fire is generally high mainly due to fire-induced mineralization. Testing Control vs Burn Severity in mineral A horizons result reported (Supp Data1) only ammonia concentrations in High and Moderate burn sites seem to be statistically different [t-test(welch)=-2.84, p=0.035, n=11 Ryan-Mod; t-test(welch)=- 3.11, p=0.023 n=11 Ryan-High] than the control sites with no significant changes in nitrate concentrations. As no Control samples were sequenced it is unclear that which ammonia oxidizing bacteria and archaea reside in these soils. Study is likely having a sample sequence coverage issue where the sequencing effort is not sufficient to cover all high/low abundant & rare populations, including those carrying amoA. Other potential N-related processes such as use of urea or protein degradation is not covered so source of N is unknown. This is critical to the arguments of the paper since fire is argued to open up niches for fast growing organisms which require high C and N. In theory, increases in inorganic N in post-fire soils should be favorable to copiotrophic microorganisms but potential copiotrophs are assumed to be in O-horizon not in A-horizon. All together this is a mix of results that require better framing and rethinking. In parallel, statements on PyC lability can use better integration with MAG and transcriptomic data. It is more thermodynamically favorable to oxidize compounds with a high NOSC in reducing (anoxic) environments. More oxidized DOM, in having been broken down by aerobic microbes should exhibit lower

aromaticity (and maybe have a lower N content). That's show in low-NOSC index and high catA activity in Low severity burn site relative to High burn severity. There is no info (or evidence) for presence of anoxic conditions in high burn severity site, NOSC index is high (L518 is not particularly useful here) and overall transcriptome activity on aromatic compounds is low (relative to low burn site). Looks like fire created substrates are yet to spark microbial activity in high burn severity sites. It is too early to make statements like L705 & L715.

We have responded to these comments in a point-by-point manner below:

Soil chemistry: We've added in more of the soil chemistry data (pH, DIN, DON, DOC, %C, %N, soil moisture), which was originally kept out because it was only analyzed for subsets of samples for other projects by collaborators. These details were added to the methods (L512) and Supplementary Note 1 was added to explain some of the results. This was left out of the main text because these analyses were only completed on subsets of samples and we don't believe they add significantly to the story and soil chemistry changes with wildfire burn severity are not novel.

Inorganic N cycling and sequencing depth: To address this comment and do a more thorough analysis of inorganic N cycling activity in our samples, we annotated the metagenome assemblies and mapped our metatranscriptomic reads to them to identify any inorganic N transcripts that were not binned and therefore absent from the genome-resolved analysis. Using this approach, we detected *amoA* transcripts in deeper soils but an absence of them in surface soils. We infer that taxa that encode the potential for nitrification lack complementary traits that facilitate their persistence one year post-fire. In contrast, as evidenced by our marker gene data, the deeper soils experience a lesser impact of fire potentially allowing these taxa to persist (L424-429).

Organic N (protein degradation): We looked at organic N (e.g., protein) degradation by analyzing the expression of peptidase genes and found differential expression of 41 peptidase genes in High S vs. Low S, expression of 2721 total peptidase genes by the featured 40 High O MAGs (L202), and the expression of peptidases by both fungal MAGs (L408). This led to our hypothesis that necromass-derived organic C and N is favoring copiotrophic activity in burned organic soils, but that inorganic N cycling is limited to deep soils at one year post-fire (see new discussion regarding this starting on L420).

NOSC of DOM: Based on our interpretations of LaRowe & Van Cappellen (2011)⁷, compounds with higher NOSOC values will yield the greatest overall delta G under aerobic conditions (due to low delta G of C oxidation coupled with the energy available reducing O₂). We agree that under anaerobic conditions, only compounds with a high NOSOC value will yield a sufficiently high overall delta G to power anaerobic heterotrophic respiration, maintaining a F_T close to 1⁷. Thus, given the inferred oxic conditions throughout these soils, we infer that DOM pools with higher NOSOC values will be more thermodynamically favorable for heterotrophic respiration as compared to lower NOSOC pools.

PyC utilization: Regarding the aromaticity of DOM across the burn severity gradient, we note that microbial pathways for the degradation of aromatic intermediate products (catechol, protocatechuate) are both present and active in burned soils. Most genes within these pathways are not differentially expressed between surface and deep soils. For example, *catA* expression is not significantly higher in low vs. high severity-impacted soils (Welch's t-test; $p=0.3791$). Therefore, we conclude that while pyDOM is being acted upon by microbes via transcript data, the bulk soil geochemical signal of increased DOM aromaticity is still present one year post-fire.

General Comments:

3.6 - Please provide a table describing the results of metagenome and metatranscriptome sequencing, assembly processes. It would be advisable to use a metric such as Nonpareil to determine the coverage and estimated number of species in these samples to show that indeed this MAG dataset corresponds to 81- 95% of the population.

We've added important metagenomic and metatranscriptomic sequencing information to Supplementary Data 2 in Sheet A (e.g., metagenomic and metatranscriptomic sequencing depth, % reads assembled and binned in final dereplicated MAG dataset). Note that we did not intend to imply that the MAG dataset corresponds to 81-95% of the population as this was relative abundance data from read mapping (see response to comment 3.3).

3.7 - Reporting statistics: When calling significant one must report the test and p- value

We went through the manuscript and ensured all uses of the word 'significant' were accompanied with the statistical test and p-value and added in this information when needed.

3.8 - MS uses active vs abundant interchangeably which is difficult to decipher from the writing at times. It would be better if activity can be strictly designated to metatranscriptomes.

We have gone through the manuscript and made sure that the term 'active' is only used when referencing the metatranscriptomic data.

3.9 - Author's should consider moving FTIR-MS analysis to earlier to describe the changes in DOM and measured N-species together.

Although we understand the merit in grouping the DOM analysis with discussing of measured N-species, we have decided to keep it in the pyOM section (beginning at L241) because we believe the FTICR-MS dataset provides justification for why we are looking at the catechol and protocatechuate degradation pathways.

3.10 - Description of post fire soil conditions is limited. MS would benefit from explaining difference in high -to-low severities shown in Fig.S1 and some description of surviving plant cover at the time of sampling. Likewise, climate and soil conditions at the time of sampling warrant description.

We've added in site climate information (L480) and more details about conditions during the time of sampling (L497).

Point-by-point Comments:

3.11 - L75-76: Statement suggesting the studies focused on post-fire soil microbiome is mostly descriptive is not accurate – there are some interesting ecological theory applications taking advantage of fire disturbance. Here are few examples:

Zhang L, Ma B, Tang C, Yu H, Lv X, Rodrigues JL, Dahlgren RA, Xu J. Habitat heterogeneity induced by pyrogenic organic matter in wildfire-perturbed soils mediates bacterial community assembly processes. *The ISME Journal*. 2021 Jan 29:1-3.

Yang S, Zheng Q, Yang Y, Yuan M, Ma X, Chiariello NR, Docherty KM, Field CB, Gutknecht JL, Hungate BA, Niboyet A. Fire affects the taxonomic and functional composition of soil microbial communities, with cascading effects on grassland ecosystem functioning. *Global change biology*. 2020 Feb;26(2):431-42.

Knelman JE, Schmidt SK, Garayburu-Caruso V, Kumar S, Graham EB. Multiple, compounding disturbances in a forest ecosystem: fire increases susceptibility of soil edaphic properties, bacterial community structure, and function to change with extreme precipitation event. *Soil Systems*. 2019 Jun;3(2):40.

We have re-framed the introduction and have added in new references to contextualize our study and acknowledge work that has already been done on the post-fire soil microbiome.

3.12 - L104: Nitrogen species are not reported for mineral soils in Supp1 as stated

This must be mistyped because there is no mention of N species data in this line, but we have ensured that the chemistry data is presented in Supplementary Data 1 as discussed in the methods.

3.13 - L167: Parameters used for analysis should be included as M&M or SI which ever is appropriate

We've added more information on QIIME analysis into the methods (L592-599).

3.14 - L191: Author's write "16S rRNA gene data confirmed that the soil microbial communities were not significantly different." This is very unexpected as sites that are apart tends to look significantly different. Per Fig S1 distance between sites are more than 20 km.

See response to comment 3.1.

3.15 - L221: How eukaryotic MAGs were identified is not described. M&M doesn't outline if any effort was put into binning and gives the impression that contigs identified as eukaryotic was treated as a MAG if annotated to the same species. This is just an impression, as it is not described.

We agree that we need to address how fungal MAGs were identified in the methods and have added extensive details (L651-663) and a figure (Figure S2) to address this gap in the manuscript.

3.16 - L226-228: Author's write "A metabolic pathway within a MAG is considered complete if it is $\geq 50\%$ complete because MAGs are commonly not 100% complete." There are many reasons why a MAG is not complete: reading depth, sample GC distribution, performance of the assembly and binning process, ability to annotate hypothetical pathways. This statement is out of bounds.

This statement was removed to eliminate confusion.

3.17 - L234: Please provide details on how ribosomal RNA was removed from total RNA. No kit or methodology is cited.

The Takara SMARTer Stranded Total RNA-Seq Kit v2 was used both to remove ribosomal RNA and construct the sequencing libraries. The sentence was edited for this clarification

(L678).

3.18 - L374: Fig 1 and Figure S4 are drawn at phylum scale. It is not possible to one on one compare, please rephrase the sentence

We have rephrased the sentence to say that we were able to generate MAGs of taxa shown to be important in the complementary 16S rRNA gene sequencing data (L144).

3.19 - L382-386: Authors write: “MAGs affiliated with the Actinobacteria genera Blastococcus, Mycetocola, SISG01, SCTD01, Nocardiodes, and Arthrobacter were all enriched (relative to control soils) in surface organic soil horizons (O-horizon) impacted by high severity wildfire (hereafter referred to as ‘High O’) that in most instances had been combusted to an ash layer.” But per M&M they did not analyze a control sample via metagenome sequencing. Please revise the statement.

This statement initially meant to say that these genera were enriched in surface samples impacted by high severity wildfire in the 16S rRNA gene sequencing data. We agree with the reviewer that this was confusing to the reader and have made sure that statements like

this are revised throughout the metagenomic sections of the manuscript. This specific paragraph has been greatly revised in the edited manuscript with new analyses, as well (L155-177).

3.20 - L387: Here please refer to figure showing the enrichment of Actinobacteria MAGs (MAGs RYN_93, RYN_94, RYN_101, RYN_124, RYN_147, RYN_169, RYN_175, RYN_216)

Along with the new list of MAGs discussed in this section (see previous comment for details), we have added a figure to the supplementary material (Figure S7) that shows the relative abundance of the MAGs (7 in High O and 2 in High A) that were significantly enriched between burn severities for a given sample depth. The standard deviation of the average relative abundances of each condition is also shown, and this figure is pointed to in the main text in L174.

3.21 - L398: Spore forming is a common Actinobacterial trait that can be found with or without fire. Similarly, heat shock proteins (L401) do exist in almost all genomes. Fig.3A should include standard deviations and appropriate statistics to show the importance of these genes post-fire.

We agree with the reviewer that these statistics are important. Although this is no longer part of the figure, we have added whether the changes are significant into the text (L180-183).

3.22 - L434-438: Another transcriptome statement with no figure or table to support it

This data was added to Supplementary Data 4 on sheet G.

3.23 - L469: This data needs to better representation. RYN_342 has an abundance of

1.16 +/- 1.32 % so it is abundant in one sample, may not be representative of any other.

We agree with the reviewer and have altered how we selected MAGs of interest to better represent the data, as discussed previously. We chose MAGs that had an average relative abundance across the triplicates >0.5% and a standard deviation that is < the average relative abundance (see L645 and review comment 3.3).

3.24 - L617: VirHostMatcher is not mentioned in M&M or SI M&M

VirHostMatcher is in the methods section in L725-727.

3.25 - L685: Please edit for meaning “...in soil mineral soil ammonium..”

This wording was fixed for clarification.

3.26 - L696: Figure S6: Higher activity of CAZY genes in A-horizon than O-horizon is surprising as they are indicative of low C availability & investment in C acquisition. So in the balance of simple(r) C substrates this figure shows that O-horizon is less C limited post-fire than A-horizon it does not “... suggesting that certain C cycling processes may be absent in High O samples.”

This figure was remade with the new groups of MAGs of interest in both surficial and deeper soils impacted by high severity wildfire and the results were not as compelling, this combined with the confusion here resulted in us removing this statement from the manuscript.

3.27 - L721:“... offering opportunities to leverage these results for more effective management of other wildfire-disturbed environments.” authors should state what do they mean by leveraging these results as this is

We agree that this is a vague statement and have removed any statement regarding using the dataset presented here to inform restoration efforts due to this and comments by another reviewer (Comment 1.30).

3.28 - Figure1: no standard deviations or statistics presented

Asterisks denoting significance (pairwise t-test; $p < 0.05$) were added for each change in relative abundance that was significant. Standard deviations were calculated but are unable to be shown on the figure because there is large variability due to the nature of this type of data (many ASVs and differences in how different genera or species are influenced by burn). We have decided not to include a table of the standard deviations because of this and do not think it adds any helpful information.

3.29 - Figure S7: Non Actinobacterial MAGs are circled as MAGs of interest but there is no explanation to why

This figure was removed because the information initially portrayed in the figure is now shown in the new Figure 2.

3.30 - Figure S8: What does the asterisk denote, significance?

The asterisk denotes a gene that is highly expressed in the given condition (cobA/cobO in High A, here). The asterisk was made larger in the legend so that its meaning is clearer.

3.31 - Supp Data 2: RYN_1 has no tax assignment – is this even a Bacteria?

RYN_1 was not classified by GTDB-TK so we inferred the taxonomy assignment by blasting ribosomal protein S2 (NCBI blastn) and found the best match to be *Myxococcus xanthus* (percent identity 79.88%). Taxonomy was only inferred to class because there was only a 79.8% percent match. This information was added to Supplementary data 3.

Reviewer #4 (Remarks to the Author):

This study is conceptually straightforward, but the analyses that went into this study are comprehensive. Essentially, the authors wanted to determine how high intensity wildfires influence the microbial communities found in soil post-fire. While there have been many studies investigating wildfire effects on soil microbial communities and microbial processes, this study is unique in that it generated and analyzed metagenome-assembled genomes that were integrated with metabolomic, metatranscriptomic, and marker gene sequencing

53data. While the techniques used here are clearly 'cutting edge', I do think the authors oversell their work a bit and some of the conclusions are not well-supported by the data presented. The authors could benefit by being more cautious in the presentation of their results and acknowledge some of the uncertainties associated with the chosen study design and the limitations thereof. More detailed comments follow below:

Thank you to the reviewer for their constructive comments. To address the comment regarding overselling the dataset, we've tried to limit speculative language where appropriate but note that metagenomics and metatranscriptomics data is speculative in nature so use this language (e.g., 'likely', 'suggests'; see comment 2.15) to not oversell the data to readers. We've also added in language clarifying that we are not trying to conclude

that the thermal resistance or stress response traits found in MAGs are caused by fire, but more so that the combination of these traits may allow the bacteria represented by these MAGs to be dominant in soil one year post-fire (comment 4.4). Additionally, to align our discussion to the caveats of the dataset (only -omics on low and high severity samples, one year post-fire), we've constrained our language to remove any mention of 'fire-responding' or that fire 'caused' a given response. See below for responses to individual comments:

4.1 - In the Abstract and Introduction, it is frequently mentioned that 'little is known regarding the the impact of high severity fire on microbially-mediated processes' (or equivalent). This is not true – there are many studies looking at how soil trace gas emissions and N dynamics (for example) are affected by wildfire. Perhaps more importantly, this study did not include any information on microbially-mediated process rates (obviously genes and transcripts do not equate with process rates) so it is a bit disingenuous to frame the Introduction in this manner.

We agree with the reviewer's comment and other similar comments by Reviewer 1 and 3 (comment 3.11). We have re-framed the abstract and introduction to better contextualize the work and acknowledge the previous work done on the post-fire soil microbiome.

4.2 - It is a bit unclear to me why the authors combined results from the two sites (Ryan and Badger Creek) for many of the analyses, especially since the ANOSIM analyses show that the soil microbial communities are significantly different (lines 201-204). This would seem like a small detail, but combining results from the 2 sites is likely to influence the patterns observed. For example, the differences in beta dispersion evident in Figure 2 could simply be a product of site-level variability in community composition (particularly across the unburned samples).

See comment 3.1 for the addressing of this comment.

4.3 - There is clearly a lot of data that went into this study, but I was surprised there was no data on soil bacterial/fungal biomass as that would have allowed the authors to determine if the 'fire responding' taxa are actually growing after the fire event, or are merely increasing in relative abundance because 'fire sensitive' taxa were removed by the fire event. Even qPCR-based estimates of

55

total bacterial/fungal DNA amounts would have been useful to include. Any conclusions about ‘fire responders’ being able to grow quickly would need to be supported by some sort of actual biomass estimates.

Although we agree that fungal and bacterial biomass data would have been helpful to include in regards to our discussion of post-fire growth rates, it was not included here because the impact of wildfire on soil microbial biomass is already well-documented in the literature (see Pressler et al., 2019; Meta-analysis⁸) and we instead focused our efforts on collecting and analyzing more novel aspects of our dataset.

4.4 - One of their main conclusions is that taxa found in soils post-fire tend to have more genes for thermal resistance (like heat shock proteins). However, such genes are likely common across a broad diversity of taxa. Likewise, it seems like a bit of a stretch to claim that ‘higher GC content may be another heat resistance trait’ when many of the ‘fire responsive’ taxa were Actinobacteria which are known to have high GC content. More generally, identifying the key traits of the ‘fire responsive’ MAGs would require a comparison against MAGs from ‘non-fire responders’ and it is not clear if this was done (maybe I missed this important information).

We agree that the unfortunate lack of metagenomic and metatranscriptomic sequencing from control samples is a limitation in our study and conclusions. We try to make up for this by using 16S data to (1) inform which MAGs we study and (2) show that the majority of MAGs we focus on are from taxonomies that notably increase or decrease in relative abundance in the marker gene data (e.g., Supplementary Note 3 and L433-448). Additionally, we note that some of the taxa (e.g., *Streptomyces*) that commonly have these traits (sporulation, stress response) are enriched in both the metagenome read mapping and marker gene data (Supplementary Note 2). Further, we believe that it is convincing that both heat shock and sporulation genes significantly increase in coverage from Low S to High S samples (L180-183). Lastly, we try to make it clear in the language that we are not claiming these traits to be unique to pyrophilous taxa or MAGs, but that these traits potentially enable the success of these taxa during and/or after wildfire and added more language into the text to clarify this (L193-196).

4.5 - Is it reasonable to claim that MAG abundance-normalized transcripts are a measure of growth rate? (lines 432-440).

We didn’t directly use MAG abundance-normalized transcript expression to quantify growth rate, we used it as a general proxy to compare MAGs and identify whether the MAGs with

56

the highest maximum growth rate (via codon usage bias and gRodon). To do so, we only used transcripts for ribosomal proteins or central metabolism pathways (e.g., glycolysis) because they have been found to be highly expressed in fast growing bacteria⁹ and predictive of cell growth¹⁰. Although we know that it's not a perfect proxy of growth rate, we think it is an interesting addition to the discussion that the MAGs that have the highest maximum growth rate do not also have the highest abundance-normalized expression of genes related to growth. References were added to the text to explain this method to the reader.

4.6 - The discussion of cobalamin production seems like a bit of a stretch (lines 489-508) – just because the genes/transcripts are there, does not mean cobalamin production is happening at higher rates in fire-affected soils (I know the authors know this, but some caveats would be useful here).

We agree that the presence of genes or transcripts does not necessarily indicate the increased production of cobalamin, but think it is noteworthy that production of cobalamin in soil is taxonomically constrained³ and that one taxa known to contribute (*Streptomyces*) is enriched in High A soils (in both 16S and metagenome data), with a corresponding increase in the expression and the differential expression of these genes (see comment 2.18). Additionally, this section has been moved to the SOM as we don't think it's one of the key findings of the manuscript.

Decision Letter, second revision:

Dear Mike,

This is {redacted} I hope all is well with you. I have taken over as handling editor of your manuscript, "A genome resolved view of wildfire legacy impacts on the soil microbiome". Thanks for your patience while we waited for the referees to turn in their reports! Your paper has now been seen by the 4 original referees, whose expertise and comments you will find at the of this email. The referees feel that the majority of their comments have been addressed and are in general satisfied with the revisions. As you will see, Reviewer #3 has some remaining concerns about reporting of the statistics and the NMDS, with some additional requests for soil chemistry and PyOM to be included more prominently in the main text. We would like to consider your response to these concerns in the form of a revised manuscript before we make a final decision on publication, however I do want to stress that no further peer review will be needed, as we feel that we'll be able to assess these changes on our end when you resubmit, without going back to Reviewer #3 for additional comment.

Please don't hesitate to let me know if you have any questions or concerns moving forward.

57We are committed to providing a fair and constructive peer-review process. Do not hesitate to contact us if there are specific requests from the reviewers that you believe are technically impossible or unlikely to yield a meaningful outcome.

If you have not done so already please begin to revise your manuscript so that it conforms to our Article format instructions at <http://www.nature.com/nmicrobiol/info/final-submission/>

The usual length limit for a Nature Microbiology Article is six display items (figures or tables) and 3,000 words. We have some flexibility, and can allow a revised manuscript at 3,500 words, but please consider this a firm upper limit. There is a trade-off of ~250 words per display item, so if you need more space, you could move a Figure or Table to Supplementary Information.

Some reduction could be achieved by focusing any introductory material and moving it to the start of your opening 'bold' paragraph, whose function is to outline the background to your work, describe in a sentence your new observations, and explain your main conclusions. The discussion should also be limited. Methods should be described in a separate section following the discussion, we do not place a word limit on Methods.

Nature Microbiology titles should give a sense of the main new findings of a manuscript, and should not contain punctuation. Please keep in mind that we strongly discourage active verbs in titles, and that they should ideally fit within 90 characters each (including spaces).

Please include a data availability statement as a separate section after Methods but before references, under the heading "Data Availability". This section should inform readers about the availability of the data used to support the conclusions of your study. This information includes accession codes to public repositories (data banks for protein, DNA or RNA sequences, microarray, proteomics data etc...), references to source data published alongside the paper, unique identifiers such as URLs to data repository entries, or data set DOIs, and any other statement about data availability. At a minimum, you should include the following statement: "The data that support the findings of this study are available from the corresponding author upon request", mentioning any restrictions on availability. If DOIs are provided, we also strongly encourage including these in the Reference list (authors, title, publisher (repository name), identifier, year). For more guidance on how to write this section please see: <http://www.nature.com/authors/policies/data/data-availability-statements-data-citations.pdf>

To improve the accessibility of your paper to readers from other research areas, please pay particular attention to the wording of the paper's opening bold paragraph, which serves both as an introduction

and as a brief, non-technical summary in about 150 words. If, however, you require one or two extra sentences to explain your work clearly, please include them even if the paragraph is over-length as a result. The opening paragraph should not contain references. Because scientists from other sub-disciplines will be interested in your results and their implications, it is important to explain essential but specialised terms concisely. We suggest you show your summary paragraph to colleagues in other fields to uncover any problematic concepts.

If your paper is accepted for publication, we will edit your display items electronically so they conform to our house style and will reproduce clearly in print. If necessary, we will re-size figures to fit single or double column width. If your figures contain several parts, the parts should form a neat rectangle when assembled. Choosing the right electronic format at this stage will speed up the processing of your paper and give the best possible results in print. We would like the figures to be supplied as vector files - EPS, PDF, AI or postscript (PS) file formats (not raster or bitmap files), preferably generated with vector-graphics software (Adobe Illustrator for example). Please try to ensure that all figures are non-flattened and fully editable. All images should be at least 300 dpi resolution (when figures are scaled to approximately the size that they are to be printed at) and in RGB colour format. Please do not submit Jpeg or flattened TIFF files. Please see also 'Guidelines for Electronic Submission of Figures' at the end of this letter for further detail.

Figure legends must provide a brief description of the figure and the symbols used, within 350 words, including definitions of any error bars employed in the figures.

Please include a statement before the acknowledgements naming the author to whom correspondence and requests for materials should be addressed.

Finally, we require authors to include a statement of their individual contributions to the paper -- such as experimental work, project planning, data analysis, etc. -- immediately after the acknowledgements. The statement should be short, and refer to authors by their initials. For details please see the Authorship section of our joint Editorial policies at http://www.nature.com/authors/editorial_policies/authorship.html

* include a point-by-point response to any editorial suggestions and to our referees. Please include your response to the editorial suggestions in your cover letter, and please upload your response to the referees as a separate document.

* ensure it complies with our format requirements for Letters as set out in our guide to authors at www.nature.com/nmicrobiol/info/gta/

* state in a cover note the length of the text, methods and legends; the number of references; number and estimated final size of figures and tables

* resubmit electronically if possible using the link below to access your home page:

{redacted}

*This url links to your confidential homepage and associated information about manuscripts you may have submitted or be reviewing for us. If you wish to forward this e-mail to co-authors, please delete this link to your homepage first.

Please ensure that all correspondence is marked with your Nature Microbiology reference number in the subject line.

Nature Microbiology is committed to improving transparency in authorship. As part of our efforts in this direction, we are now requesting that all authors identified as 'corresponding author' on published papers create and link their Open Researcher and Contributor Identifier (ORCID) with their account on the Manuscript Tracking System (MTS), prior to acceptance. This applies to primary research papers only. ORCID helps the scientific community achieve unambiguous attribution of all scholarly contributions. You can create and link your ORCID from the home page of the MTS by clicking on 'Modify my Springer Nature account'. For more information please visit www.springernature.com/orcid.

We hope to receive your revised paper within three weeks. If you cannot send it within this time, please let us know.

Yours sincerely,

{redacted}

Reviewers Comments:

60Reviewer #1 (Remarks to the Author):

The authors have addressed my comments on their earlier version. I appreciate their detailed revisions and look forward to seeing the paper in print. Minor comments:

L99-100: I suggest "from burn severity gradients" as there were four levels of burn severity within the gradient of burn severity at each site (not four gradients).

L282: the assumption that PyC is "uniformly resistant to decay" is out-dated. Revise to something like "considered to be largely resistant to decay." It is also well-established that PyC is not uniform.

Reviewer #2 (Remarks to the Author):

I commend the authors for their careful revision of their manuscript. I have no further comments or suggestions.

Reviewer #3 (Remarks to the Author):

Author's provide thoughtful and detailed responses to the points raised, many thanks for their efforts. Nelson et al. studies the impact of wildfire burn severity gradients in soil microbiomes of coniferous forests in western USA. Author's use a wide range of analysis tools (sequencing amplicon, DNA & RNA), FTIR-MS and aim to provide a multi-faceted description of microbial (bacterial, fungal, and viral) potential, activity at sampling time (apprx. one year after fire). Reading through several times, manuscript continue to reiterate on existing knowledge and does not necessarily explain it further & the analysis presented in the revised version doesn't improve upon the first draft. Authors seek to answer two hypotheses here: 1) higher severity wildfire would result in an increasingly altered soil microbiome and 2) taxa colonizing burned soils would encode functional traits that favor their persistence (e.g., the capacity to utilize fire altered substrates and rapidly recolonize empty soil niches). With the wealth of data available we get limited information on both. As also remarked in the earlier review, distinguishing the novelty of findings is important as well as providing new insights is necessary. Ideas provided for potential impact of necromass are interesting, but there is no support data to point the reader to potential avenues. "Ecosystem implications" section is in part improved with added analysis but doesn't provide a new info and largely confirmatory.

Author's responded to the site differences comment by "16S rRNA data different between Badger Creek and Ryan Fire - Unfortunately here the ANOSIM stat result was initially misinterpreted and incorrectly not changed in the text." Glad this cleared up however Figure 2 from earlier submission and Figure 1 from revision are exactly the same - hence authors are still not taking site differences into account while doing their amplicon-based statistics. Site differences are nested within and likely (till shown otherwise) impacting conclusions. Different soils can respond to fire severity at a different magnitude, there is enough replication in amplicon work to sort this out.

61Author's responded by "NMDS response - As the reviewer mentioned, there are many studies that use NMDS to visualize trends in their marker gene datasets with stress values > 0.1 in high-impact journals..." this remains a misinterpretation of the results. Stress values 0.1-0.2 which are considered fair because of the lack of good fit of the data to the assumptions of the analysis. It would be good to not to lay very strong conclusions based on them. Biological data is noisy and given the large variation observed in control samples and embedded site related differences NMDS struggle to fit this data. If PCoA would be applied, for example, authors would see how much (by percentage) of their observed variation can be attributed to fire events. Looking at the clustering (and strong mix of direct & indirect environmental filters) probably not much of the variation can be explained by fire event. Also, why not report on correlation with soil chemistry (now that the data is included as well)? Looking at the figures community structure between different fire intensities are not really decoupled from each other & there is post fire decrease in diversity at all stages – which is observed. It remains a concern that amplicon data has not been analyze to its potential. Authors hint on potential ecological events on selection – which can be model from amplicon data (earliest example by Ferrenberg et al 2013 ISME J)- but this is not done.

Growth rate and fire intensity (L208-217): Thanks for the detailed explanation for the processing and data QC. Author's write "there was a significant negative correlation (Spearman's $r = -0.18$, $p < 0.05$) between MAG relative abundance in High S.." which is largely driven by 3 abundant Actinobacteria of lower than average growth rate. Author's interpret this as "indicating that High S conditions may select for microorganisms that are able to grow quickly". Fast growers are clearly not abundant so is the meaning that these populations do grow fast but does not become abundant? A recent emerging hypothesis is that microbial recovery is not dominated by fast-growing populations but by deterministic selection in the stressful & C-limited post-fire soils. Here authors write only 22/40 of the MAGs (taking advantage of post-fire conditions), so fast growth is not the only winning strategy. By using differentially abundant MAGs having 90% of the total gene expression, these MAGS (slower and fast growing alike) have key info on post-fire life of bacteria. By the results presented here, statement of "...bacteria sampled one-year post-wildfire likely occupied niches in the immediate aftermath of wildfire through rapid cell growth." is out of bounds.

PyOM degradation: Approach here remains one of the examples of limited novelty provided while having a good set for mining new information. While it is reasonable to focus on aerobic central pathways of hydrocarbon degradation, it is by definition is expected to be present and active. Pyronema, a common pyrophilic fungus, shown to degrade PyOM through catechol and quinate. By literature aromatic catabolism should be prevalent in these soils, it would be good to present what is present in MAGs and active. Figure S15 has a lot of interesting information about gene transcripts to presented in very broad categories, but this figure is now very buried into the SI.

Representation of this large data set needs legibility. Calculations of percent abundances and coverages remains difficult to follow. Some examples: amoA analysis was done in metagenome assemblies (not sure what is the min contig length), assemblies are reported as R85-R118, reader needs to figure out which sample is which condition. Another; L157 writes "Combined, these MAGs accounted for an average relative abundance of ~60% in High S samples and ~34% in surface soils impacted by low severity wildfire ('Low S' soils..." supplementary data provided reports 15-36% mapping of reads to

62MAGs (SheetA). Looking at this data MAG distribution is skewed towards samples 32-95 MAGs per sample, not sure at this stage what number to consider as it is not quite clear if the samples were cross compared.

Point-by-point Comments:

L1: "Legacy" suggests long term consequence by definition, this study looks into immediate direct effects of fires

L4: Some authors are removed but didn't see explanation

L59: Statement too general and conflict with edits suggested by all reviewers

L77: Dove et al 2022 ISME J can be cited, this paper covers a different time scale of recovery than presented here via gene and genome centric approaches

L83: This is a very ambitious and unfortunately unfounded statement. Can single time point multi-omics work bridge all-these efforts at a field scale?

L110: Deep has different meanings to different readers, it would be good to state the depth here so meaning is clear - it is done in figure legends which is helpful

L113: difficult to decipher pairwise testing here: which pair are tested with the color scheme is not intermediately clear also there is no control for multiple testing so instead of this Tukeys HSD or permanova would be more straight forward to interpret

L115: Fig. S5 shows that in surface soils bacterial populations at moderate and high fire are different than the controls; whereas fungal communities are different than the controls in all fire severities. This figure shows that in surface soils bacteria mostly impacted by higher fire severities and fungi is always impacted. At 5-10 cm bacteria is impacted across the severity gradient and fungi is not. Text written here is not reflecting the results displayed.

L116: as the Table S2 refers to Characteristics of WGCNA networks not distance centroid calculations this section warrants a further explanation

L124: consistent increases in community dissimilarity is not a muted response, essentially might be priming + growth response

L142: Bioproject ID maybe move it to data availability or M&M

L158: Authors should explicitly explain (as they did in their responses) what these percentages mean.

L186: "futured MAGs" is confusing

L220: This statement is out of bounds, abundant bacteria is not really the fast-growing ones

L227: kit used removes (most of the) ribosomes so how can they be reliably used to do analysis?

L278: it is very common for bacteria to perform most energetically favorable part of a pathway and not the rest (when conditions are called for). to assume complete degradation and hand of assume mixing and connectivity in soil - which is unknown

L337: methods used here is not specifically targeting viral populations. is has been shown that while valuable viral populations identified from shotgun metagenomes have their limitations

L415-416: these populations are shown to recover in time

Figure 3: there is X-Y plotting data in panel A that can be removed

Reviewer #4 (Remarks to the Author):

I appreciate that the authors carefully considered my comments and the comments from the other

63reviewers when revising the manuscript. I have read through the revised manuscript and I think it is now significantly improved.

Author Rebuttal, second revision:

Reviewer #1 (Remarks to the Author):

The authors have addressed my comments on their earlier version. I appreciate their detailed revisions and look forward to seeing the paper in print. Minor comments:

L99-100: I suggest “from burn severity gradients” as there were four levels of burn severity within the gradient of burn severity at each site (not four gradients).

This has been changed to ‘Near surface soils (0-5 cm depth) were collected approximately one-year post-fire from four burn severity gradient transects (including control, low, moderate, and high burn severity) at two wildfires that occurred in 2018 along the Colorado-Wyoming border’ because there were four burn severity gradient transects and four burn conditions within each transect (L96-98).

L282: the assumption that PyC is “uniformly resistant to decay” is out-dated. Revise to something like “considered to be largely resistant to decay.” It is also well-established that PyC is not uniform.

This has been changed to clarify that we are not meaning that PyC is uniform in nature but that it’s generally considered to be resistant to decay (L262).

Reviewer #2 (Remarks to the Author):

I commend the authors for their careful revision of their manuscript. I have no further comments or suggestions.

Reviewer #3 (Remarks to the Author):

Author’s provide thoughtful and detailed responses to the points raised, many thanks for their efforts. Nelson et al. studies the impact of wildfire burn severity gradients in soil microbiomes of coniferous forests in western USA. Author’s use a wide range of analysis tools (sequencing amplicon, DNA & RNA), FTIR-MS and aim to provide a multi-faceted description of microbial (bacterial, fungal, and viral) potential, activity at sampling time (apprx. one year after fire). Reading through several times, manuscript continue to reiterate on existing knowledge and does not necessarily explain it further & the analysis presented in the revised version doesn’t improve upon the first draft. Authors seek to answer two hypotheses here: 1) higher

64severity wildfire would result in an increasingly altered soil microbiome and 2) taxa colonizing burned soils would encode functional traits that favor their persistence (e.g., the capacity to utilize fire altered substrates and rapidly recolonize empty soil niches). With the wealth of data available we get limited information on both. As also remarked in the earlier review, distinguishing the novelty of findings is important as well as providing new insights is necessary. Ideas provided for potential impact of necromass are interesting, but there is no support data to point the reader to potential avenues. “Ecosystem implications” section is in part improved with added analysis but doesn’t provide a new info and largely confirmatory.We fundamentally disagree with this reviewer and note that the three other reviewers see the novelty and importance of this work. Briefly in response, this manuscript is the first study to provide a comprehensive, integrated analysis of the post-fire bacterial, fungal, and viral components of the soil microbiome within the southern Rocky Mountains, an area experiencing increasingly severe and frequent wildfire events outside its historical wildfire regime. Using a suite of trans-disciplinary methods (amplicon sequencing, metagenomics, metatranscriptomics, FTICR-MS), we link post-fire soil community composition to function at the genome-level, detailing microbial traits that support growth in fire-impacted soils and report the likely ecosystem implications of these metabolisms. The importance and novelty of this manuscript goes further than the data and reported findings; we also provide an extensive, publicly available metagenomic and metatranscriptomic (coupled with amplicon and chemistry information) dataset available for other researchers to mine for different metabolisms that fell outside of the scope of this study. In contrast to the point made by the reviewer regarding the use of these data to support or investigate our hypotheses, we in fact *do* use our data throughout the manuscript to support the proposed hypotheses. The example mentioned by the reviewer is the utilization of necromass, which we investigate through the expression of peptidases (see L188-192). We are generally unclear why this reviewer is discounting these findings and the importance of this work in the field of soil microbiology.

Author's responded to the site differences comment by "16S rRNA data different between Badger Creek and Ryan Fire - Unfortunately here the ANOSIM stat result was initially misinterpreted and incorrectly not changed in the text." Glad this cleared up however Figure 2 from earlier submission and Figure 1 from revision are exactly the same – hence authors are still not taking site differences into account while doing their amplicon-based statistics. Site differences are nested within and likely (till shown otherwise) impacting conclusions. Different soils can respond to fire severity at a different magnitude, there is enough replication in amplicon work to sort this out. Author's responded by "NMDS response - As the reviewer mentioned, there are many studies that use NMDS to visualize trends in their marker gene datasets with stress values > 0.1 in high-impact journals..." this remains a misinterpretation of the results. Stress values 0.1-0.2 which are considered fair because of the lack of good fit of the data to the assumptions of the analysis. It would be good to not to lay very strong conclusions based on them. Biological data is noisy and given the large variation observed in control samples and embedded site related differences NMDS struggle to fit this data. If PCoA would be applied, for example, authors would see how much (by percentage) of their observed variation can be attributed to fire events. Looking at the clustering (and strong mix of direct & indirect environmental filters) probably not much of the variation can be explained by fire event. Also, why not

report on correlation with soil chemistry (now that the data is included as well)? Looking at the figures community structure between different fire intensities are not really decoupled from each other & there is post fire decrease in diversity at all stages – which is observed. It remains a concern that amplicon data has not been analyze to its potential. Authors hint on potential ecological events on selection – which can be model from amplicon data (earliest example by Ferrenberg et al 2013 ISME J)- but this is not done.

The focus of our study was using genome-resolved metagenomics and metatranscriptomics to provide important context to laboratory studies and complement existing compositional amplicon studies. Given the word and space constraints in this journal, and the novelty of

our genome and expression datasets, we minimized our analyses of the amplicon data and instead focused on the novel aspects of the omics datasets. We accept that this a broad paper with a very rich dataset and have tried to keep analyses and results as focused as possible, and believe that adding in more analyses (e.g., ecological modeling) would further dilute the impact and takeaway messages. We understand that there are more analyses that could be done on this dataset but note that the entire dataset is publicly accessible and available for any future researchers to mine, and we envision this data being used for future work. Although the stress of the NMDS is >0.1 , its purpose is to visualize the shifting communities with increasing burn severity (and differences between soil depths) and we argue that it does this well. We understand that there are site differences but the statistical analyses we provide are clear and solid (e.g., ANOSIM). We recognize that the sites had differing impacts but stress that this is not a crucial element to the broader story, as our multi-omics work is focused on the Ryan Fire sites. Lastly, we choose not to correlate the amplicon data with the soil chemistry because the chemistry analyses were done on subsets of samples that do not match across analytes so that we cannot apply them to the larger amplicon dataset (see previous response to reviewers document, comment 3.5).

Growth rate and fire intensity (L208-217): Thanks for the detailed explanation for the processing and data QC. Author's write "there was a significant negative correlation (Spearman's $r = -0.18$, $p < 0.05$) between MAG relative abundance in High S.." which is largely driven by 3 abundant Actinobacteria of lower than average growth rate. Author's interpret this as "indicating that High S conditions may select for microorganisms that are able to grow quickly". Fast growers are clearly not abundant so is the meaning that these populations do grow fast but does not become abundant? A recent emerging hypothesis is that microbial recovery is not dominated by fast-growing populations but by deterministic selection in the stressful & C-limited post-fire soils. Here authors write only 22/40 of the MAGs (taking advantage of post-fire conditions), so fast growth is not the only winning strategy. By using differentially abundant MAGs having 90% of the total gene expression, these MAGS (slower and fast growing alike) have key info on post-fire life of bacteria. By the results presented here, statement of "...bacteria sampled one-year post-wildfire likely occupied niches in the immediate aftermath of wildfire through various strategies likely including rapid cell growth." is out of bounds.

We believe the reviewer has misinterpreted Figure 2A; this analysis does show that there is a significant negative correlation between MAG relative abundance in High S samples and the estimated maximum growth rate. As the reviewer notes, there are three MAGs that appear to drive much of this significant correlation that are the most abundant in High S soils and have lower than average growth rates (i.e., they grow *faster* than average growth rates). We also highlight multiple *winning* strategies of MAGs in high severity-impacted soils

throughout the manuscript (stress tolerance, thermal resistance, PyC and necromass utilization), and have changed the text to '*These insights suggest that abundant bacteria sampled one-year post-wildfire likely occupied niches in the immediate aftermath of wildfire through various strategies likely including rapid cell growth*' to indicate that fast growth is likely not the only successful growth strategy (L203).

PyOM degradation: Approach here remains one of the examples of limited novelty provided while having a good set for mining new information. While it is reasonable to focus on aerobic central pathways of hydrocarbon degradation, it is by definition is expected to be present and active. Pyronema, a common pyrophilic fungus, shown

to degrade PyOM through catechol and quinate. By literature aromatic catabolism should be prevalent in these soils, it would be good to present what is present in MAGs and active. Figure S15 has a lot of interesting information about gene transcripts to presented in very broad categories, but this figure is now very buried into the SI.

We focused on the ortho- and meta-cleavage pathways of catechol and protocatechuate (27 total genes) because they are intermediate products of aerobic aromatic degradation¹, and we felt this was a comprehensive way to inventory the degradation of a large suite of diverse aromatics without looking at each one separately. We also note that we could have written an entire paper on pyrogenic C degradation but focused only on these pathways because they are the intermediate degradation products. Further, although there are laboratory studies demonstrating the ability to degrade aromatics by known pyrophilous taxa², this study bridges that work to field compositional studies by providing genomic evidence of these pathways in pyrophilous taxa from soil samples. We would expect these pathways to be present and active, but there is only one very recent genome-resolved study detecting the pathways in a field setting³. To complement the information on catechol and protocatechuate processing, we have provided some more aromatic degradation information, focusing on the potential for aerobic degradation of phenylacetate and benzoyl-CoA (L258, Figure S11). Finally, Figure S15 shows functional shifts across conditions and is in the SOM because of its broad nature.

Representation of this large data set needs legibility. Calculations of percent abundances and coverages remains difficult to follow. Some examples: *amoA* analysis was done in metagenome assemblies (not sure what is the min contig length), assemblies are reported as R85-R118, reader needs to figure out which sample is which condition. Another; L157 writes “Combined, these MAGs accounted for an average relative abundance of ~60% in High S samples and ~34% in surface soils impacted by low severity wildfire (‘Low S’ soils...” supplementary data provided reports 15-36% mapping of reads to MAGs (SheetA). Looking at this data MAG distribution is skewed towards samples 32-95 MAGs per sample, not sure at this stage what number to consider as it is not quite clear if the samples were cross compared.

Minimum contig length (2500bp) for the inorganic N cycling gene-resolved analysis was added to the methods section. We have edited Supplementary Data 2 so that the assemblies in SheetA are labeled with their corresponding condition (low/high severity S/D). Throughout the text, we’ve ensured that we do not use sample names that are meaningless or difficult for the reader to interpret sample conditions, and hopefully adding this piece of information to SheetA in the supplementary data sheet clears this further. Regarding the relative abundance comments, we understand that shotgun metagenomics and

metatranscriptomics methods do not capture the entire community and we clearly show in Supplementary Data 2 the percentage of reads that were captured in the assemblies (~70-88%) and MAGs (~15-39%). However, it is worth noting that regardless of the analysis technique – amplicon sequencing or metagenomic sequencing – one must consider that the whole community will not be 100% represented in the resulting data. This is an inherent caveat of these molecular microbiology analyses. Despite this, it is standard in the field to report the relative abundance of either ASVs or MAGs and using *relative* abundance explicitly implies that the whole community is not being captured. Regarding the last sentence ('Looking at this data MAG distribution...'), we are unclear what the reviewers concern with the data is here.

Point-by-point Comments:

L1: “Legacy” suggests long term consequence by definition, this study looks into immediate direct effects of fires

We chose to include the term ‘legacy’ because this study does not look at the immediate direct effects of fires but focuses on the effects one year following the disturbance. Given this time frame, we feel that the term ‘legacy’ is appropriate.

L4: Some authors are removed but didn’t see explanation

The two authors from UC Davis ({redacted}) were removed because, following their request and conversation with these individuals, together we decided that they did not contribute enough to the viral analyses to be included as authors on the manuscript. S.E.G., under mentorship by J.B.E., completed a first round of analyses on the viral data but these were redone by A.R.N. for the final analyses presented in the manuscript. Thus, they did not contribute to the analyses presented herein.

L59: Statement too general and conflict with edits suggested by all reviewers

We agree that this statement is very general and have amended to ‘*little is known about the impact of high severity fire on soil microbiome function in these high elevation, coniferous ecosystems*’ (L57-60). We also note here that in the last round of revision we’ve added in citations to all relevant literature to this research question throughout the rest of the introduction to acknowledge research already done on the post-fire soil microbiome. Though there are compositional and functional studies of how fire impacts the soil microbiome, there are very few that integrate the bacterial, fungal, and viral components of the microbiome, and that specifically focus on the impact of high severity fire on the coniferous ecosystems of the southern Rockies.

L77: Dove et al 2022 ISME J can be cited, this paper covers a different time scale of recovery than presented here via gene and genome centric approaches

This paper was not yet published with the last resubmission and has been added.

L83: This is a very ambitious and unfortunately unfounded statement. Can single time point multi-omics work bridge all-these efforts at a field scale?

Yes, our argument for our chosen methodological approach is that it connects laboratory studies on cultured isolates to compositional field studies by linking function to genomes. This study is also important because compositional or lab studies have shown that pyrophilous soil taxa are taxonomically constrained (e.g., actinobacteria *Blastococcus* and *Arthrobacter*), and we provide an extensive genome database that encompasses these pyrophilous taxa reported in compositional or laboratory studies. Although there is more to be done in this area (e.g., time series studies) and we don’t argue otherwise, we believe this

72is an important next step in the field of soil microbiology focusing on post-fire microbial ecology. We don't try to claim that this study answers all the remaining questions regarding post-fire soil microbiome function but strongly believe it begins to address the disconnect between laboratory and field compositional studies.

L110: Deep has different meanings to different readers, it would be good to state the depth here so meaning is clear - it is done in figure legends which is helpfulThis is a good point and we try to be clear on these depths throughout the text to not confuse readers. The depths that correspond to our 'shallow' and 'deep' labeling are now added here.

L113: difficult to decipher pairwise testing here: which pair are tested with the color scheme is not intermediately clear also there is no control for multiple testing so instead of this Tukeys HSD or permanova would be more straight forward to interpret
We are unclear but think this comment refers to the pairwise comparisons in Figure 1e and 1f to assess which burn severity conditions are significantly different in either bacterial or fungal diversity in shallow (Figure 1e) or deep (Figure 1f) soils. This is a simple boxplot and similar stats (pairwise comparisons of all conditions, pairwise t-tests) that we use throughout the manuscript on different figures (e.g., Figure 3b, Figure S5, Figure S8, Figure S9) and had no issues from other reviewers on these stats.

L115: Fig. S5 shows that in surface soils bacterial populations at moderate and high fire are different than the controls; whereas fungal communities are different than the controls in all fire severities. This figure shows that in surface soils bacteria mostly impacted by higher fire severities and fungi is (not?) always impacted. At 5-10 cm bacteria is impacted across the severity gradient and fungi is not. Text written here is not reflecting the results displayed.

Our wording here may have been unclear and we feel that the reviewer thought that this whole paragraph (L105-121) was about Figure S5, which it is not. We have tightened the wording in this paragraph so that we specify whether we are discussing the bacteria or fungi data (or both) and it is clear which data we are referencing. Additionally, beta dispersion does not look at whether communities are different but quantifies the homogeneity in group dispersion. We use this analysis to investigate whether microbial communities become more homogenous to one another with burn severity. We strongly believe the combined alpha and beta diversity analyses support our conclusion that *“Combined amplicon sequencing data analyses highlight the susceptibility of surface soils to wildfire, resulting in less diverse and inter-connected microbial communities. In contrast, the microbiome in deep soil displays a more muted response to wildfire, potentially due to insulation from soil heating”* (L117-121).

L116: as the Table S2 refers to Characteristics of WGCNA networks not distance centroid calculations this section warrants a further explanation

The text here was edited to clarify that the second half of the sentence is in relation to the WGCNA - *“Similarly, as fungal and bacterial diversity decreased with burn severity, beta dispersion (‘distance to centroid’) calculations revealed increasingly similar bacterial communities (Figure S5) with less complex community structures (via WGCNA;*

Supplementary Note 2, Table S2)" (L109-111).

L124: consistent increases in community dissimilarity is not a muted response, essentially might be priming + growth response

We came to this conclusion of the muted effects of wildfire on deep vs. shallow soils because there are clear alpha and beta diversity signals of fire in shallow soils that are absent in deeper soils. We are unclear exactly what this comment refers to here but retain that the increase in beta dispersion in the bacterial communities from control to burned soils (Figure S5) might be due to heterogeneous impacts of fire on deeper soils and think that L116 explains this hypothesis well.

L142: Bioproject ID maybe move it to data availability or M&M

The Bioproject ID has been removed here and is now only in the methods and data availability sections.

L158: Authors should explicitly explain (as they did in their responses) what these percentages mean.

Relative abundance is a common and standard way of expressing differences in microbial composition across conditions in the field of microbial ecology. The use of 'relative' abundance explicitly implies that the value is relative to what was captured in the data, not to the entire *in situ* community. We are clear in the supplementary data that overviews the assemblies that not all reads were captured in the assemblies or MAGs, and our values for % reads assembled and binned are similar to those in other soil microbiome studies. Further, we note here that the composition and relative abundance of the MAGs mirror trends we see in the 16S rRNA gene sequencing data so, although we are not capturing the entire community with our methods, the amplicon and shotgun metagenomics data mirrors one another. Thus, we do not think it is necessary to explicitly explain this in the text where we are already limited by the word count.

L186: "futured MAGs" is confusing

There is maybe a typo here - the text initially said '(including 5 of the featured MAGs)' but this sentence has now been changed to '*In 16 MAGs, thermal resistance was complimented by genes for mycothiol biosynthesis.*' after cutting text to meet the word count.

L220: This statement is out of bounds, abundant bacteria is not really the fast-growing ones

See previous growth rate and fire intensity comment. We believe the reviewer has misinterpreted the figure here and mistook lower maximum doubling times as *slower* growth rate when a lower value here indicates **faster** growth rate.

L227: kit used removes (most of the) ribosomes so how can they be reliably used to do analysis?

We understand the misunderstanding here - the Takara SMARTer Stranded Total RNA-Seq Kit removes ribosomal RNA for constructing metatranscriptomic sequencing libraries, but does not remove the messenger RNA for ribosomes, which is what we are analyzing here.

L278: it is very common for bacteria to perform most energetically favorable part of a pathway and not the rest (when conditions are called for). to assume complete degradation and hand of assume mixing and connectivity in soil - which is unknown

We agree and do not claim anything about the connectivity or mixing of soil here, this

statement only says that metabolic handoffs are likely important and/or necessary for the complete degradation of catechol and protocatechuate. We are not assuming complete degradation of these compounds, only showing what genes are encoded or expressed within the dataset.

L337: methods used here is not specifically targeting viral populations. is has been shown that while valuable viral populations identified from shotgun metagenomes have their limitations

Yes, while we did not use specific virome methods to recover viromes from our soils, you can still get good viral information and data from pulling viruses from shotgun metagenomes, as evidenced by virus papers published in high-impact journals (e.g., Daly et al., 2019, Nature Microbiology⁶; Li et al., 2021, ISME⁷). Although we understand that viromes can outperform metagenomes in viral recovery⁸, we note that there are limitations to either methods, as there are in many microbiome methods in complex soils.

L415-416: these populations are shown to recover in time

We are unclear what the reviewer is asking here as this study is at one timepoint. We think they are talking about nitrifier or N fixer populations over time since fire, which we note is a short-term effect of fire in (L399).

Figure 3: there is X-Y plotting data in panel A that can be removed

We are unclear what the reviewer is asking and interpret this comment as them saying there is excess data in Figure 3A that can be removed. This data is already processed as much as possible, as here we are only showing unique compounds to each condition.

These van Krevelen diagrams are extremely common for displaying FTICR-MS data.

Reviewer #4 (Remarks to the Author):

I appreciate that the authors carefully considered my comments and the comments from the other reviewers when revising the manuscript. I have read through the revised manuscript and I think it is now significantly improved.

Decision Letter, third revision:

Dear Mike,

77Thank you for submitting your revised manuscript "A genome resolved view of wildfire legacy impacts on the soil microbiome" (NMICROBIOL-21082060C). We'll be happy in principle to publish it in Nature Microbiology, pending minor revisions to comply with our editorial and formatting guidelines.

If the current version of your manuscript is in a PDF format, please email us a copy of the file in an editable format (Microsoft Word)-- we can not proceed with PDFs at this stage.

Thank you again for your interest in Nature Microbiology Please do not hesitate to contact me if you have any questions.

Sincerely,

{redacted}

Decision Letter, final checks:

Dear Dr. Wilkins,

Thank you for your patience as we've prepared the guidelines for final submission of your Nature Microbiology manuscript, "A genome resolved view of wildfire legacy impacts on the soil microbiome" (NMICROBIOL-21082060C). Please carefully follow the step-by-step instructions provided in the attached file, and add a response in each row of the table to indicate the changes that you have made. Please also check and comment on any additional marked-up edits we have proposed within the text. Ensuring that each point is addressed will help to ensure that your revised manuscript can be swiftly handed over to our production team.

In recognition of the time and expertise our reviewers provide to Nature Microbiology's editorial

2process, we would like to formally acknowledge their contribution to the external peer review of your manuscript entitled "A genome resolved view of wildfire legacy impacts on the soil microbiome". For those reviewers who give their assent, we will be publishing their names alongside the published article.

Nature Microbiology offers a Transparent Peer Review option for new original research manuscripts submitted after December 1st, 2019. As part of this initiative, we encourage our authors to support increased transparency into the peer review process by agreeing to have the reviewer comments, author rebuttal letters, and editorial decision letters published as a Supplementary item. When you submit your final files please clearly state in your cover letter whether or not you would like to participate in this initiative. Please note that failure to state your preference will result in delays in accepting your manuscript for publication.

Cover suggestions

As you prepare your final files we encourage you to consider whether you have any images or illustrations that may be appropriate for use on the cover of Nature Microbiology.

Nature Microbiology has now transitioned to a unified Rights Collection system which will allow our Author Services team to quickly and easily collect the rights and permissions required to publish your work. Approximately 10 days after your paper is formally accepted, you will receive an email in providing you with a link to complete the grant of rights. If your paper is eligible for Open Access, our Author Services team will also be in touch regarding any additional information that may be required to arrange payment for your article.

Please note that *Nature Microbiology* is a Transformative Journal (TJ). Authors may publish their research with us through the traditional subscription access route or make their paper immediately open access through payment of an article-processing charge (APC). Authors will not be

3required to make a final decision about access to their article until it has been accepted. [Find out more about Transformative Journals](https://www.springernature.com/gp/open-research/transformative-journals)

Authors may need to take specific actions to achieve [compliance with funder and institutional open access mandates](https://www.springernature.com/gp/open-research/funding/policy-compliance-faqs). If your research is supported by a funder that requires immediate open access (e.g. according to [Plan S principles](https://www.springernature.com/gp/open-research/plan-s-compliance)) then you should select the gold OA route, and we will direct you to the compliant route where possible. For authors selecting the subscription publication route, the journal's standard licensing terms will need to be accepted, including [self-archiving policies](https://www.nature.com/nature-portfolio/editorial-policies/self-archiving-and-license-to-publish). Those licensing terms will supersede any other terms that the author or any third party may assert apply to any version of the manuscript.

Please use the following link for uploading these materials:
{redacted}

Best regards,

{redacted}

Final decision:

Dear Mike,

I am pleased to accept your Article "Wildfire-dependent changes in soil microbiome diversity and function" for publication in Nature Microbiology. Thank you for having chosen to submit your work to us and many congratulations.

4Once your paper is typeset, you will receive an email with a link to choose the appropriate publishing options for your paper and our Author Services team will be in touch regarding any additional information that may be required. Once your paper has been scheduled for online publication, the Nature press office will be in touch to confirm the details.

Acceptance of your manuscript is conditional on all authors' agreement with our publication policies (see <https://www.nature.com/nmicrobiol/editorial-policies>). In particular your manuscript must not be published elsewhere and there must be no announcement of the work to any media outlet until the publication date (the day on which it is uploaded onto our website).

Please note that *Nature Microbiology* is a Transformative Journal (TJ). Authors may publish their research with us through the traditional subscription access route or make their paper immediately open access through payment of an article-processing charge (APC). Authors will not be required to make a final decision about access to their article until it has been accepted. [Find out more about Transformative Journals](https://www.springernature.com/gp/open-research/transformative-journals)

Authors may need to take specific actions to achieve [compliance with funder and institutional open access mandates](https://www.springernature.com/gp/open-research/funding/policy-compliance-faqs). If your research is supported by a funder that requires immediate open access (e.g. according to [Plan S principles](https://www.springernature.com/gp/open-research/plan-s-compliance)) then you should select the gold OA route, and we will direct you to the compliant route where possible. For authors selecting the subscription publication route, the journal's standard licensing terms will need to be accepted, including [self-archiving policies](https://www.nature.com/nature-portfolio/editorial-policies/self-archiving-and-license-to-publish). Those licensing terms will supersede any other terms that the author or any third party may assert apply to any version of the manuscript.

An online order form for reprints of your paper is available at a

href="https://www.nature.com/reprints/author-reprints.html">https://www.nature.com/reprints/author-reprints.html. All co-authors, authors' institutions and authors' funding agencies can order reprints using the form appropriate to their geographical region.
